# PROPGCL: Unleashing the Power of Propagation in Node-Level Graph Contrastive Learning

## Abstract

Graph contrastive learning (GCL) has recently gained substantial attention, leading to the development of various methodologies. In this work, we reveal that a simple training-free propagation operator PROP, achieves competitive results over dedicatedly designed GCL methods across diverse node classification benchmarks. We elucidate PROP's effectiveness by drawing connections with established graph learning algorithms. By decoupling the propagation and transformation phases of graph neural networks, we find that the transformation weights are inadequately learned in GCL and perform no better than random on node classification. When the contrastive and downstream objects are misaligned, the attendance of transformation causes the overfitting to the contrastive loss and harms downstream performance. In light of these insights, we remove the transformation entirely and introduce an efficient GCL method termed **PROPGCL**. We provide theoretical guarantees for PROPGCL and demonstrate its effectiveness through a comprehensive evaluation of node classification tasks.

## 1 Introduction

Graph contrastive learning (GCL) has emerged as a promising paradigm for learning graph representations in an unsupervised manner. By leveraging inherent structural information, GCL has achieved state-of-the-art performance on graph learning tasks (Velickovic et al., 2019; Zhang & Chen, 2018; You et al., 2020). However, GCL often involves intricate encoders and large-scale hyperparameter tuning, raising the question of whether such complexity is necessary for effective learning.

In this work, we challenge the conventional wisdom that highly parameterized models are essential for achieving strong performance in node-level GCL. Instead, we explore a simple yet powerful alternative: uniform propagation, abbreviated as **PROP**, which involves no trainable layers. Remarkably, PROP demonstrates competitive performance on various node classification benchmarks, often matching or surpassing more complicated GCL methods. This raises two important questions:

1. *How can the training-free PROP perform so well?*

2. *Why do some existing GCL methods exhibit suboptimal performance?*

To understand why PROP can perform comparably to GCL, we position it as a non-parametric smoothing mechanism on a rewired graph through iterative optimization. Additionally, we demonstrate that PROP inherently performs alignment in contrastive learning by viewing multi-hop neighboring representations as positive samples, which elucidates the core strength in enhancing feature clustering. This analysis explains the success of PROP and highlights the potential of simpler models in GCL.

To figure out the reason behind existing GCLs' deficiency, we adopt a decoupling perspective and independently analyze the transformation and propagation phases within the GCL encoder. Our extensive analysis reveals a key limitation that existing node-level GCL methods often struggle to learn meaningful *transformation* weights, which perform no better than random counterparts. Moreover, transformation causes the learned representations to overfit to the contrastive loss. When the contrastive objective misaligns with downstream tasks, the overfitting will cause downstream degradation.

Building on these insights, we propose an efficient method, **PROPGCL**, which eliminates all transformation layers and extends the strength of PROP with graph-adaptive filters to learn flexible propagation coefficients. We provide theoretical guarantees for PROPGCL's advantage in the case where contrastive and downstream objectives are misaligned. To validate the effectiveness of PROPGCL, we conduct extensive experiments across diverse node classification benchmarks, including both homophilic and heterophilic datasets. Our results demonstrate that PROPGCL consistently outperforms existing GCL methods with appreciably fewer computational resources.

The key contributions of this work are outlined as follows:

- We establish PROP, a training-free propagation operator, as a strong baseline in graph self-supervised learning on node classification tasks. We explain its effectiveness by connecting PROP with classical graph algorithms.
- From a decoupling perspective, we reveal that existing node-level GCL methods often struggle to learn effective transformation weights. The parameter-intensive transformation causes overfitting to the contrastive loss and harms the performance when contrastive and downstream objectives are misaligned.
- We propose PROPGCL, a simple method that removes the transformation entirely and enhances PROP with graph-adaptive propagation coefficients. We provide theoretical guarantees for PROPGCL's effectiveness and rigorously evaluate PROPGCL across diverse node classification benchmarks, demonstrating its superiority over current GCL methods in terms of both accuracy and efficiency, particularly on heterophilic datasets.

## 2  RELATED WORKS

**GCL Designing Principles.** Popular GCL design approaches predominantly focus on three aspects: augmentation generation, view selection, and contrastive objectives. Augmentation strategies have been explored to enhance representation learning, such as topology-based, label-invariant, and spectral augmentations (Zhu et al., 2021b; Li et al., 2022b; Trivedi et al., 2022; Liu et al., 2022). For view selection, many works focus on hard negative mining (Robinson et al., 2021; Yang et al., 2023; Niu et al., 2024) and the necessity of positive pairs (Guo et al., 2023b). Meanwhile, contrastive objectives are often grounded in the mutual information maximization principle (Velickovic et al., 2019) or the information bottleneck principle (Xu et al., 2021). With the design complexity growing, we are concerned about whether such intricacy is truly necessary for effective graph learning. In practice, we find a training-free and propagation-only operator PROP achieves competitive results over many GCL methods (although not all GCLs), and we provide reasonable insights into its effectiveness.

**Simplifying GCL Architectures.** Recent efforts have introduced various strategies to reduce the complexity of existing methods. Some approaches remove the traditional augmentation process by employing K-means clustering, adding noise to the embedding space, or introducing invariant-discriminative losses (Yu et al., 2022; Lee et al., 2022; Li et al., 2023a). Zheng et al. (2022) simplify similarity computations by discriminating between two groups of summarized instances, rather than comparing all nodes. Li et al. (2023b) observe lower layers in deep networks suffer from degradation and propose an efficient blockwise training strategy. Other works explore using simpler models like MLPs or linear layers as the backbone encoder for GCL (Liu et al., 2023; Salha et al., 2019). However, these methods continue to rely on transformation layers that introduce additional parameters. In contrast, our method PROPGCL relies solely on a minimal-parameter propagation layer. This design reduces complexity while maintaining plug-and-play adaptability across various GCL frameworks.

## 3  BACKGROUND

### 3.1  GRAPH CONTRASTIVE LEARNING PIPELINE

The GCL pipeline often includes two stages, *pretraining* and *evaluation*. In the pretraining stage, graph views are first generated through augmentation approaches. The encoder $f$, usually defaulting to Graph Neural Networks (GNNs), embeds the graph views into node-level or graph-level representations. GCL learns the encoder weights by maximizing representation consistency between different views. The purpose of pretraining is to learn high-quality representations without relying on labeled data. In the evaluation stage, a simple linear classifier $g$ is trained in a supervised manner to map the

pretrained representations to the downstream label space. This evaluation protocol is called *linear probing*, which enables a fair comparison of representations learned by different GCL methods.

## 3.2 POLYNOMIAL GRAPH NEURAL NETWORKS

One of the foundational works of GNNs is GCN (Kipf & Welling, 2017), which propagates information from local neighborhoods and then transforms the aggregated representation in each layer by $\mathbf{H}^{(l+1)} = \sigma(\tilde{\mathbf{A}}\mathbf{H}^{(l)}\mathbf{W}^{(l)})$, where $\mathbf{H}^{(0)} = \mathbf{X}$ denotes node features, $\tilde{\mathbf{A}}$ is the normalized adjacency matrix, $\mathbf{W}^{(l)}$ is transformation weights in the $l$-th layer, and $\sigma$ is the activation function.

**Decoupled GNNs.** In GCN, propagating information and transforming representation are inherently intertwined in each layer. However, this tight coupling of operations can lead to limitations, including oversmoothing and scalability issues (Wu et al., 2019; Liu et al., 2020; Dong et al., 2021). Therefore, simpler yet effective models are proposed by decoupling the two operations (Wu et al., 2019; Gasteiger et al., 2019a; He et al., 2020). For instance, SGC (Wu et al., 2019) composes two decoupled phases of 1) *propagation* which uniformly aggregates information from $K$-hops neighboring nodes by $\mathbf{H}' = \hat{\mathbf{A}}^K\mathbf{X}$, and 2) *transformation* which transforms features by $\mathbf{H} = \sigma(\mathbf{H}'\mathbf{W})$.

**Polynomial GNNs.** Despite the simplicity of SGC and its follow-ups, the propagation procedure is fixed and shows limited expressiveness on more complicated graph structures (Balcilar et al., 2021; Nt & Maehara, 2019; Zhu et al., 2021a). To solve this, *polynomial GNNs* replace the uniform propagation with learnable combinations of polynomial basis functions to approximate arbitrary spectral filters (Chien et al., 2021; He et al., 2021; 2022). Similarly, polynomial GNNs can be expressed in a unified propagation and transformation framework,

$$\text{Propagation:} \quad \mathbf{H}_1 = \sum_{k=0}^{K-1} \theta_k g_k(\mathbf{L})\mathbf{X}, \tag{1}$$

$$\text{Transformation:} \quad \mathbf{H} = \sigma(\mathbf{H}_1\mathbf{W}), \tag{2}$$

where $\boldsymbol{\theta} \in \mathbb{R}^K$ are learnable *propagation coefficients*, $g_k(\mathbf{L})$ represents the *polynomial basis functions* applied to the graph Laplacian matrix $\mathbf{L}$, $\mathbf{W}$ is learnable *transformation weights*. Notably, the flexibility of learning spectral filters helps polynomial GNNs capture intricate structures in heterophily graphs where connected nodes tend to have different labels (He et al., 2021; 2022; Chien et al., 2021).

# 4 PROPAGATION IS A STRONG BASELINE FOR GRAPH SELF-SUPERVISED LEARNING

In this section, we demonstrate that even without trainable networks, the uniform propagation is in itself a strong baseline for graph self-supervised learning (GSSL) on node classification. We benchmark its performance on a wide range of datasets and reveal the rationale by connecting propagation to established graph learning algorithms.

## 4.1 BENCHMARK PROPAGATION AMONG GRAPH SELF-SUPERVISED LEARNING BASELINES

**Method.** We consider an operator **PROP** that aggregates features within $K$-hop neighbors:

$$\mathbf{H}_{\text{PROP}} = \hat{\mathbf{A}}^K\mathbf{X}, \tag{3}$$

where $\hat{\mathbf{A}} = \mathbf{D}'^{-\frac{1}{2}}\mathbf{A}'\mathbf{D}'^{-\frac{1}{2}}$ with $\mathbf{A}' = \mathbf{A} + \mathbf{I}$. Note that the formulation of PROP has no essential difference from SGC. We name the method *PROP* instead of *SGC* to avoid confusion with the common use of SGC in GCL literature, which often contains the transformation weights $\mathbf{W}$ and serves as the encoder (Chen & Kou, 2023; Gao et al., 2023). **Our goal is not to propose a new formulation, but to establish it as a strong training-free baseline that has long been overlooked in the GCL literature and explore the underlying rationale.**

**Datasets.** For homophily benchmarks, we choose popular citation network datasets Cora, CiteSeer, and PubMed (Sen et al., 2008; Namata et al., 2012), Amazon co-purchase datasets Photo, Computers (Shchur et al., 2018). For heterophily benchmarks, we include Wikipedia datasets Squirrel, Chameleon (Rozemberczki et al., 2021) and WebKB datasets Texas, Wisconsin, and Cornell (Pei et al., 2020).

Table 1: Test accuracy (%) of PROP and graph self-supervised (GSSL) baselines on node classification benchmarks, with blue indicating the best method, and orange the second-best.

| Training | Encoder | Homophily | | | | | | Heterophily | | | | | |
|---|---|---|---|---|---|---|---|---|---|---|---|---|---|
| | | Cora | CiteSeer | PubMed | Computers | Photo | Mean | Squirrel | Chameleon | Texas | Wisconsin | Cornell | Mean |
| Supervised | GCN | 87.5 ± 1.0 | 80.2 ± 0.6 | 87.0 ± 0.3 | 88.4 ± 0.3 | 93.5 ± 0.4 | 87.3 | 47.6 ± 0.8 | 64.1 ± 1.6 | 76.4 ± 4.1 | 62.6 ± 2.8 | 64.4 ± 4.1 | 63.0 |
| | ChebNetII | 87.2 ± 0.8 | 79.9 ± 0.8 | 88.5 ± 0.1 | 90.1 ± 0.4 | 94.9 ± 0.3 | 88.1 | 56.7 ± 1.3 | 72.3 ± 1.5 | 92.6 ± 1.8 | 89.3 ± 3.6 | 90.5 ± 1.6 | 80.3 |
| *Unsupervised Graph Embedding* | | | | | | | | | | | | | |
| DeepWalk | Word2Vec | 80.6 ± 0.8 | 63.1 ± 1.0 | 81.9 ± 0.2 | 87.3 ± 0.4 | 91.5 ± 0.5 | 80.9 | 43.3 ± 0.7 | 60.8 ± 1.3 | 53.4 ± 4.8 | 43.6 ± 4.1 | 44.6 ± 3.1 | 49.2 |
| Node2Vec | Word2Vec | 80.2 ± 1.2 | 68.1 ± 0.9 | 80.7 ± 0.3 | 85.5 ± 0.4 | 90.3 ± 0.5 | 81.0 | 39.7 ± 1.0 | 59.2 ± 1.1 | 56.2 ± 4.6 | 43.6 ± 2.8 | 45.6 ± 2.8 | 48.9 |
| *GSSL with Vanilla GCN* | | | | | | | | | | | | | |
| GRACE | GCN | 86.9 ± 1.0 | 75.6 ± 0.7 | 85.3 ± 0.2 | 82.3 ± 0.2 | 90.1 ± 0.3 | 84.0 | 43.8 ± 1.0 | 62.3 ± 0.9 | 73.6 ± 4.3 | 67.0 ± 1.8 | 65.6 ± 9.0 | 62.5 |
| DGI | GCN | 85.8 ± 1.0 | 78.6 ± 0.7 | 82.3 ± 0.3 | 79.6 ± 0.4 | 80.6 ± 1.2 | 81.4 | 37.1 ± 0.8 | 52.4 ± 1.3 | 82.6 ± 2.3 | 72.1 ± 2.4 | 80.3 ± 2.0 | 64.9 |
| GAE | GCN | 84.9 ± 1.3 | 75.7 ± 0.8 | 84.7 ± 0.3 | 76.3 ± 0.5 | 90.5 ± 0.3 | 82.4 | 36.2 ± 0.9 | 56.8 ± 1.6 | 60.0 ± 4.3 | 56.9 ± 4.9 | 57.0 ± 6.7 | 53.4 |
| VGAE | GCN | 85.1 ± 1.0 | 75.6 ± 0.7 | 84.6 ± 0.3 | 76.4 ± 0.5 | 88.3 ± 0.6 | 82.0 | 43.4 ± 0.6 | 61.4 ± 1.0 | 73.1 ± 3.4 | 60.8 ± 4.5 | 65.0 ± 7.4 | 60.8 |
| MVGRL | GCN | 84.0 ± 1.0 | 74.5 ± 0.8 | 83.6 ± 0.4 | 83.5 ± 0.5 | 89.2 ± 0.4 | 83.0 | 31.3 ± 0.6 | 57.9 ± 1.6 | 77.7 ± 2.0 | 65.8 ± 3.5 | 67.5 ± 7.9 | 60.0 |
| CCA-SSG | GCN | 86.7 ± 0.9 | 79.7 ± 0.6 | 84.8 ± 0.4 | 82.8 ± 0.3 | 91.2 ± 0.4 | 85.0 | 40.6 ± 0.7 | 57.8 ± 1.0 | 79.3 ± 3.1 | 71.1 ± 1.4 | 72.6 ± 4.9 | 64.3 |
| BGRL | GCN | 85.1 ± 0.7 | 76.5 ± 0.9 | 84.0 ± 0.2 | 82.8 ± 0.4 | 86.1 ± 0.4 | 82.9 | 36.8 ± 0.7 | 55.5 ± 1.8 | 79.7 ± 3.6 | 67.5 ± 3.9 | 71.0 ± 10.3 | 62.1 |
| GCA | GCN | 84.7 ± 1.0 | 76.5 ± 0.8 | 85.0 ± 0.2 | 79.3 ± 0.2 | 89.5 ± 0.3 | 83.0 | 41.0 ± 0.9 | 59.4 ± 1.1 | 78.0 ± 2.6 | 74.0 ± 2.1 | 66.9 ± 7.1 | 63.8 |
| ProGCL | GCN | 84.6 ± 1.0 | 78.0 ± 0.5 | 86.9 ± 0.2 | 91.2 ± 0.5 | 84.3 ± 0.4 | 85.0 | 49.5 ± 0.6 | 67.5 ± 1.1 | 77.9 ± 3.8 | 71.4 ± 2.5 | 66.6 ± 11.3 | 66.6 |
| *GSSL with Polynomial GNNs* | | | | | | | | | | | | | |
| GRACE | ChebNetII | 83.4 ± 0.9 | 74.8 ± 0.6 | 84.9 ± 0.3 | 84.1 ± 0.4 | 89.2 ± 0.5 | 83.3 | 37.9 ± 0.8 | 55.7 ± 1.0 | 77.9 ± 2.8 | 86.4 ± 3.6 | 75.7 ± 3.6 | 66.7 |
| | BernNet | 82.8 ± 1.1 | 75.4 ± 0.9 | 84.2 ± 0.2 | 85.8 ± 0.4 | 89.7 ± 0.4 | 83.6 | 40.6 ± 0.7 | 54.7 ± 1.3 | 75.4 ± 3.6 | 88.3 ± 3.1 | 74.2 ± 4.1 | 66.7 |
| | GPRGNN | 82.4 ± 1.0 | 75.4 ± 1.0 | 84.6 ± 0.3 | 81.0 ± 0.7 | 90.1 ± 0.5 | 82.7 | 38.2 ± 0.7 | 53.8 ± 1.4 | 78.7 ± 4.4 | 71.3 ± 3.9 | 77.7 ± 5.7 | 63.9 |
| DGI | ChebNetII | 83.4 ± 0.9 | 71.3 ± 1.2 | 81.9 ± 0.4 | 79.6 ± 0.3 | 78.7 ± 0.7 | 79.0 | 34.3 ± 0.6 | 51.0 ± 1.0 | 80.8 ± 2.1 | 81.8 ± 3.0 | 80.8 ± 1.6 | 65.7 |
| | BernNet | 81.5 ± 1.0 | 73.4 ± 0.5 | 82.8 ± 0.2 | 79.2 ± 0.6 | 78.3 ± 0.5 | 79.1 | 32.4 ± 0.9 | 47.4 ± 1.8 | 82.8 ± 2.1 | 78.3 ± 2.3 | 83.6 ± 2.6 | 64.9 |
| | GPRGNN | 82.4 ± 1.4 | 74.7 ± 1.0 | 80.9 ± 0.2 | 77.8 ± 0.6 | 77.8 ± 0.6 | 78.1 | 32.8 ± 0.6 | 51.0 ± 1.4 | 80.0 ± 2.0 | 70.0 ± 3.8 | 78.9 ± 3.8 | 62.5 |
| *Training-free Method* | | | | | | | | | | | | | |
| \ | **PROP** | 85.5 ± 0.8 | 78.9 ± 0.6 | 82.9 ± 0.5 | 87.5 ± 0.5 | 93.0 ± 0.3 | 85.6 | 58.5 ± 1.0 | 68.8 ± 1.4 | 86.2 ± 3.1 | 89.0 ± 3.3 | 86.2 ± 3.1 | 77.8 |

**Settings.** We consider two categories of representative GSSL methods as baselines: traditional graph embeddings and deep learning methods (graph autoencoders and contrastive learning). Given the superiority of polynomial GNNs, we also compare GCLs with polynomial GNNs. In the pretraining stage, we maintain consistency in the hyperparameter search space across methods as much as possible. In the evaluation stage, we adopt linear probing following Zhu et al. (2020b); Hassani & Khasahmadi (2020). We follow Chien et al. (2021); Chen et al. (2024) to randomly split the nodes into 60%, 20%, and 20%. Each experiment is repeated ten times with mean and standard deviation of accuracy score reported. Experiments with public fixed splitting are also conducted in Appendix D. We mainly evaluate transductive settings and also explore inductive settings on benchmarks Reddit and PPI in Appendix C. See more experimental details in Appendix V.

**Results.** As shown in Table 1, **even without training, PROP maintains a superior performance over competing methods.** For homophily benchmarks, PROP achieves comparable performances with GSSL baselines. PROP reaches an average of 85.6% while the best-performing GSSL methods have 85.0%. For heterophilic benchmarks, PROP exceeds other methods by a large margin of over 10% on average performance, including GCLs with polynomial GNNs. We hypothesize that under unsupervised signals, learning weights is more challenging for complex heterophily graphs. As further shown in Section 5.1, the learned transformation weights tend to lose informativeness. Therefore, PROP shows more improvement on heterophily graphs by removing the misleading weights. Notably, GSSL baselines often require time-intensive training and extensive hyperparameter tuning, while PROP operates without back-propagation and has only one hyperparameter, the propagation step.

## 4.2 UNDERSTANDING PROP FROM ESTABLISHED GRAPH LEARNING ALGORITHMS

Reviewing well-established graph algorithms, we can understand PROP's effectiveness by connecting it with the graph smoothing mechanism and graph alignment learning. See proofs in Appendix H.

**Propagation as non-parametric graph smoothing.** By aggregating features from neighboring nodes, cascaded propagation performs iterative representation updates. Inspired by Zhu et al. (2021a), we show in the following theorem that with an appropriate learning step, the Dirichlet energy of a rewired $k$-hop graph is minimized by propagation and converges to zero for non-bipartite graphs.

**Theorem 4.1.** *For a learning step size of $\alpha = 0.5$, the propagation operator (Equation 3) optimizes the spectral energy objective $\mathcal{L}(\mathbf{H}) = \mathbf{H}^\top (\mathbf{I} - \hat{\mathbf{A}}^k)\mathbf{H}$, which represents the Dirichlet energy on a rewired graph, where neighboring nodes are defined over $k$-hop connections.*

Based on the iterative optimization, propagation alone can be regarded as a *non-parametric* approach that smooths out the neighborhood over the $k$-hop graph, which helps explain the effectiveness

of PROP on graphs beyond just the homophilous category. Note that when the propagation step approaches infinity, node representations converge to identical values, also known as over-smoothing (Oono & Suzuki, 2020; Cai & Wang, 2020). However, the total propagation step is practically limited to a finite range, which provably improves the performance before oversmoothing kicks in (Keriven, 2022), as also supported by our experimental results.

**Propagation as graph alignment learning.** The propagation operator can also be understood as a special alignment part in contrastive learning, where positive samples are randomly drawn from *neighboring* nodes. We define the joint distribution of positive pairs as $p(\boldsymbol{x}_i, \boldsymbol{x}_j) = \hat{A}_{ij} / \sum_{i,j} \hat{A}_{ij}$, where $\hat{A}_{ij}$ denotes the normalized edge weight between node $v_i$ and node $v_j$ on the $k$-hop graph. This neighboring-node view demonstrates competitive performance in real scenarios (Lee et al., 2022; Shen et al., 2023) with further illustration in Appendix G. Based on the definition, the alignment loss is:

$$\mathcal{L}_{\text{align}}(f) = -\mathbb{E}_{\boldsymbol{x}_i, \boldsymbol{x}_j \sim p(\boldsymbol{x}_i, \boldsymbol{x}_j)}[f(\boldsymbol{x}_i)^\top f(\boldsymbol{x}_j)]. \tag{4}$$

Intuitively, this alignment objective will bring the representation of neighboring nodes together. As shown in the following theorem, propagation minimizes this alignment loss at its optimum, indicating that propagation implicitly performs the alignment in contrastive learning.

**Theorem 4.2.** *Let $f_k(\boldsymbol{x}_i) = \mathbf{H}_i^{(k)}, \forall\, i \in [N]$ be unit vectors, then $\lim_{k \to \infty} \mathcal{L}_{\text{align}}(f_k) = -1$.*

### 4.3 FURTHER INSIGHTS INTO PROP

Through a systematic re-evaluation, we establish propagation as a strong baseline within the GCL literature. PROP does not evidence that the GCL paradigm is unnecessary, but rather evidence that many current parametric designs may be overcomplicated. Below, we clarify the differences between PROP and related methods and provide further insights.

**PROP and raw node features.** A training-free option is directly using raw node features, *i.e.*, $\mathbf{X}$. However, feeding raw features into a downstream linear classifier sometimes results in degraded performance, as shown in Appendix E. We argue that propagation is essential for incorporating structural information, even in heterophily graphs, and helps particularly when node features are noisy or uninformative. See detailed discussions in Appendix F.

**PROP and random GNNs.** Early works have shown the non-trivial ability of GNNs with random weights (Kipf & Welling, 2017). The key distinction between PROP and random GNNs is whether the transformation weights are incorporated. In the unsupervised setting, random introduces noise under insufficient supervision signals. Empirically, in later sections, we reveal that incorporating random weights in GCL performs worse than eliminating them.

**PROP and Graph-Augmented MLPs (GA-MLPs).** GA-MLPs, like SGC and APPNP (Gasteiger et al., 2019a), also adopt the decoupling perspective by preprocessing raw features with graph operators and then training an MLP in a supervised manner. The key difference is that the transformation learns in this *supervised* paradigm is critical, whose removal will downgrade the performance. However, as further revealed in our work, the MLP weights are poorly learned under *unsupervised* settings and harm the downstream task. From another perspective, if we combine PROP with downstream linear-probing, they are formally equivalent to GA-MLPs. We will not resort to any wordplay on this issue. However, our intention is not to claim PROP as a novel method, but rather to highlight its value as a long-overlooked yet strong baseline in GCL literature.

**PROP on graph classification task.** We also benchmark PROP among GSSL baselines on the graph classification task. As shown in Appendix B, PROP achieves an average performance gap of 2.82% relative to the best-performing methods, a notable result given its training-free nature. We hypothesize that the slight gap arises because the single-node features do not directly map to the global graph label, necessitating advanced transformation or pooling operations. The theoretical understandings in our paper focus primarily on node connections within a single graph, aligning more closely with node classification. While PROP demonstrates some promise in graph classification, its potential in this area warrants further investigation.

## 5 DISSECTING THE LIMITATIONS OF GNNs IN GCL

To understand why existing GCL methods often fail to outperform PROP, we decouple the propagation and transformation phases, a widely adopted perspective in designing GNNs (Gasteiger et al., 2019a;b; Li et al., 2022a). Our analysis shows that GCL methods struggle to learn effective transformation weights but have promising potential in the propagation phase. Moreover, the transformation causes an overfitting on the CL objective, potentially degrading the downstream performance. This finding reveals the limitations of GCL and paves the way for more effective GCL methods.

### 5.1 FEATURE TRANSFORMATION IS INEFFECTIVE IN GCL

We first empirically compare the characteristics of the transformation weights learned by supervised learning (SL) and GCL. As revealed in Figure 1, the SL weights have a substantial variance across different neuron positions, while the GCL weights exhibit more uniform smoothness, suggesting that specific neurons in SL play pivotal roles in distinguishing features, whereas the GCL transformation learning process appears overly generalized, diminishing the richness of feature representation.

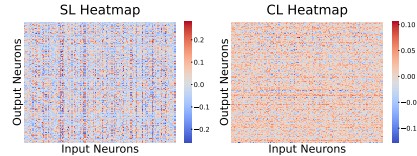

Figure 1: Characterization of the transformation weights learned by SL and GCL. Appendix T provides results of more benchmarks and GCL methods.

To further verify the ineffectiveness of the transformation weights learned by GCL, we conduct experiments by comparing them with random weights. In practice, we consider a decoupled encoder $\mathbf{H} = \sigma(\mathbf{H}_{\mathrm{PROP}}\mathbf{W})$ where $\mathbf{W}$ is the transformation weights. We compare the weights learned through GCL with a random matrix whose element is independently sampled from a Gaussian distribution. As shown in Table 2, **the transformation weights learned by GCL are not remarkably better than random counterparts**. The model with random weights attains an average performance of 73.43%, even surpassing the 72.86% reached by the transformation weights learned through GCL. We conduct comprehensive experiments by varying GCL backbones, propagators, and random initialization methods, and conclusions are consistent as detailed in Appendix I. Notably, although *random projection* (Bingham & Mannila, 2001) is well-established and proven effective in various works (Li et al., 2006; Freund et al., 2007; Bauw et al., 2021), GCL should aim to *learn* weights tailored to data, rather than relying on a random matrix. Therefore, the results indicate that many GCL methods fail to learn informative transformation weights as expected.

Table 2: Test accuracy (%) of node classification benchmarks, comparing the transformation weights learned through GCL with random weights. Blue indicates the best, while the underlined is the second best. We present the DGI method and results for more GCL methods in Appendix I.

| Training | Cora | CiteSeer | PubMed | Squirrel | Chameleon | Texas | Wisconsin | Cornell | **Mean** |
|---|---|---|---|---|---|---|---|---|---|
| GCL | 83.23 ± 0.74 | 74.24 ± 0.55 | 82.10 ± 0.33 | 45.92 ± 0.65 | 64.00 ± 1.33 | 81.15 ± 2.13 | 71.88 ± 2.50 | 80.33 ± 1.80 | 72.86 |
| Randomize $\mathbf{W}$ | 83.02 ± 0.94 | 70.04 ± 0.82 | 83.87 ± 0.53 | 49.62 ± 0.99 | 67.94 ± 1.16 | 80.33 ± 1.81 | 72.25 ± 2.25 | 80.33 ± 1.97 | 73.43 |

### 5.2 LEARNING PROPAGATION IS PROMISING IN GCL

Now, we comprehensively examine both transformation and propagation phases. While polynomial GNNs incorporate learnable parameters in both phases (Equation 1 and 2), GCLs with polynomial GNNs tend to underperform, as shown in Table 1. This issue is often attributed to the mismatch between the strong fitting capacity of polynomial filters and the lack of supervision signals (Chen et al., 2022; 2024). However, our following experiments demonstrate that GCLs can effectively learn polynomial filter coefficients.

From the decoupling perspective, there are three *conjectures* as to why polynomial GNNs underperform in GCL: (1) GCL learns ineffective transformation weights, (2) GCL learns suboptimal propagation coefficients, or (3) a combination of both. To investigate the cause, we separately replace the propagation coefficients $\boldsymbol{\theta}$ and the transformation weights $\mathbf{W}$ with well-trained parameters from the supervised setting. Specifically, we first train polynomial GNNs via supervised learning and save the optimized parameters as $\mathbf{W}_{\mathrm{SL}}$ and $\boldsymbol{\theta}_{\mathrm{SL}}$. We then proceed with the following experiments: (1). ***Fix-propagation***. Corresponding to the first conjecture, we initialize and freeze propagation coefficients with the well-trained $\boldsymbol{\theta}_{\mathrm{SL}}$, and only learn transformation weights $\mathbf{W}$ through GCL. (2).

***Fix-transformation***. Corresponding to the second conjecture, we initialize and freeze transformation weights with the well-trained $\mathbf{W}_{\mathrm{SL}}$, and only learn propagation coefficients $\theta$ through GCL. (3). ***All-one baseline***. We further consider a baseline with well-trained transformation weights $\mathbf{W}_{\mathrm{SL}}$ and a fixed all-one propagation coefficient $\mathbb{1}$.

The experimental results are summarized in Table 3. For the first conjecture, the fix-propagation model averages 72.19%, significantly lower than the supervised model's 80.41%, and sometimes even underperforms the original GCL method. It indicates that **GCL struggles to learn effective transformation weights** (like $\mathbf{W}_{\mathrm{SL}}$) **even with strong filters**. For the second conjecture, the fix-transformation model achieves an average performance of 79.65%, closely matching that of the supervised model. In contrast, the all-one baseline yields a lower accuracy of 75.56%, confirming that the learned propagation coefficients are effective. Thus, **GCL can learn informative propagation coefficients with well-trained transformation weights**. For further validation of propagation learning, in Appendix J, we conduct flip experiments by fixing parameters with GCL-trained ones and get a similar conclusion, with the learned propagation coefficients presented in Appendix U.

The observation suggests potential *few-shot learning* applications with limited ground-truth labels for training. In Appendix O, we initially explore training propagation coefficients via CL while optimizing transformation weights with supervision. However, in unsupervised settings, optimal transformation weights are unattainable. In later sections, we provide an effective GCL solution with learnable propagation only.

Table 3: Test accuracy (%) of node classification benchmarks. We freeze propagation coefficients with optimal $\theta_{\mathrm{SL}}$ and *learn* transformation weights through GCL (or the opposite). $\mathbb{1}$ denotes an all-one vector. Blue indicates the best, while underlined is the second-best.

| | $\theta$ | $\mathbf{W}$ | Cora | CiteSeer | PubMed | Squirrel | Chameleon | Texas | Wisconsin | Cornell | **Mean** |
|---|---|---|---|---|---|---|---|---|---|---|---|
| SL | $\theta_{\mathrm{SL}}$ | $\mathbf{W}_{\mathrm{SL}}$ | **88.39 ± 0.74** | 79.67 ± 0.72 | 87.11 ± 0.25 | **49.34 ± 1.09** | 69.52 ± 0.96 | 89.67 ± 2.13 | 91.25 ± 2.75 | 88.36 ± 3.11 | 80.41 |
| GCL | *Learn* | *Learn* | 83.42 ± 0.92 | 74.79 ± 0.57 | 84.92 ± 0.26 | 37.90 ± 0.79 | 55.67 ± 0.96 | 77.87 ± 2.79 | 86.38 ± 3.63 | 75.74 ± 3.61 | 72.09 |
| Fix-propagation | $\theta_{\mathrm{SL}}$ | *Learn* | 80.26 ± 0.95 | 76.15 ± 0.80 | 82.41 ± 0.64 | 40.31 ± 0.60 | 59.06 ± 1.58 | 78.69 ± 4.75 | 87.88 ± 2.75 | 72.79 ± 5.57 | 72.19 |
| Fix-transformation | *Learn* | $\mathbf{W}_{SL}$ | 87.47 ± 0.67 | **81.11 ± 0.55** | **87.69 ± 0.24** | 45.74 ± 1.57 | 64.95 ± 2.19 | **90.00 ± 2.46** | **91.38 ± 3.50** | **88.85 ± 4.10** | 79.65 |
| All-one baseline | $\mathbb{1}$ | $\mathbf{W}_{SL}$ | 78.24 ± 0.92 | 78.72 ± 0.48 | 84.75 ± 0.33 | 35.98 ± 0.77 | 59.61 ± 1.07 | 89.34 ± 3.93 | 89.38 ± 2.25 | 88.49 ± 3.77 | 75.56 |

### 5.3 TRANSFORMATION ENHANCES OVERFITTING TO CL OBJECTIVE

To explore why the transformation phase brings ineffectiveness, we compare GCL with/without transformation from the optimization perspective. We find that during training, transformation weights incur an overfitting to the contrastive learning objective, while keeping only propagation alleviates the overfitting. As demonstrated in Figure 2, GCL with transformation (the ChebNet polynomial expansion is used as the propagation operator) rapidly drives the CL training loss to near zero. In contrast, GCL without transformation maintains a moderate loss level, reflecting its resistance to over-optimizing the CL objective.

Optimizing the contrastive loss to its minimum is preferred if the pretext objective is well aligned with the downstream tasks. However, when positive samples misalign with intra-class samples, forcing InfoNCE loss to the minimum could result in a poor downstream performance, as theoretically proved in Wang et al. (2022). Lacking prior downstream knowledge, it's infeasible for GCL to select perfect positive samples, especially for heterophilic graphs with complicated structures. Thus, the overfitting to contrastive loss negatively transfers to downstream tasks.

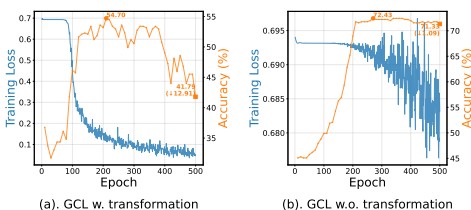

(a). GCL w. transformation  (b). GCL w.o. transformation

Figure 2: Overfitting to the contrastive loss. More examples are shown in Appendix P.

While we employ early-stopping for all baselines in Table 1, our experiments show it fails to resolve this overfitting issue. We also tried possible strategies, including $l_1$ regularization, whitening techniques (Bell & Sejnowski, 1997), and normalization methods (Hua et al., 2021; Guo et al., 2023a), but find these approaches offer limited improvement in Appendix S. Meticulously designed frameworks and advanced contrastive principles may overcome the limitations. However, for the free-structured graph data, there are no precise or even intuitive definitions of semantic equivalence (unlike images or text), bringing much difficulty into designing reasonable contrastive principles. In the following section, we propose a simple solution by

directly removing the transformation phase. Although easy in formulation, the method demonstrates competitive performances across diverse benchmarks, with a great advantage of efficiency.

# 6 PROPGCL: GRAPH CONTRASTIVE LEARNING THAT ONLY LEARNS PROPAGATION

## 6.1 PROPGCL

PROP's strong performance suggests that a simple model without transformation can achieve competitive results. However, the fixed uniform propagation has limited effectiveness in complex scenarios like heterophilic graphs. Therefore, we enhance PROP by introducing learnable graph-adaptive filter coefficients, leveraging GCL's propagation-learning potential. Specifically, for a given GCL framework, we replace the original encoder with the learnable spectral propagation,

$$\mathbf{H}_{\text{PROPGCL}} = \sum_{k=0}^{K-1} \theta_k g_k(\mathbf{L})\mathbf{X},$$ (5)

where $\boldsymbol{\theta} \in \mathbb{R}^K$ is learnable propagation coefficients, $g_k(\mathbf{L})$ represents polynomial basis functions. For clarity, we denote the revised GCL framework with the prefix *PROP*, *e.g.*, PROP-GRACE.

## 6.2 THEORETICAL ANALYSIS

We previously show that when the contrastive object misaligns with the downstream task, overfitting to the CL loss will cause performance degradation. In the following analysis, we decompose such imperfect CL loss into downstream-relevant and -irrelevant components, and prove that in such cases, our PROPGCL is guaranteed to learn better representations than PROP and the backbone GCL.

**Definition 6.1.** *(Optimal Propagation Decomposition)* Let $T^* = \arg\min_T \mathcal{L}_{\text{CL}}(T \cdot X)$ be the optimal operator for the contrastive learning loss. We assume $T^*$ is a function of bounded variation, i.e., $T^* \in BV(\Omega; \mathbb{R}^d)$. By the Jordan decomposition theorem for bounded variation mappings, $T^* = f + g$, where $f$ is the continuous component and $g$ is the discontinuous component.

**Assumption 6.2.** *(Approximation Properties)* We assume the continuous $f$ corresponds to the downstream-relevant component, *e.g.*, informative signals on the graph, while the discontinuous $g$ represents nuisance or disconnected signals. Based on Chebyshev polynomial theory, continuous functions can be well approximated by polynomials: $\inf_\theta \|f - \sum_k \theta_k A^k\|_F \leq \epsilon_f = C_f K^{-s}$, where $s > 0$ and $C_f$ is a constant. Discontinuous noises yield large polynomial approximation errors: $\inf_\theta \|g - \sum_k \theta_k A^k\|_F \geq \epsilon_g > 0$, where $\epsilon_g \gg \epsilon_f > 0$.

**Assumption 6.3.** *(Task Misalignment)* When the contrastive learning objective is misaligned with downstream tasks, we have $\|g\|_F = \alpha\|f\|_F$ with $\alpha \neq 1$.

Based on the assumptions, we have the following theorem with proof in Appendix H.

**Theorem 6.1.** *Under Assumptions 6.2 and 6.3, when $\alpha > \frac{\epsilon_f}{\|f\|_F}$, we have:*

$$\|\mathbf{H}_{\text{PROPGCL}} - f\mathbf{X}\|_F < \min\left(\|\mathbf{H}_{\text{PROP}} - f\mathbf{X}\|_F, \|\mathbf{H}_{\text{GCL}} - f\mathbf{X}\|_F\right).$$

The theory shows that when CL and downstream objectives are misaligned (large $\alpha$), PROPGCL performs better than both baselines. By learning representations that balance CL optimization with downstream relevance, PROPGCL maintains higher CL loss than GCL while achieving better downstream performance, which further explains the empirical observation in Figure 2.

## 6.3 EXPERIMENTAL RESULTS

**Benchmarks.** Besides previous benchmarks, we also consider a recently proposed heterophily benchmark (Platonov et al., 2023b) and large OGB benchmarks ogbn-arxiv and ogbn-products (Hu et al., 2020). Experimental settings are kept the same as Section 4.1.

**Baselines.** For the baseline, we include PROP, which outperforms well-known GSSL methods as outlined in Section 4.1. Additionally, we consider GCL methods specifically designed for heterophilic

graphs, including PolyGCL (Chen et al., 2024), HGRL (Chen et al., 2022), GraphACL (Xiao et al., 2024), SP-GCL (Wang et al., 2023), and DSSL (Xiao et al., 2022). Our approach builds upon GRACE and DGI as main backbones and uses the scale-friendly method GGD (Zheng et al., 2022) for large OGB graphs. We utilize the Chebyshev basis as the polynomial function and conduct an ablation study of basis choices in Appendix L. We mainly adopt the linear-probing evaluation and also estimate clustering quality of unsupervised representations detailed in Appendix M.

**Results.** The main results on node classification benchmarks are presented in Table 4. **Our method surpasses the PROP baseline and GCL methods on most benchmarks, especially for heterophily datasets where many traditional GCL methods struggle.** For homophily benchmarks, PROP-GRACE achieves the highest average accuracy of 88.76%, with PROP-DGI securing the second-highest at 88.42%. Our approach attains the best performance in 3 out of 6 benchmarks and performs comparably to the best methods in the remaining cases. For heterophily benchmarks, PROP-DGI attains an average accuracy of 73.71%, surpassing the state-of-the-art PolyGCL by a margin of 4.23%. Our method ranks first on 4 out of 6 benchmarks and second-best on the remaining two.

On the recent heterophily benchmark in Table 5, PROP-GRACE surpasses its backbone GRACE by 3.99% on average, and PROP-DGI achieves the best results in 2 out of 5 benchmarks with an average performance of 70.22%, second only to PolyGCL's 71.68%. Notably, PolyGCL is designed especially for heterophily graphs, whereas PROP-DGI builds on a more general DGI framework. On large benchmarks in Table 6, our method performs comparably with the backbone method while achieving higher efficiency. Remarkably, PROP-GGD outperforms GGD by 0.16% in accuracy on ogbn-products, accompanied by a 25.44% reduction in training time. Moreover, PROPGCL also presents better robustness on hyperparameters selection and noisy features (Appendix N).

Table 4: Test accuracy (%) of node classification benchmarks, comparing PROPGCL with PROP and GCL baselines. Blue indicates the best method, while underlined represents the second-best choice.

| Method | Homophily | | | | | | | Heterophily | | | | | | |
|---|---|---|---|---|---|---|---|---|---|---|---|---|---|---|
| | Cora | CiteSeer | PubMed | Photo | Computers | CS | Mean | Squirrel | Chameleon | Actor | Texas | Wisconsin | Cornell | Mean |
| PROP | 85.48 ± 0.75 | 78.87 ± 0.63 | 82.89 ± 0.48 | 93.01 ± 0.28 | 87.54 ± 0.47 | 95.15 ± 0.19 | 87.16 | 58.48 ± 1.03 | 68.82 ± 1.42 | 39.36 ± 0.91 | 86.23 ± 3.11 | **89.00 ± 3.25** | 86.23 ± 3.11 | 71.35 |
| GRACE | 86.90 ± 1.03 | 75.60 ± 0.71 | 85.31 ± 0.23 | 90.10 ± 0.30 | 82.29 ± 0.23 | 92.99 ± 0.18 | 85.53 | 43.78 ± 0.99 | 62.30 ± 0.94 | 37.76 ± 0.77 | 73.61 ± 4.26 | 67.00 ± 1.75 | 65.57 ± 9.02 | 58.34 |
| DGI | 85.80 ± 0.95 | 78.58 ± 0.70 | 82.27 ± 0.31 | 80.63 ± 1.15 | 79.58 ± 0.39 | 93.48 ± 0.17 | 83.39 | 37.14 ± 0.80 | 52.38 ± 1.29 | 34.44 ± 0.45 | 82.62 ± 2.30 | 72.13 ± 2.38 | 80.33 ± 1.97 | 58.84 |
| PolyGCL | 86.19 ± 0.76 | 79.07 ± 0.82 | 86.69 ± 0.24 | 92.70 ± 0.18 | 88.91 ± 0.25 | 95.30 ± 0.07 | 88.14 | 56.09 ± 0.87 | 72.17 ± 1.12 | **40.50 ± 0.78** | 86.72 ± 2.13 | 85.50 ± 4.00 | 75.90 ± 2.46 | 69.48 |
| SP-GCL | 84.68 ± 0.81 | 76.43 ± 0.63 | **86.98 ± 0.23** | 92.65 ± 0.48 | 89.04 ± 0.35 | 91.95 ± 0.24 | 86.91 | 58.11 ± 0.70 | 70.98 ± 0.90 | 30.40 ± 1.11 | 81.97 ± 2.79 | 76.00 ± 3.75 | 65.74 ± 6.39 | 63.87 |
| HGRL | 85.39 ± 1.00 | 79.84 ± 0.91 | 85.12 ± 0.30 | **93.61 ± 0.22** | 85.89 ± 0.22 | 95.57 ± 0.12 | 87.57 | 38.89 ± 0.85 | 55.69 ± 1.03 | 37.09 ± 0.68 | 84.10 ± 4.75 | 86.13 ± 3.00 | 84.59 ± 4.27 | 64.57 |
| GraphACL | 87.41 ± 1.00 | 79.17 ± 0.55 | 85.71 ± 0.27 | 92.86 ± 0.33 | 86.43 ± 0.35 | 94.17 ± 0.16 | 87.63 | 53.77 ± 0.89 | 66.94 ± 1.05 | 38.73 ± 0.86 | 84.43 ± 1.80 | 80.00 ± 2.50 | 79.51 ± 1.80 | 67.23 |
| DSSL | **87.60 ± 1.18** | 79.52 ± 1.10 | 86.62 ± 0.24 | 93.15 ± 0.46 | 88.53 ± 0.38 | 94.10 ± 0.18 | 88.25 | 47.56 ± 0.98 | 68.85 ± 3.77 | 35.64 ± 0.51 | 85.90 ± 2.62 | 79.00 ± 2.75 | 80.98 ± 2.13 | 67.77 |
| **PROP-GRACE** | 87.42 ± 0.95 | **81.56 ± 0.83** | 86.19 ± 0.35 | 93.32 ± 0.31 | 88.12 ± 0.23 | **95.95 ± 0.14** | **88.76** | 55.09 ± 0.81 | 71.73 ± 1.18 | 39.35 ± 0.81 | 89.84 ± 1.81 | 88.50 ± 3.63 | 86.72 ± 2.46 | 71.87 |
| **PROP-DGI** | 86.19 ± 1.05 | 80.78 ± 0.65 | 85.14 ± 0.22 | 92.78 ± 0.37 | **89.81 ± 0.20** | 95.82 ± 0.18 | 88.42 | **60.53 ± 0.66** | **74.11 ± 0.96** | 39.53 ± 0.84 | **91.80 ± 2.30** | 88.88 ± 2.50 | **87.38 ± 2.62** | 73.71 |

Table 5: Test accuracy (%) of recent heterophily graph benchmarks. Blue indicates the best method, while the underlined represents the second-best.

| Method | roman empire | amazon ratings | minesweeper | tolokers | questions | Mean |
|---|---|---|---|---|---|---|
| PROP | 63.95 ± 0.33 | 40.22 ± 0.22 | 74.10 ± 0.58 | 71.74 ± 0.51 | 70.23 ± 0.59 | 64.05 |
| DGI | 62.64 ± 0.22 | 38.71 ± 0.23 | 80.01 ± 0.65 | 74.95 ± 0.58 | 68.05 ± 0.61 | 64.87 |
| GRACE | 59.04 ± 0.22 | 39.79 ± 0.28 | 75.89 ± 0.50 | 74.26 ± 0.73 | 72.15 ± 0.62 | 64.22 |
| PolyGCL | 71.11 ± 0.47 | **44.09 ± 0.31** | **86.11 ± 0.41** | **83.70 ± 0.59** | 73.41 ± 0.84 | **71.68** |
| SP-GCL | 55.72 ± 0.34 | 43.02 ± 0.38 | 72.38 ± 0.64 | 76.69 ± 0.60 | 73.91 ± 0.74 | 64.34 |
| HGRL | 63.31 ± 0.33 | 39.65 ± 0.32 | 52.14 ± 0.44 | 74.34 ± 0.45 | OOM | – |
| GraphACL | 59.66 ± 0.37 | 42.68 ± 0.19 | 67.73 ± 0.72 | 74.93 ± 0.73 | 74.48 ± 0.51 | 63.90 |
| DSSL | 44.48 ± 0.33 | 40.44 ± 0.16 | 82.05 ± 0.50 | 73.88 ± 0.76 | 69.08 ± 0.82 | 61.99 |
| **PROP-GRACE** | 68.04 ± 0.25 | 42.76 ± 0.26 | 80.83 ± 0.58 | 77.51 ± 0.77 | 71.95 ± 0.92 | 68.21 (↑3.99) |
| **PROP-DGI** | **74.66 ± 0.27** | 43.14 ± 0.28 | 80.50 ± 0.62 | 77.93 ± 0.54 | **74.88 ± 0.76** | 70.22 (↑5.35) |

Table 6: Test accuracy (%) and training time on large OGB benchmarks. Train time denotes the training time per epoch in seconds.

| Benchmark | Method | Test Acc | Train Time |
|---|---|---|---|
| ogbn-arxiv | GGD | 70.26 ± 0.15 | 1.02 |
| | **PROP-GGD** | 69.71 ± 0.06 (↓ 0.55) | 0.78 (↓ 23.15%) |
| ogbn-products | GGD | 75.71 ± 0.24 | 284.39 |
| | **PROP-GGD** | 75.87 ± 0.20 (↑ 0.16) | 212.05 (↓ 25.44%) |

## 6.4 EFFICIENCY ANALYSIS

Thanks to the elimination of transformation weights, **PROPGCL demonstrates appreciable improvements in efficiency compared to its backbone methods, both in terms of computational time and memory usage,** as shown in Table 7. For instance, PROP-GRACE achieves an 84.29% reduction in training time per epoch relative to GRACE on the CS dataset. Regarding memory efficiency, PROP-GRACE consumes over 99% less memory for the encoder

Table 7: Time and space efficiency comparison. *Improvement* refers to the percentage increase in speed or decrease in memory consumption.

| Metric | Method | Photo | Computers | CS | Squirrel | Chameleon |
|---|---|---|---|---|---|---|
| Time | GRACE | 0.2872 | 0.4639 | 1.5111 | 0.7004 | 0.2295 |
| | **PROP-GRACE** | 0.2400 | 0.3626 | 0.2374 | 0.2581 | 0.1450 |
| | *Improvement* | **16.44%** | **21.84%** | **84.29%** | **63.15%** | **36.82%** |
| Memory | GRACE | 2518.04 | 2562.04 | 2562.04 | 5206.04 | 5678.04 |
| | **PROP-GRACE** | 5.86 | 6.04 | 6.04 | 16.36 | 18.21 |
| | *Improvement* | **99.77%** | **99.76%** | **99.76%** | **99.69%** | **99.68%** |

on various benchmarks. Evaluations on more benchmarks and basis functions consistently confirm the efficiency gains in Appendix Q, where we also provide a detailed time complexity analysis.

## 7 CONCLUSION

In this work, we establish PROP, a training-free propagation operator, as a strong self-supervised learning baseline for node classification, supported by linking it to established graph algorithms. From a decoupling perspective, we observe that transformation weights learned via GCL exhibit uninformative characteristics and cause an overfitting to the CL objective. To address this, we introduce a novel approach PROPGCL that focuses solely on learning propagation coefficients through GCL, achieving state-of-the-art performance across diverse node classification benchmarks. We believe this work paves the way for developing lightweight and effective GCL methods, with potential for advancing both research and practical applications in graph learning.

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

## ETHICS STATEMENT

We are not aware of any specific ethical concerns related to this work. All experiments are conducted on publicly available or synthetic datasets, without the use of sensitive or proprietary information.

## REPRODUCIBILITY STATEMENT

We provide complete details of our methods, hyperparameters, datasets, and evaluation metrics in both the main paper and the appendix. To further support transparency and reproducibility, we will release our code upon acceptance.

## THE USE OF LARGE LANGUAGE MODELS (LLMS)

In this work, LLMs are primarily employed for polishing the language of the manuscript to ensure grammatical correctness and coherence. Importantly, all conceptual development, theoretical analysis, experimental design, and result interpretation are conducted independently by the authors. The use of LLMs is strictly limited to auxiliary tasks, ensuring that the scientific contributions of this paper remain entirely unaffected by such tools.

## A    APPENDIX

## B    EXPERIMENTS OF PROP ON THE GRAPH CLASSIFICATION TASK

**Methods.** To get the global graph representation, we first aggregate node features within $K$-hop neighbors without any trainable weights, then average pool aggregated node features into a global representation, *i.e.*,

$$\mathbf{H}_{\text{PROP}} = \frac{1}{N} \sum_i \mathbf{H}_i, \quad \mathbf{H} = \hat{\mathbf{A}}^K \mathbf{X}, \tag{6}$$

where $N$ is the number of nodes, $\mathbf{H}_i$ is the representation of the node $v_i$, $\hat{\mathbf{A}} = \mathbf{D}'^{-\frac{1}{2}} \mathbf{A}' \mathbf{D}'^{-\frac{1}{2}}$ with $\mathbf{A}' = \mathbf{A} + \mathbf{I}$.

**Datasets.** We choose molecules datasets MUTAG (Debnath et al., 1991) and NCI1 (Wale et al., 2008), bioinformatics datasets PROTEINS (Borgwardt et al., 2005), and DD (Dobson & Doig, 2003), social networks IMDB-BINARY, IMDB-MULTI (Yanardag & Vishwanathan, 2015), and COLLAB (Yanardag & Vishwanathan, 2015).

**Baselines.** We consider three categories of representative methods as baselines: 1) graph kernel methods including GL (Shervashidze et al., 2009), WL (Shervashidze et al., 2011), and DGK (Yanardag & Vishwanathan, 2015), 2) traditional graph embedding methods including node2vec (Grover & Leskovec, 2016), sub2vec (Adhikari et al., 2018), and graph2vec (Narayanan et al., 2017), 3) contrastive learning methods including InfoGraph (Sun et al., 2020), GraphCL (You et al., 2020), MVGRL (Hassani & Khasahmadi, 2020), JOAOv2 (You et al., 2021), ADGCL (Suresh et al., 2021).

**Settings.** Following You et al. (2020), we train the model in an unsupervised manner and feed the learned representation into a downstream SVM classifier. To keep comparison fairness, we tune hyperparameters in a unified combination, and keep the search space among methods as consistent as possible. Details can be found in Appendix V.

**Results.** As shown in Table 8, PROP surpasses most graph kernels and traditional embeddings and performs comparably with GCL methods. PROP achieves an average performance gap of 2.82% relative to the best-performing methods, a notable result given its training-free nature. We hypothesize that the slight gap arises because the single-node features do not directly map to the global graph label, necessitating advanced transformation or pooling operations. Another optional choice is utilizing Laplacian positional embeddings or random-walk embeddings as widely discussed in the literature of Graph Transformers (Yun et al., 2019; Ying et al., 2021; Rampášek et al., 2022). We leave deeper research on the graph classification task for future work.

Table 8: Test accuracy (%) of graph classification benchmarks, comparing PROP and GSSL methods. The compared results are from published papers, and − indicates that results are unavailable. We report the performance gap between one method and the best method, averaged across datasets in the **Mean Gap** column. **Bold** indicates the best method, while underlined represents the second-best.

|  | PROTEINS | MUTAG | DD | NCI1 | IMDB-B | IMDB-M | COLLAB | **Mean Gap** ↓ |
|---|---|---|---|---|---|---|---|---|
| | | | *Graph Kernel* | | | | | |
| GL | − | 81.66 ± 2.11 | − | − | 65.87 ± 0.98 | − | − | 7.60 |
| WL | 72.92 ± 0.56 | 80.72 ± 3.00 | − | 80.01 ± 0.50 | 72.30 ± 3.44 | − | − | 2.88 |
| DGK | 73.30 ± 0.82 | 87.44 ± 2.72 | − | **80.31 ± 0.46** | 66.96 ± 0.56 | − | − | 2.37 |
| | | | *Traditional Graph Embedding* | | | | | |
| node2vec | 57.49 ± 3.57 | 72.63 ± 10.20 | − | 54.89 ± 1.61 | − | − | − | 16.61 |
| sub2vec | 53.03 ± 5.55 | 61.05 ± 15.80 | − | 52.84 ± 1.47 | 55.26 ± 1.54 | − | − | 19.79 |
| graph2vec | 73.30 ± 2.05 | 83.15 ± 9.25 | − | 73.22 ± 1.81 | 71.10 ± 0.54 | − | − | 3.54 |
| | | | *Graph Contrastive Learning* | | | | | |
| MVGRL | − | 75.40 ± 7.80 | − | − | 63.60 ± 4.20 | − | − | 11.87 |
| InfoGraph | **74.44 ± 0.31** | 89.01 ± 1.13 | 72.85 ± 1.78 | 76.20 ± 1.06 | **73.03 ± 0.87** | 48.66 ± 0.67 | 70.65 ± 1.13 | 2.07 |
| GraphCL | 74.39 ± 0.45 | 86.80 ± 1.34 | **78.62 ± 0.40** | 77.87 ± 0.41 | 71.14 ± 0.44 | 48.49 ± 0.63 | 71.36 ± 1.15 | **1.52** |
| JOAOv2 | 74.07 ± 1.10 | 87.67 ± 0.79 | 77.40 ± 1.15 | 78.36 ± 0.53 | 70.83 ± 0.25 | − | 69.33 ± 0.34 | 1.78 |
| ADGCL | 73.81 ± 0.46 | **89.70 ± 1.03** | 75.10 ± 0.39 | 69.67 ± 0.51 | 72.33 ± 0.56 | **49.89 ± 0.66** | **73.32 ± 0.61** | 2.21 |
| **PROP** | 71.07 ± 0.30 | 87.44 ± 1.53 | 78.39 ± 0.37 | 75.24 ± 0.14 | 71.22 ± 0.28 | 47.11 ± 0.18 | 69.07 ± 0.05 | 2.82 |

## C  EXPERIMENTS OF PROP IN THE INDUCTIVE SETTING

We conduct experiments in the inductive setting on the single-graph dataset Reddit and the multiple-graph dataset PPI. The experimental settings, including data splitting and training hyperparameters, follow those in Hamilton et al. (2017). The results are summarized in Table 9. For PPI (a multi-graph benchmark with 50-dimensional node features), PROP ($K=2$) achieves an F1 score of 0.7527, which is comparable to GRACE's score of 0.7548. For Reddit, PROP ($K=2$) achieves an F1 score of 0.8452, outperforming GRACE which achieves 0.8185. These results validate the effectiveness of PROP in node classification tasks under the inductive setting.

Table 9: F1 score comparison of PROP and GRACE on benchmarks PPI and Reddit. **Bold** indicates the best, while underlined represents the second-best choice.

| Method | F1 Score (PPI) | F1 Score (Reddit) |
|---|---|---|
| GRACE | **0.7548** | 0.8185 |
| PROP ($K=0$) | 0.7076 | 0.5852 |
| PROP ($K=1$) | 0.7493 | 0.8457 |
| PROP ($K=2$) | 0.7527 | **0.8452** |

## D  EXPERIMENTS OF PROP WITH A FIXED PUBLIC-SPLITTING.

In Section 4.1, we evaluate PROP and graph self-supervised methods on the node classification task with a random splitting. To ensure that the conclusion is not limited to a specific split setting, we evaluate the models on the publicly available fixed splits following Zhu et al. (2021c); Zhang et al. (2021). In practice, we use the public splitting introduced in Pei et al. (2020) for most datasets. There is no available public splitting for Amazon-Photo and Amazon-Computers, so we randomly split the dataset into 1/1/8 as the train/validation/test set, differing from the splitting in Section 4.1. Other experimental settings are kept the same. As shown in Table 10, on 6 in 10 benchmarks, PROP performs the best among baselines and exceeds the runner-up ProGCL by 4.23% on average. The results verify the effectiveness of PROP in different data-splitting cases.

Table 10: Test accuracy (%) of PROP and other graph self-supervised methods on node classification benchmarks with the public splitting. **Bold** indicates the best method, while underlined represents the second-best choice.

| Method | Cora | CiteSeer | PubMed | Photo | Computers | Squirrel | Chameleon | Texas | Wisconsin | Cornell | Mean |
|---|---|---|---|---|---|---|---|---|---|---|---|
| DeepWalk | 80.87 ± 1.07 | 63.14 ± 1.05 | 81.55 ± 0.27 | 84.66 ± 0.40 | 89.59 ± 0.18 | 43.32 ± 0.79 | 60.81 ± 1.27 | 53.44 ± 5.09 | 43.63 ± 4.25 | 44.59 ± 2.95 | 64.56 |
| Node2Vec | 84.27 ± 0.70 | 66.04 ± 1.83 | 81.33 ± 0.36 | 83.92 ± 0.31 | 89.31 ± 0.20 | 38.41 ± 1.19 | 59.50 ± 2.30 | 60.81 ± 1.89 | 55.10 ± 3.73 | 60.54 ± 3.24 | 67.92 |
| GAE | 85.96 ± 1.03 | 72.78 ± 1.11 | 85.06 ± 0.49 | 75.29 ± 0.53 | 89.50 ± 0.26 | 35.56 ± 1.27 | 56.51 ± 1.62 | 62.43 ± 4.86 | 61.18 ± 3.53 | 60.27 ± 3.51 | 68.45 |
| VGAE | 86.20 ± 0.76 | 73.26 ± 0.65 | 85.19 ± 0.43 | 72.17 ± 0.33 | 86.90 ± 0.38 | 42.38 ± 1.13 | 60.29 ± 1.05 | 63.78 ± 3.51 | 59.61 ± 2.75 | 60.54 ± 2.16 | 69.03 |
| GRACE | 84.10 ± 1.01 | 70.41 ± 0.92 | 84.79 ± 0.38 | 78.51 ± 0.44 | 87.80 ± 0.41 | 39.65 ± 0.87 | 55.83 ± 1.05 | 64.59 ± 4.59 | 58.82 ± 4.91 | 60.81 ± 2.16 | 68.53 |
| DGI | 87.20 ± 0.99 | 72.50 ± 1.49 | 82.55 ± 0.38 | 71.35 ± 0.57 | 80.43 ± 0.63 | 36.61 ± 1.05 | 52.02 ± 1.32 | 70.54 ± 2.97 | 63.53 ± 3.92 | 61.62 ± 2.16 | 67.84 |
| MVGRL | 83.44 ± 0.72 | 71.61 ± 0.73 | 82.48 ± 0.30 | 80.96 ± 0.67 | 86.87 ± 0.41 | 31.48 ± 0.83 | 58.77 ± 1.45 | 68.38 ± 2.98 | 62.94 ± 3.53 | 61.62 ± 2.16 | 68.86 |
| CCA-SSG | **87.71 ± 0.75** | **75.42 ± 0.80** | 85.55 ± 0.40 | 78.96 ± 0.33 | **90.91 ± 0.38** | 40.16 ± 0.74 | 54.98 ± 1.18 | 68.65 ± 3.78 | 64.12 ± 4.31 | 61.89 ± 2.43 | 70.84 |
| BGRL | 85.77 ± 0.89 | 72.66 ± 1.54 | 84.63 ± 0.49 | 74.43 ± 0.91 | 85.50 ± 0.59 | 37.20 ± 1.07 | 53.82 ± 1.67 | 67.03 ± 2.70 | 60.59 ± 3.14 | 60.81 ± 2.43 | 68.24 |
| GCA | 86.60 ± 0.79 | 74.71 ± 1.18 | 86.44 ± 0.34 | 75.63 ± 0.46 | 88.77 ± 0.54 | 41.33 ± 0.88 | 59.28 ± 1.54 | 69.46 ± 2.97 | 62.94 ± 2.75 | 61.89 ± 2.16 | 70.71 |
| ProGCL | 85.45 ± 0.85 | 73.61 ± 1.10 | **86.86 ± 0.41** | 81.64 ± 0.70 | 89.91 ± 0.31 | 50.23 ± 0.86 | 67.81 ± 1.47 | 69.46 ± 2.97 | 62.75 ± 2.75 | 61.35 ± 1.35 | 72.91 |
| PROP | 84.57 ± 0.82 | 74.55 ± 1.09 | 84.65 ± 0.24 | **84.78 ± 0.38** | 90.83 ± 0.34 | **57.20 ± 1.41** | **68.71 ± 1.18** | **71.35 ± 4.60** | **79.61 ± 3.14** | **75.14 ± 3.78** | **77.14** |

## E  EXPERIMENTS OF PROP WITH DIFFERENT AGGREGATION STEPS

In this section, we present the accuracy of PROP with different propagation steps. We find that the best step choice varies among datasets, but a shallow propagation is enough in most cases. As shown in Figure 3, only one-step propagation performs best in datasets including Cora, CiteSeer, Chameleon, Squirrel, Computers, and Photo. For Texas, Wisconsin, Cornell, Actor, and CS, the raw features, (*i.e.*, zero propagation step) are enough. Moreover, when the performance achieves its best, raising the propagation step will cause a degradation.

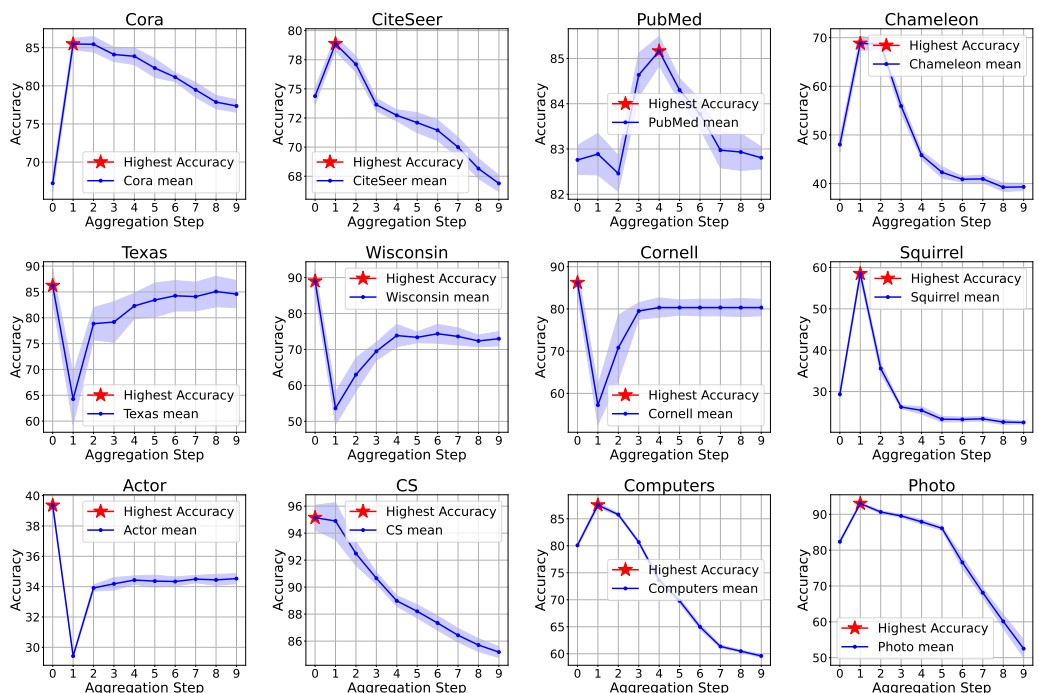

Figure 3: Accuracy (%) of PROP with different propagation steps. We mark the best step choice with a red star. Experiments are conducted ten times and the shadow denotes the derivation.

## F  COMPARISON BETWEEN PROP AND RAW FEATURES

### F.1  GRAPH STRUCTURE AS EFFECTIVE SUPERVISED SIGNALS

The taxonomy of homophily and heterophily is commonly used to assess whether the graph structure is informative for training GCN-like models. However, beyond this traditional dichotomy, recent metrics characterizing graph properties have been proposed, showing a closer relationship with GNN performance (Mao et al., 2023; Luan et al., 2023; Platonov et al., 2023a). For instance, Ma et al. (2021) observe that the inter-class similarity on the Squirrel dataset is slightly higher than the intra-class similarity for most classes, which helps explain the moderate performance of GCN on this dataset.

However, the performance of GCN-like models is influenced by the interplay between graph structure and node features. Therefore, poor performance of GCN does not necessarily imply that the graph structure is ineffective, nor does it imply the opposite. To verify this, we design experiments based on the mutual information between labels and graph elements, including graph structure and node features. To decouple the effects of structure and node features, we use an MLP instead of a GCN as the training model, with node features $\mathbf{X}$, adjacency matrix $\mathbf{A}$, and their concatenation as inputs, respectively.

The results are shown in Table 11. Surprisingly, **for some heterophily datasets, MLPs using the graph structure as input achieve satisfactory performance**. For instance, on the Squirrel dataset, which has a low homophily ratio of 0.22, the MLP based on the graph structure achieves an accuracy of 73.58%. This suggests that, even with a low homophily ratio, the graph structure can still serve as a highly effective supervision signal for label prediction.

### F.2  NODE FEATURE PERTURBATION EXPERIMENTS

PROP demonstrates significant advantages over Raw Features (RF), particularly in scenarios where node features are noisy or uninformative. To illustrate this, we compare PROP and RF under noise-perturbation and masking-perturbation settings. For noise-perturbation, Gaussian noise is added

Table 11: Test accuracy (%) of MLP with different input signals on node classification benchmarks. $\mathcal{H}(G)$ denotes the edge homophily ratio introduced in Zhu et al. (2020a). Lower $\mathcal{H}(G)$ denotes graphs with a high heterophily level. $[,]$ denotes concatenation. **Bold** indicates the best, while underlined represents the second-best choice.

| | Cora | CiteSeer | PubMed | Chameleon | Squirrel | Actor |
|---|---|---|---|---|---|---|
| $\mathcal{H}(G)$ | 0.81 | 0.74 | 0.80 | 0.23 | 0.22 | 0.22 |
| MLP($\mathbf{X}$) | 73.64 | 70.72 | 85.75 | 49.34 | 35.06 | **36.51** |
| MLP($\mathbf{A}$) | 78.27 | 57.81 | 81.41 | **77.41** | **73.58** | 21.84 |
| MLP($[\mathbf{X}, \mathbf{A}]$) | **82.29** | **73.57** | **85.83** | 71.05 | 67.63 | 31.84 |

to the original node features. For masking-perturbation, we randomly mask the channels of node features with varying mask ratios in [20%, 40%, 60%, 80%]. As shown in Table 12 and Table 13, PROP consistently outperforms RF across various benchmarks when node features are perturbed. For instance, in the noise-perturbation setting, PROP achieves an average performance improvement of over 33% compared to RF. Similarly, in the masking-perturbation setting, PROP maintains its superiority even with a mask ratio as high as 80%.

Table 12: Test accuracy (%) of noise-perturbed node classification benchmarks, comparing PROP and Raw Features (RF). We add noise from a normal distribution onto the original features to generate randomly noisy node features. **Bold** indicates the best method.

| Method | Cora | CiteSeer | PubMed | Photo | Computers | Squirrel | Chameleon | **Mean** |
|---|---|---|---|---|---|---|---|---|
| RF | $39.90 \pm 6.85$ | $32.31 \pm 8.47$ | $57.28 \pm 5.69$ | $42.60 \pm 7.57$ | $54.57 \pm 6.27$ | $21.34 \pm 1.03$ | $25.47 \pm 2.47$ | 39.07 |
| PROP | $\mathbf{76.73 \pm 2.02}$ | $\mathbf{69.25 \pm 2.44}$ | $\mathbf{81.50 \pm 2.00}$ | $\mathbf{73.76 \pm 11.58}$ | $\mathbf{70.23 \pm 7.74}$ | $\mathbf{48.94 \pm 6.14}$ | $\mathbf{69.39 \pm 2.15}$ | **69.97** |

Table 13: Test accuracy (%) of mask-perturbed node classification benchmarks, comparing PROP and Raw Features (RF). We randomly mask a proportion of features to generate perturbed node features. **Bold** indicates the best method.

| Mask ratio | Method | Cora | CiteSeer | PubMed | Photo | Computers | Squirrel | Chameleon | **Mean** |
|---|---|---|---|---|---|---|---|---|---|
| 20% | RF | $54.01 \pm 3.40$ | $60.34 \pm 4.24$ | $70.00 \pm 4.66$ | $65.87 \pm 5.16$ | $68.59 \pm 4.98$ | $28.37 \pm 0.67$ | $41.77 \pm 2.78$ | 55.56 |
| | PROP | $\mathbf{76.19 \pm 3.76}$ | $\mathbf{71.87 \pm 2.68}$ | $\mathbf{83.85 \pm 0.99}$ | $\mathbf{89.78 \pm 1.51}$ | $\mathbf{83.37 \pm 2.18}$ | $\mathbf{47.13 \pm 4.50}$ | $\mathbf{64.40 \pm 2.45}$ | **73.80** |
| 40% | RF | $49.10 \pm 2.61$ | $44.68 \pm 9.49$ | $58.36 \pm 5.81$ | $50.62 \pm 9.53$ | $53.56 \pm 9.74$ | $25.67 \pm 1.97$ | $34.99 \pm 4.88$ | 45.28 |
| | PROP | $\mathbf{61.25 \pm 6.68}$ | $\mathbf{54.87 \pm 10.25}$ | $\mathbf{76.85 \pm 4.43}$ | $\mathbf{76.16 \pm 10.29}$ | $\mathbf{64.66 \pm 10.61}$ | $\mathbf{38.68 \pm 5.98}$ | $\mathbf{53.90 \pm 6.67}$ | **60.91** |
| 60% | RF | $46.95 \pm 5.67$ | $36.10 \pm 8.12$ | $55.88 \pm 4.87$ | $44.29 \pm 7.96$ | $53.85 \pm 7.58$ | $23.22 \pm 2.27$ | $30.72 \pm 4.09$ | 41.57 |
| | PROP | $\mathbf{54.47 \pm 6.93}$ | $\mathbf{42.59 \pm 10.70}$ | $\mathbf{63.68 \pm 9.19}$ | $\mathbf{60.27 \pm 14.32}$ | $\mathbf{60.69 \pm 8.46}$ | $\mathbf{28.47 \pm 6.50}$ | $\mathbf{41.03 \pm 8.97}$ | **50.17** |
| 80% | RF | $48.33 \pm 3.69$ | $30.18 \pm 5.64$ | $52.01 \pm 3.18$ | $41.47 \pm 5.78$ | $57.87 \pm 2.63$ | $21.93 \pm 2.04$ | $28.42 \pm 3.13$ | 40.03 |
| | PROP | $\mathbf{49.06 \pm 6.39}$ | $\mathbf{33.77 \pm 9.83}$ | $\mathbf{57.89 \pm 8.73}$ | $\mathbf{57.89 \pm 8.73}$ | $\mathbf{60.37 \pm 5.14}$ | $\mathbf{26.35 \pm 5.38}$ | $\mathbf{34.64 \pm 9.06}$ | **44.90** |

# G  INTUITIVE ILLUSTRATION OF NEIGHBORING-NODE VIEW

Using neighboring nodes can be understood as a form of view generation in GCL. Formally, this involves designing a permutation matrix $\mathbf{P}$ that transforms the graph such that $\mathbf{A}' = \mathbf{P}^\top \mathbf{A} \mathbf{P}$ and $\mathbf{X}' = \mathbf{P}\mathbf{X}$. The same row of $\mathbf{X}$ (or $\mathbf{A}$) and $\mathbf{X}'$ (or $\mathbf{A}'$) corresponds to neighboring nodes in the original graph. This kind of view generation is also applied in previous works and shows satisfying experimental performance (Lee et al., 2022; Shen et al., 2023).

Consider a simple example of a triangle graph with three nodes $v_1$, $v_2$, and $v_3$, connected as $(v_1, v_2)$, $(v_1, v_3)$ $(v_2, v_3)$. A specific permutation $\mathbf{P} = \begin{pmatrix} 0 & 1 & 0 \\ 0 & 0 & 1 \\ 1 & 0 & 0 \end{pmatrix}$ transforms the original graph's adjacency

matrix $\mathbf{A} = \begin{pmatrix} 0 & 1 & 1 \\ 1 & 0 & 1 \\ 1 & 1 & 0 \end{pmatrix}$, $\mathbf{X} = \begin{pmatrix} \boldsymbol{x}_1 \\ \boldsymbol{x}_2 \\ \boldsymbol{x}_3 \end{pmatrix}$ into $\mathbf{A}' = \mathbf{P}^\top \mathbf{A} \mathbf{P} = \begin{pmatrix} 0 & 1 & 1 \\ 1 & 0 & 1 \\ 1 & 1 & 0 \end{pmatrix}$, $\mathbf{X}' = \mathbf{P}\mathbf{X} = \begin{pmatrix} \boldsymbol{x}_2 \\ \boldsymbol{x}_3 \\ \boldsymbol{x}_1 \end{pmatrix}$.

The corresponding nodes in $\mathcal{G} = (\mathbf{A}, \mathbf{X})$ and $\mathcal{G}' = (\mathbf{A}', \mathbf{X}')$ form positive pairs.

Based on random sampling, other choices of $\mathbf{P}$ are possible, such as transforming $\mathbf{X} = (\boldsymbol{x}_1, \boldsymbol{x}_2, \boldsymbol{x}_3)^\top$ to $\mathbf{X}' = (\boldsymbol{x}_3, \boldsymbol{x}_1, \boldsymbol{x}_2)^\top$. For node $v_1$, the probabilities of transferring to $v_2$ and $v_3$ are equal. When the sampling process is repeated sufficiently, the positive samples $(v_1, v_2)$ and $(v_1, v_3)$ are sampled with approximately equal frequency, corresponding to the neighboring set in the propagation procedure.

More formally, consider the alignment loss defined in the paper,

$$\mathcal{L}_{\text{align}}(f) = -\mathbb{E}_{\boldsymbol{x}_i, \boldsymbol{x}_j \sim p(\boldsymbol{x}_i, \boldsymbol{x}_j)}[f(\boldsymbol{x}_i)^\top f(\boldsymbol{x}_j)].$$

Here, the probability distribution $p(\boldsymbol{x}_i, \boldsymbol{x}_j) = \hat{A}_{ij} / \sum_{i,j} \hat{A}_{ij}$ is defined as the normalized edge weight between nodes $v_i$ and $v_j$ in the $k$-hop graph. When the sampling process is efficient, we can approximate the neighbor sets in the propagation as positive pairs.

# H    PROOF OF THEOREMS

## H.1    PROOF OF THEOREM 4.1

Here we present the proof of Theorem 4.1, restated for reference.

**Theorem 4.1.** *For a learning step size of $\alpha = 0.5$, the propagation operator (Equation 3) optimizes the spectral energy objective $\mathcal{L}(\mathbf{H}) = \mathbf{H}^\top (\mathbf{I} - \hat{\mathbf{A}}^k)\mathbf{H}$, which represents the Dirichlet energy on a rewired graph, where neighboring nodes are defined over $k$-hop connections.*

*Proof.* We consider the rewired $k$-hop graph with the adjacency matrix denoted as $\tilde{\mathbf{A}} = \hat{\mathbf{A}}^k$. The Dirichlet energy on the $k$-hop graph is $\mathcal{L}(\mathbf{H}) = \mathbf{H}^\top \tilde{\mathbf{L}} \mathbf{H}$, where $\tilde{\mathbf{L}} = \mathbf{I} - \tilde{\mathbf{A}}$. The gradient update of the Dirichlet energy objective gives the following update rule of node features $\mathbf{H}$,

$$\mathbf{H} - \alpha \frac{\partial \mathcal{L}(\mathbf{H})}{\partial \mathbf{H}} = \mathbf{H} - 2\alpha \tilde{\mathbf{L}} \mathbf{H} = ((1 - 2\alpha)\mathbf{I} + 2\alpha \tilde{\mathbf{A}})\mathbf{H}, \tag{7}$$

where the $\alpha$ is the step size. When we choose the learning rate $\alpha = 0.5$, we recover the propagation operation in Equation 3, *i.e.*, $\mathbf{H}_{\text{new}} = \tilde{\mathbf{A}}\mathbf{H} = \hat{\mathbf{A}}^k \mathbf{H}$.

$\square$

## H.2    PROOF OF THEOREM 4.2

Here we present the proof of Theorem 4.2, restated for reference.

**Theorem 6.1.** *Under Assumptions 6.2 and 6.3, when $\alpha > \frac{\epsilon_f}{\|f\|_F}$, we have:*

$$\|\mathbf{H}_{PROPGCL} - f\mathbf{X}\|_F < \min\left(\|\mathbf{H}_{PROP} - f\mathbf{X}\|_F, \|\mathbf{H}_{GCL} - f\mathbf{X}\|_F\right).$$

*Proof.* Again, we consider the rewired $k$-hop graph with the adjacency matrix denoted as $\tilde{\mathbf{A}} = \hat{\mathbf{A}}^k$. A key step is to notice that the alignment objective Equation 4 is closely relevant to the Dirichlet energy when $f(\boldsymbol{x}_i) = \mathbf{H}_i, \forall i \in [N]$ :

$$\mathcal{L}_{\text{align}}(f) = -\sum_{i,j} \tilde{A}_{ij}[\mathbf{H}_i^\top \mathbf{H}_j]/(\sum_{i,j} \tilde{A}_{ij}) = \mathbf{H}^\top \tilde{\mathbf{A}} \mathbf{H}/(\sum_{i,j} \tilde{A}_{ij}) = \mathbf{H}^\top (\mathbf{I} - \tilde{\mathbf{L}})\mathbf{H}/(\sum_{i,j} \tilde{A}_{ij}). \tag{8}$$

It is easy to see that graph convolution converges to identical vectors, known as oversmoothing. Therefore, we have $\forall i, j, (\mathbf{H}_\infty)_i = (\mathbf{H}_\infty)_j$. Therefore,

$$\lim_{k \to \infty} \mathcal{L}_{\text{align}}(f_k) = \mathbf{H}_\infty^\top \tilde{\mathbf{A}} \mathbf{H}_\infty/(\sum_{i,j} \tilde{A}_{ij}) = (\sum_{i,j} \tilde{A}_{ij})/(\sum_{i,j} \tilde{A}_{ij}) = -1,$$

which concludes the proof.

$\square$

### H.3 PROOF OF THEOREM 6.1

Here we present the proof of Theorem 6.1, restated for reference.

**Theorem 6.1.** *Under Assumptions 6.2 and 6.3, when $\alpha > \frac{\epsilon_f}{\|f\|_F}$, we have:*

$$\|\mathbf{H}_{PROPGCL} - f\mathbf{X}\|_F < \min\left(\|\mathbf{H}_{PROP} - f\mathbf{X}\|_F, \|\mathbf{H}_{GCL} - f\mathbf{X}\|_F\right).$$

*Proof.* Since GCL has sufficient capacity to fit $T^* = f + g$, at convergence we have:

$$\|\mathbf{H}_{\text{GCL}} - (f + g)\mathbf{X}\|_F \approx 0.$$

However, for downstream performance, we care about proximity to $fX$, *i.e.*,

$$\|\mathbf{H}_{\text{GCL}} - f\mathbf{X}\|_F = \|\mathbf{H}_{\text{GCL}} - (f + g)\mathbf{X} + g\mathbf{X}\|_F.$$

By applying the triangle inequality, we obtain:

$$\|\mathbf{H}_{\text{GCL}} - f\mathbf{X}\|_F \geq \|g\mathbf{X}\|_F - \|\mathbf{H}_{\text{GCL}} - (f + g)\mathbf{X}\|_F.$$

When GCL overfits to the CL loss, it yields:

$$\|\mathbf{H}_{\text{GCL}} - f\mathbf{X}\|_F \geq \|g\mathbf{X}\|_F = \alpha\|f\|_F\|\mathbf{X}\|_F. \tag{9}$$

According to the Chebyshev approximation theory, continuous functions admit exponentially fast polynomial approximation, while discontinuous mappings incur large approximation error (Xu et al., 2019; Rahaman et al., 2019). Let $\hat{\theta}$ be the learned parameters. Then we have

$$\|\mathbf{H}_{\text{PROPGCL}} - f\mathbf{X}\|_F = \left\|\sum_k \hat{\theta}_k \mathbf{A}^k \mathbf{X} - f\mathbf{X}\right\|_F \leq \epsilon_f \|\mathbf{X}\|_F. \tag{10}$$

PROP is a special case of PROPGCL by letting $\theta_K = 1, \theta_i = 0, i \neq K$. Therefore, PROP satisfies $\|\mathbf{A}^k - f\|_F \geq \delta$ with $\delta > \epsilon_f$, leading to:

$$\|\mathbf{H}_{\text{PROP}} - f\mathbf{X}\|_F = \|\mathbf{A}^K \mathbf{X} - f\mathbf{X}\|_F \geq \delta\|\mathbf{X}\|_F. \tag{11}$$

From Equation 9, Equation 10, and Equation 11, we obtain that: For PROPGCL and PROP, we have $\epsilon_f\|\mathbf{X}\|_F < \delta\|\mathbf{X}\|_F$. This holds since $\epsilon_f < \delta$.

For PROPGCL and GCL, we have $\epsilon_f\|\mathbf{X}\|_F < \alpha\|f\|_F\|\mathbf{X}\|_F$. This holds when $\alpha > \frac{\epsilon_f}{\|f\|_F}$.

Therefore, under the stated conditions, we finally have:

$$\|\mathbf{H}_{\text{PROPGCL}} - f\mathbf{X}\|_F < \min\left(\|\mathbf{H}_{\text{PROP}} - f\mathbf{X}\|_F, \|\mathbf{H}_{\text{GCL}} - f\mathbf{X}\|_F\right),$$

which ends the proof. $\qquad\square$

## I EXPERIMENTS ON GCL WITH RANDOM WEIGHTS

In Section 5.1, we show that in the DGI method, after replacing the trained transformation weights with a random Gaussian matrix, the downstream performance does not deteriorate as expected. We conclude that the transformation weights learned in GCL are not better than random. To enhance the generalizability of our conclusion, we extended our experimental evaluations to include more GCL methods, propagators, and initialization methods. The experimental settings are kept the same.

**Variants on GCL methods.** Table 14 shows the results using the GRACE and BGRL methods. For GRACE, replacing the transformation weights with random weights raises the performance from 73.93% to 74.51% on average. For BGRL, the replacement brings an increase of more than 2% in average performance.

**Variants on initialization methods.** We compare GCL weights with four random initializations: Gaussian, Uniform, Kaiming (He et al., 2015), Xavier (Glorot & Bengio, 2010)). Table 15 shows that all randomized weights perform comparably to (even slightly better than) GCL-trained weights, confirming the GCL weights deficiency.

**Variants on the propagators.** We consider an alternative APPNP-like propagator (Gasteiger et al., 2019a):

$$\mathbf{H}_{\text{APPNP}} = (1 - \alpha)\mathbf{A}^k\mathbf{X} + \alpha\mathbf{X},$$

where $\alpha$ is the teleport (or restart) probability. As shown in Table 16, for the APPNP propagator, GCL-learned weights still show no significant advantage over different random weights.

Although we can not exhaustively try all GCL random variants, the results of representative variants above are able to verify that many GCL methods fail to learn effective transformation weights.

Table 14: Test accuracy (%) of node classification benchmarks with GRACE and BGRL methods, comparing the GCL-learned transformation weights and random weights. **Bold** indicates the best-performing weights in each GCL method.

| Method | Weights | Cora | CiteSeer | PubMed | Squirrel | Chameleon | Texas | Wisconsin | Cornell | **Mean** |
|---|---|---|---|---|---|---|---|---|---|---|
| GRACE | GCL-learned | **83.15 ± 0.82** | **74.97 ± 0.56** | **81.53 ± 0.25** | 48.46 ± 0.95 | 67.24 ± 1.42 | **84.75 ± 2.95** | 70.88 ± 2.00 | 80.49 ± 2.13 | 73.93 |
| | Randomize **W** | 82.91 ± 0.72 | 69.93 ± 0.59 | 81.39 ± 0.40 | **53.82 ± 0.79** | **69.67 ± 1.01** | 84.59 ± 2.79 | **73.25 ± 1.38** | 80.49 ± 2.30 | **74.51** |
| BGRL | GCL-learned | **83.27 ± 0.79** | 73.40 ± 0.93 | **81.36 ± 0.29** | 40.43 ± 0.77 | 65.07 ± 0.96 | 81.97 ± 3.11 | **73.38 ± 2.25** | 80.00 ± 2.13 | 72.36 |
| | Randomize **W** | 82.43 ± 0.44 | **73.85 ± 0.74** | 80.77 ± 0.28 | **54.12 ± 0.67** | **71.40 ± 1.16** | **84.59 ± 3.11** | 71.38 ± 5.25 | **80.33 ± 1.97** | **74.86** |

Table 15: Test accuracy (%) of node classification benchmarks with DGI method, comparing the transformation weights learned and random weights initialized in different methods. **Bold** indicates the best method, while underlined is the second-best.

| Training | Cora | CiteSeer | PubMed | Squirrel | Chameleon | Texas | Wisconsin | Cornell | **Mean** |
|---|---|---|---|---|---|---|---|---|---|
| GCL | **83.23 ± 0.74** | **74.24 ± 0.55** | 82.10 ± 0.33 | 45.92 ± 0.65 | 64.00 ± 1.33 | 81.15 ± 2.13 | 71.88 ± 2.50 | 80.33 ± 1.80 | 72.86 |
| Gaussian-random | 83.02 ± 0.94 | 70.04 ± 0.82 | **83.87 ± 0.53** | **49.62 ± 0.99** | 67.94 ± 1.16 | 80.33 ± 1.81 | 72.25 ± 2.25 | 80.33 ± 1.97 | 73.43 |
| Uniform-random | 82.63 ± 1.05 | 70.63 ± 1.13 | 83.38 ± 0.50 | 44.49 ± 1.03 | **68.42 ± 0.92** | 82.62 ± 2.62 | 73.25 ± 2.25 | **80.82 ± 1.80** | 73.28 |
| Kaiming-random | 82.46 ± 0.71 | 69.09 ± 0.71 | 83.68 ± 0.32 | 44.99 ± 0.63 | 68.42 ± 1.53 | 82.46 ± 2.79 | **75.75 ± 3.38** | 80.66 ± 1.97 | **73.44** |
| Xavier-random | 82.45 ± 0.74 | 68.90 ± 0.74 | 83.56 ± 0.43 | 45.02 ± 0.64 | 68.34 ± 1.47 | **82.95 ± 2.30** | 75.13 ± 1.75 | 80.82 ± 1.97 | 73.40 |

Table 16: Test accuracy (%) of node classification benchmarks with DGI method and APPNP propagator, comparing the GCL-learned transformation weights and different random weights. **Bold** indicates the best method, while underlined is the second-best.

| Training | Cora | CiteSeer | PubMed | Squirrel | Chameleon | Texas | Wisconsin | Cornell | **Mean** |
|---|---|---|---|---|---|---|---|---|---|
| GCL | 84.79 ± 0.80 | 75.47 ± 0.76 | 82.25 ± 0.24 | 40.74 ± 0.61 | **58.99 ± 1.40** | 80.33 ± 1.97 | 87.00 ± 2.50 | 80.33 ± 1.80 | 73.74 |
| Gaussian-random | **85.42 ± 0.99** | 76.49 ± 0.55 | **84.85 ± 0.16** | **45.76 ± 0.69** | 58.95 ± 1.23 | **82.79 ± 3.28** | **88.50 ± 2.63** | **83.44 ± 3.61** | **75.78** |
| Uniform-random | 85.34 ± 0.84 | **76.81 ± 0.68** | 84.60 ± 0.24 | 43.87 ± 0.90 | 58.42 ± 0.96 | 78.69 ± 2.62 | 86.88 ± 1.25 | 78.52 ± 2.46 | 74.14 |
| Kaiming-random | 83.23 ± 1.00 | 75.68 ± 0.79 | 83.76 ± 0.15 | 39.31 ± 0.91 | 55.89 ± 1.44 | 81.15 ± 4.59 | 87.25 ± 3.25 | 81.15 ± 4.43 | 73.43 |
| Xavier-random | 83.02 ± 0.69 | 75.28 ± 0.61 | 83.10 ± 0.19 | 38.55 ± 0.86 | 55.60 ± 1.20 | 81.15 ± 4.10 | 87.63 ± 3.13 | 78.69 ± 6.23 | 72.88 |

## J    Flip CL-SL experiments in Section 5

In the flip experiment, we first train the network parameters via GCL and save the learned transformation weights $\mathbf{W}_{\text{CL}}$ and propagation coefficients $\boldsymbol{\theta}_{\text{CL}}$. We then proceed with the following experiments:

**Experiment 1 (Fix-transformation).** We initialize and freeze transformation weights with the GCL-trained $\mathbf{W}_{\text{CL}}$, and only learn propagation coefficients $\boldsymbol{\theta}$ through supervised learning.

**Experiment 2 (Fix-propagation).** We initialize and freeze propagation coefficients with the GCL-trained $\boldsymbol{\theta}_{\text{SL}}$, and only learn transformation weights $\mathbf{W}$ through supervised learning.

**Experiment 3 (All-one baseline).** We further consider a baseline with GCL-trained transformation weights $\mathbf{W}_{\text{CL}}$ and a fixed all-one propagation coefficients $\mathbb{1}$.

As shown in Table 17, despite using the propagation coefficients learned via GCL, the model still achieves satisfying performances of 77.57%, compared to the original supervised model with 80.41%. However, after replacing the transformation weights with GCL-learned ones, the performance deteriorates largely with an accuracy of only 65.01%. The results further confirm our conclusion in Section 5.2 that GCL learns effectively during the propagation phase.

Table 17: Test accuracy (%) of node classification benchmarks. We freeze the propagation coefficients with $\boldsymbol{\theta}_{\mathrm{CL}}$ (or the transformation weights with $\mathbf{W}_{\mathrm{CL}}$), and *learn* the transformation weights (or propagation coefficients) in the supervised setting. $\mathbb{1}$ denotes an all-one vector. **Bold** indicates the best, while underlined represents the second-best choice.

| Method | $\theta$ | W | Cora | CiteSeer | PubMed | Squirrel | Chameleon | Texas | Wisconsin | Cornell | **Mean** |
|---|---|---|---|---|---|---|---|---|---|---|---|
| SL | *Learn* | *Learn* | **88.39 ± 0.74** | **79.67 ± 0.72** | **87.11 ± 0.25** | **49.34 ± 1.09** | **69.52 ± 0.96** | **89.67 ± 2.13** | **91.25 ± 2.75** | **88.36 ± 3.11** | **80.41** |
| CL | $\theta_{\mathrm{CL}}$ | $\mathbf{W}_{\mathrm{CL}}$ | 83.42 ± 0.92 | 74.79 ± 0.57 | 84.92 ± 0.26 | 37.90 ± 0.79 | 55.67 ± 0.96 | 77.87 ± 2.79 | 86.38 ± 3.63 | 75.74 ± 3.61 | 72.09 |
| Fix-transformation | *Learn* | $\mathbf{W}_{\mathrm{CL}}$ | 76.62 ± 2.12 | 76.25 ± 0.64 | 83.32 ± 0.46 | 36.56 ± 0.61 | 52.41 ± 2.06 | 60.16 ± 6.39 | 75.25 ± 4.38 | 59.51 ± 5.08 | 65.01 |
| Fix-propagation | $\theta_{\mathrm{CL}}$ | *Learn* | 87.06 ± 0.53 | 79.55 ± 0.74 | 85.76 ± 0.23 | 41.44 ± 1.06 | 64.44 ± 0.74 | 87.38 ± 2.95 | 90.63 ± 3.00 | 84.26 ± 2.62 | 77.57 |
| All-one baseline | $\mathbb{1}$ | *Learn* | 71.74 ± 3.22 | 75.92 ± 0.61 | 79.38 ± 0.47 | 33.27 ± 0.61 | 42.32 ± 0.90 | 55.41 ± 4.43 | 74.13 ± 4.13 | 60.82 ± 6.56 | 61.65 |

## K  DETAILS ABOUT POLYNOMIAL GNNS

In this section, we introduce polynomial GNNs from the spectral perspective. Developed from graph signal processing, *graph convolution* means transforming the graph signals to the Fourier domain and then back to the vertex domain after suitable filtering, *i.e.*, $\mathbf{H} = \mathbf{U}g_\theta(\boldsymbol{\Lambda})\mathbf{U}^\top\mathbf{X}$, where $g_\theta$ is the filter, $\mathbf{U}$ is the matrix of eigenvectors of graph Laplacian $\mathbf{L}$, $\boldsymbol{\Lambda}$ is the diagonal matrix of eigenvalues. The problem arises when the parameters in $g_\theta(\boldsymbol{\Lambda})$ are entirely unconstrained, leading to a lack of spatial localization in the convolution and a high time complexity due to eigenvalue decomposition.

These issues can be overcome with the use of a polynomial filter $g_\theta(\boldsymbol{\Lambda}) = \sum_{k=0}^{K-1} \theta_k \boldsymbol{\Lambda}^k$, where the parameter $\boldsymbol{\theta} \in \mathbb{R}^K$ is a vector of polynomial coefficients. Therefore, the graph convolution can be reformulated as $\mathbf{H} = (\sum_{k=0}^{K-1} \theta_k \mathbf{L}^k)\mathbf{X}$. We call GNNs using the polynomial approximated filters as *polynomial GNNs*. As one of the pioneer works, ChebNet (Defferrard et al., 2016) uses Chebyshev polynomial parametrization to localize filters as $g_\theta(\boldsymbol{\Lambda}) = \sum_{k=0}^{K} \theta_k T_k(\tilde{\boldsymbol{\Lambda}})$, where $\tilde{\boldsymbol{\Lambda}} = 2\boldsymbol{\Lambda}/\lambda_{\max} - \mathbf{I}$, $\boldsymbol{\theta}$ is the Chebyshev coefficients, and $T_k(\tilde{\boldsymbol{\Lambda}})$ is the Chebyshev polynomial of order $k$ recursively calculated by $T_k(x) = 2xT_{k-1}(x) - T_{k-2}(x)$ with $T_0(x) = 1$ and $T_1(x) = x$.

In Section 6, we consider three popular polynomial GNN variants. GPRGNN (Chien et al., 2021) uses the monomial basis functions evaluated at $\hat{\mathbf{A}}$, *i.e.*, $g_\theta(\boldsymbol{\Lambda}) = \sum_{k=0}^{K-1} \theta_k (\mathbf{I} - \hat{\mathbf{L}})^k$ with $\boldsymbol{\theta}$ as learnable coefficients. BernNet (He et al., 2021) uses the Bernstein polynomial approximation, *i.e.*, $g_\theta(\boldsymbol{\Lambda}) = \sum_{k=0}^{K-1} \theta_k \frac{1}{2^k} \binom{K}{k} (2\mathbf{I} - \mathbf{L})^{K-k} \mathbf{L}^k$ with $\boldsymbol{\theta}$ as learnable coefficients. ChebNetII (He et al., 2022) enhances the original Chebyshev polynomial approximation by Chebyshev interpolation, formulated as $g_\theta(\boldsymbol{\Lambda}) = \frac{2}{K+1} \sum_{k=0}^{K} \sum_{j=0}^{K} \theta_j T_k(x_j) T_k(\hat{\mathbf{L}})$, where $x_j = \cos((j + 1/2)\pi/(K + 1))$ are the Chebyshev nodes of $T_{K+1}$, and $\boldsymbol{\theta}$ are learnable coefficients.

## L  BASIS POLYNOMIAL FUNCTIONS ANALYSIS OF PROPGCL

Polynomial GNNs variants mainly differ in the polynomial basis function choices, *e.g.*, the monomial basis in GPRGNN (Chien et al., 2021), the Bernstein basis in BernNet (He et al., 2021), and the Chebyshev basis in ChebNetII (He et al., 2022). We have introduced detailed basis function formulations in Appendix K.

In this section, we compare different basis polynomial functions used in PROPGCL. Here, we consider the Chebyshev basis, Bernstein basis, and monomial basis. As shown in Table 18 and Table 19, the performance of PROPGCL is relatively robust in the choice of basis functions. For homophily benchmarks, PROP-GRACE with Chebyshev basis and the PROP-DGI with monomial basis achieve the best, surpassing the second slightly by 0.05% on average. For heterophily benchmarks, the best PROP-DGI with the Chebyshev basis achieves 73.71% on average, and the Bernstein basis ranks second. In general, the Chebyshev basis is preferred in PROPGCL.

## M  CLUSTERING QUALITY ESTIMATION

To exclude the impact of linear-probing, we also evaluate the clustering quality of raw features and representations learned by GRACE and PROP-GRACE. We conduct KMeans on unsupervised representations and esitimate two clustering metrics *Clustering Accuracy* and *Normalized Mutual Information (NMI)*. As shown in Table 20 and Table 21, PROP-GRACE outperforms both baselines

Table 18: Test accuracy (%) of homophily node classification benchmarks, comparing different basis polynomial functions in PROPGCL. **Bold** indicates the best method, while underlined represents the second-best choice.

| Method | Basis | Cora | CiteSeer | PubMed | Photo | Computers | CS | **Mean** |
|---|---|---|---|---|---|---|---|---|
| | Chebyshev | 87.42 ± 0.95 | 81.56 ± 0.83 | 86.19 ± 0.35 | 93.32 ± 0.31 | 88.12 ± 0.23 | 95.95 ± 0.14 | **88.76** |
| PROP-GRACE | Bernstein | **87.52 ± 1.20** | 81.69 ± 0.86 | 85.90 ± 0.25 | 93.42 ± 0.24 | 87.77 ± 0.22 | **95.97 ± 0.13** | 88.71 |
| | monomial | 87.34 ± 1.13 | **81.86 ± 0.79** | 86.41 ± 0.23 | 93.19 ± 0.26 | 86.85 ± 0.34 | 95.91 ± 0.15 | 88.59 |
| | Chebyshev | 86.19 ± 1.05 | 80.78 ± 0.65 | 85.14 ± 0.22 | 92.78 ± 0.37 | **89.81 ± 0.20** | 95.82 ± 0.18 | 88.42 |
| PROP-DGI | Bernstein | 86.49 ± 0.99 | 80.93 ± 0.72 | 85.80 ± 0.40 | 93.53 ± 0.26 | 89.77 ± 0.25 | 95.46 ± 0.16 | 88.66 |
| | monomial | 86.86 ± 1.02 | 81.69 ± 0.86 | **86.56 ± 0.33** | **93.72 ± 0.25** | 88.18 ± 0.34 | 95.57 ± 0.14 | **88.76** |

Table 19: Test accuracy (%) of heterophily node classification benchmarks, comparing different basis polynomial functions in PROPGCL. **Bold** indicates the best method, while underlined represents the second-best choice.

| Method | Basis | Squirrel | Chameleon | Actor | Texas | Wisconsin | Cornell | **Mean** |
|---|---|---|---|---|---|---|---|---|
| | Chebyshev | 55.09 ± 0.81 | 71.73 ± 1.18 | 39.35 ± 0.81 | 89.84 ± 1.81 | 88.50 ± 3.63 | 86.72 ± 2.46 | 71.87 |
| PROP-GRACE | Bernstein | 48.51 ± 0.85 | 70.02 ± 0.88 | 39.33 ± 0.81 | 90.16 ± 1.31 | 89.00 ± 3.25 | **88.52 ± 2.95** | 70.92 |
| | monomial | 51.96 ± 0.69 | 69.28 ± 1.05 | 39.52 ± 0.89 | 84.43 ± 2.62 | 84.13 ± 4.50 | 88.20 ± 2.79 | 69.59 |
| | Chebyshev | **60.53 ± 0.66** | **74.11 ± 0.96** | **39.53 ± 0.84** | 91.80 ± 2.30 | 88.88 ± 2.50 | 87.38 ± 2.62 | **73.71** |
| PROP-DGI | Bernstein | 53.08 ± 0.83 | 71.20 ± 0.81 | 39.48 ± 0.77 | 92.46 ± 1.48 | **91.63 ± 3.00** | 87.38 ± 2.63 | 72.54 |
| | monomial | 56.65 ± 0.77 | 72.12 ± 0.72 | 37.80 ± 0.57 | **93.11 ± 1.80** | 83.63 ± 5.88 | 81.97 ± 2.95 | 70.88 |

on average, demonstrating better clustering effectiveness. Compared to the state-of-the-art performance in linear probing, PROP-GRACE fails to consistently surpass GRACE across all benchmarks. Therefore, we recommend adopting PROPGCL in a CL+linear-probing use case, *i.e.*, training a simple linear classifier on the unsupervised representations in downstream tasks.

Table 20: Clustering Accuracy (%) of node classification benchmarks, comparing Raw Features (RF), GRACE, and PROP-GRACE. **Bold** indicates the best method, while underlined is the second-best.

| | Cora | CiteSeer | PubMed | Squirrel | Computers | Photo | Chameleon | Texas | Wisconsin | Cornell | **Mean** |
|---|---|---|---|---|---|---|---|---|---|---|---|
| RF | 30.06 | 37.60 | 59.86 | 38.21 | 37.52 | 20.32 | 23.36 | 44.26 | **51.79** | **44.26** | 38.72 |
| GRACE | 43.24 | 56.36 | **64.68** | 31.06 | **47.22** | 24.51 | 26.75 | **46.45** | 43.03 | 32.24 | 41.55 |
| PROP-GRACE | **51.81** | **67.45** | 61.39 | **39.97** | 46.01 | **31.46** | **29.07** | **46.45** | 41.83 | 41.53 | **45.70** |

# N    ROBUSTNESS COMPARISON

## N.1    NOISY FEATURES SENSITIVITY ANALYSIS

In Appendix F, we evaluate PROP's performance under node feature perturbations. Here, we extend this analysis to PROPGCL (using PROP-GRACE as a representative) and compare it against two baselines: raw features (RF) and PROP. We examine two perturbation scenarios- 1). *Noise Perturbation*: Gaussian noise is added to the original node features to generate noisy inputs; 2). *Masking Perturbation*: Random channels of the node features are masked at varying ratios in 20%, 40%, 60%, and 80%.

As shown in Tables 22 and 23, PROP-GRACE exhibits significantly stronger robustness compared to both RF and PROP. Specifically, it outperforms RF by >30% on noise-perturbed features and maintains consistent improvements across all masking ratios. These results highlight the advantages of PROP-GRACE's on noisy or low-dimensional features.

Table 21: NMI of node classification benchmarks, comparing Raw Features (RF), GRACE, and PROP-GRACE. **Bold** indicates the best method, while underlined is the second-best.

| | Cora | CiteSeer | PubMed | Squirrel | Computers | Photo | Chameleon | Texas | Wisconsin | Cornell | **Mean** |
|---|---|---|---|---|---|---|---|---|---|---|---|
| RF | 0.1031 | 0.1504 | 0.3105 | 0.2231 | 0.2567 | 0.0040 | 0.0123 | **0.2018** | **0.3738** | **0.2018** | 0.1838 |
| GRACE | 0.3476 | 0.3166 | 0.2257 | 0.2179 | **0.4584** | 0.0150 | 0.0163 | 0.1897 | 0.2382 | 0.0345 | 0.2060 |
| PROP-GRACE | **0.3623** | **0.4136** | **0.3380** | **0.3071** | 0.4039 | **0.0818** | **0.0885** | 0.1491 | 0.1044 | 0.0536 | **0.2302** |

Table 22: Test accuracy (%) of noise-perturbed node classification benchmarks, comparing Raw Features (RF), PROP and PROP-GRACE. We add noise from a normal distribution onto the original features to generate randomly noisy node features. **Bold** indicates the best method.

| Method | Cora | CiteSeer | PubMed | Photo | Computers | Squirrel | Chameleon | **Mean** |
|---|---|---|---|---|---|---|---|---|
| RF | $39.90 \pm 6.85$ | $32.31 \pm 8.47$ | $57.28 \pm 5.69$ | $42.60 \pm 7.57$ | $54.57 \pm 6.27$ | $21.34 \pm 1.03$ | $25.47 \pm 2.47$ | 39.07 |
| PROP | $76.73 \pm 2.02$ | $69.25 \pm 2.44$ | **$81.50 \pm 2.00$** | $73.76 \pm 11.58$ | $70.23 \pm 7.74$ | $48.94 \pm 6.14$ | **$69.39 \pm 2.15$** | 69.97 |
| PROP-GRACE | **$80.77 \pm 0.92$** | **$70.85 \pm 1.20$** | $81.17 \pm 0.29$ | **$80.07 \pm 0.48$** | **$72.06 \pm 0.67$** | **$58.47 \pm 0.72$** | $67.79 \pm 1.20$ | **73.03** |

### N.2 HYPERPARAMETER SENSITIVITY ANALYSIS

In this section, we undertake a hyperparameter sensitivity analysis to compare PROPGCL with its GCL backbone counterpart. The investigation entails manipulating a spectrum of hyperparameters to assess their impact on performance metrics. Specifically, we focus on two pivotal hyperparameters within the model architecture: the hidden dimension and the number of propagation steps. Figure 4 illustrates that the performance of DGI is notably sensitive to perturbations in hyperparameters. For instance, on the Cora dataset, a reduction in the hidden dimension from 256 to 128 results in a substantial accuracy decrement of approximately 40%. Conversely, as shown in Figure 5, the robustness of PROP-DGI is evident across various hyperparameter configurations, with a sharp decline only observed when using small neural networks.

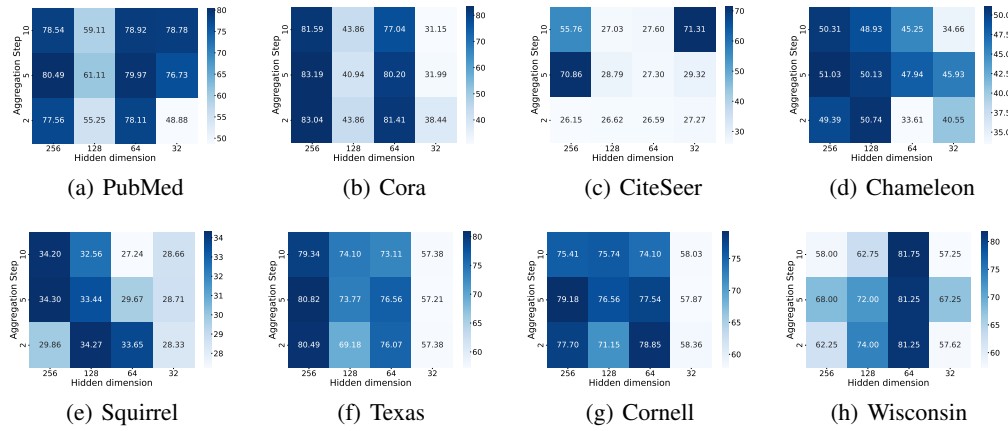

Figure 4: Hyperparameter sensitivity analysis of DGI with ChebNetII as the encoder. We evaluate the performances by varying the hidden dimension and propagation step.

## O TRIALS IN THE FEW-SHOT LEARNING SETTING

In Section 5, we observe that GCL has the potential to learn good propagation coefficients given well-trained transformation weights. It inspires methods in the *few-shot* scenario, where a model is tasked with achieving effective generalization from a minimal number of labeled examples per class.

In this study, we examine the $N$-shot case, where $N$ denotes the number of examples per class used for training and is commonly chosen as 3 or 5. For our approach, we train the propagation coefficients via GCL and then focus on optimizing the transformation weights supervisedly on the given support

Table 23: Test accuracy (%) of modified node classification benchmarks, comparing Raw Features (RF), PROP, and PROP-GRACE. We randomly mask a proportion of features to generate perturbed node features. **Bold** indicates the best method, while underlined represents the second-best.

| Mask ratio | Method | Cora | CiteSeer | PubMed | Photo | Computers | Squirrel | Chameleon | **Mean** |
|---|---|---|---|---|---|---|---|---|---|
| 20% | RF | $54.01 \pm 3.40$ | $60.34 \pm 4.24$ | $70.00 \pm 4.66$ | $65.87 \pm 5.16$ | $68.59 \pm 4.98$ | $28.37 \pm 0.67$ | $41.77 \pm 2.78$ | 55.56 |
| | PROP | $76.19 \pm 3.76$ | $71.87 \pm 2.68$ | **$83.85 \pm 0.99$** | **$89.78 \pm 1.51$** | **$83.37 \pm 2.18$** | $47.13 \pm 4.50$ | $64.40 \pm 2.45$ | 73.80 |
| | PROP-GRACE | **$80.36 \pm 0.84$** | **$73.27 \pm 0.66$** | $82.12 \pm 0.16$ | $88.00 \pm 0.42$ | $79.19 \pm 0.50$ | **$56.93 \pm 0.48$** | **$67.37 \pm 1.40$** | 75.32 |
| 40% | RF | $49.10 \pm 2.61$ | $44.68 \pm 9.49$ | $58.36 \pm 5.81$ | $50.62 \pm 9.53$ | $53.56 \pm 9.74$ | $25.67 \pm 1.97$ | $34.99 \pm 4.88$ | 45.28 |
| | PROP | $61.25 \pm 6.68$ | $54.87 \pm 10.25$ | $76.85 \pm 4.43$ | $76.16 \pm 10.29$ | $64.66 \pm 10.61$ | $38.68 \pm 5.98$ | $53.90 \pm 6.67$ | 60.91 |
| | PROP-GRACE | **$80.79 \pm 1.07$** | **$73.78 \pm 0.86$** | **$81.55 \pm 0.18$** | **$87.38 \pm 0.50$** | **$71.29 \pm 0.29$** | **$53.21 \pm 0.53$** | **$64.38 \pm 1.09$** | 73.20 |
| 60% | RF | $46.95 \pm 5.67$ | $36.10 \pm 8.12$ | $55.88 \pm 4.87$ | $44.29 \pm 7.96$ | $53.85 \pm 7.58$ | $23.22 \pm 2.27$ | $30.72 \pm 4.09$ | 41.57 |
| | PROP | $54.47 \pm 6.93$ | $42.59 \pm 10.70$ | $63.68 \pm 9.19$ | $60.27 \pm 14.32$ | $60.69 \pm 8.46$ | $28.47 \pm 6.50$ | $41.03 \pm 8.97$ | 50.17 |
| | PROP-GRACE | **$78.39 \pm 1.13$** | **$72.01 \pm 1.11$** | **$79.13 \pm 0.20$** | **$78.87 \pm 0.50$** | **$70.06 \pm 0.87$** | **$47.06 \pm 0.85$** | **$63.76 \pm 1.18$** | 69.90 |
| 80% | RF | $48.33 \pm 3.69$ | $30.18 \pm 5.64$ | $52.01 \pm 3.18$ | $41.47 \pm 5.78$ | $57.87 \pm 2.63$ | $21.93 \pm 2.04$ | $28.42 \pm 3.13$ | 40.03 |
| | PROP | $49.06 \pm 6.39$ | $33.77 \pm 9.83$ | $57.89 \pm 8.73$ | $57.89 \pm 8.73$ | $60.37 \pm 5.14$ | $26.35 \pm 5.38$ | $34.64 \pm 9.06$ | 44.90 |
| | PROP-GRACE | **$60.20 \pm 1.40$** | **$63.83 \pm 1.13$** | **$65.29 \pm 0.44$** | **$71.38 \pm 1.04$** | **$64.85 \pm 0.98$** | **$38.84 \pm 1.13$** | **$55.80 \pm 1.44$** | 60.03 |

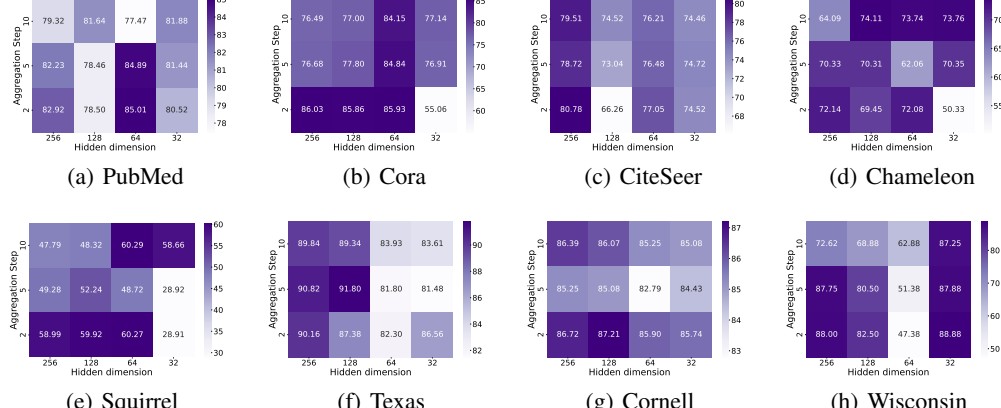

Figure 5: Hyperparameter sensitivity analysis of PROP-DGI with the Chebyshev basis. We evaluate the performances by varying the hidden dimension and propagation step.

examples. The method is termed as `Fix-prop SL`. For the baseline, we consider the ChebNetII models trained via supervised learning (SL) and contrastive learning (CL).

As illustrated in Table 24, this approach yields improvements on several benchmarks. For instance, `Fix-prop SL` enhances SL accuracy from 57.51% to 72.60% on Cora in the 5-shot case, and from 39.19% to 65.39% in the 3-shot case. However, the `Fix-prop SL` approach has minimal impact on the Squirrel and Chameleon datasets. The results demonstrate the potential of integrating SL and CL from a decoupling perspective in the few-shot scenario. Notably, we keep hyperparameters consistent across all training methods and benchmarks, leaving ample room for further exploration beyond this initial investigation.

Table 24: , comparing models trained with SL, CL, and Fix-prop SL settings. **Bold** indicates the best, while underlined represents the second-best choice.

| | Training | Cora | CiteSeer | PubMed | Squirrel | Chameleon |
|---|---|---|---|---|---|---|
| 5 Shot | SL | $57.51 \pm 2.29$ | $43.11 \pm 3.75$ | $59.62 \pm 2.56$ | $20.15 \pm 0.30$ | $22.09 \pm 1.60$ |
| | CL | $66.88 \pm 2.29$ | **$55.02 \pm 4.64$** | $63.20 \pm 2.64$ | **$28.41 \pm 0.87$** | **$36.92 \pm 2.52$** |
| | Fix-prop SL | **$72.60 \pm 1.43$** | $53.26 \pm 4.03$ | **$67.66 \pm 2.58$** | $20.60 \pm 0.90$ | $23.30 \pm 1.91$ |
| 3 Shot | SL | $39.19 \pm 3.96$ | $37.52 \pm 2.25$ | $55.89 \pm 2.55$ | $20.27 \pm 0.55$ | $21.40 \pm 1.26$ |
| | CL | $64.46 \pm 4.34$ | **$55.85 \pm 5.15$** | $59.88 \pm 3.49$ | **$25.89 \pm 1.54$** | **$36.12 \pm 1.34$** |
| | Fix-prop SL | **$65.39 \pm 2.15$** | $46.90 \pm 3.40$ | **$61.46 \pm 5.49$** | $20.38 \pm 0.69$ | $27.85 \pm 3.02$ |

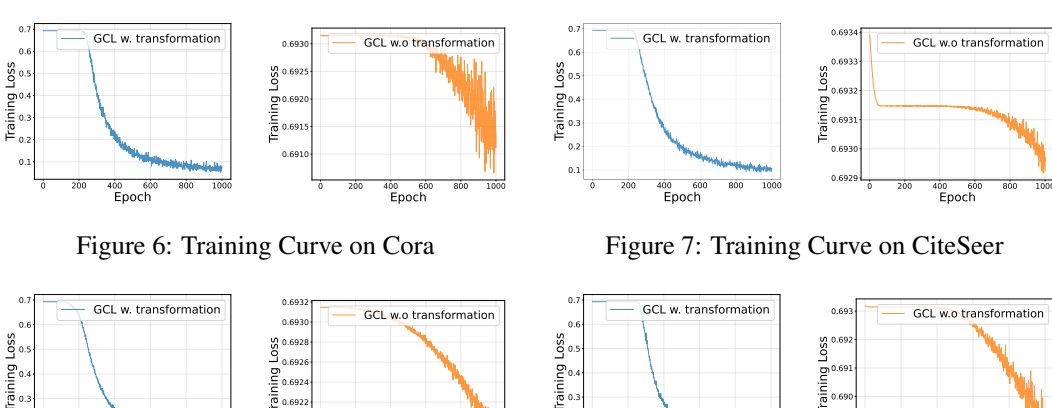

Figure 6: Training Curve on Cora

Figure 7: Training Curve on CiteSeer

Figure 8: Training Curve on Computers

Figure 9: Training Curve on Photo

## P  CONTRASTIVE TRAINING LOSS CURVES

As demonstrated in Figure 6 to Figure 9, across multiple benchmarks, GCL with transformation rapidly drives the CL training loss to near zero. In contrast, GCL without transformation maintains a moderate loss level, reflecting its resistance to over-optimizing the CL objective. It verifies the conclusion in Section 5.3 that transformation leads to the overfitting to contrastive loss and may negatively transfer to downstream tasks.

## Q  EFFICIENCY ANALYSIS OF PROPGCL

By excluding transformation weights, PROPGCL demonstrates greater efficiency than the baseline models in both time and memory usage, as evidenced by Tables 25 to 27. For example, PROP-GRACE reduces training time per epoch by 84.29% compared to GRACE with Chebyshev basis on the CS dataset. In terms of memory consumption, PROP-GRACE reduces encoder memory usage by over 99% across various benchmarks relative to the original baseline. Remarkably, PROP-GGD achieves a 20% reduction in training time compared to GRACE on large-scale OGB benchmarks, underscoring the scalability of PROPGCL for large-scale graph learning tasks.

In most real-world graph scenarios, PROPGCL demonstrates significantly higher time efficiency compared to its backbone, even for large-scale graphs. For edge cases involving extremely dense graphs and high feature dimensionality, we propose a lightweight solution—prepending *a random projection layer* before propagation, whose efficacy is validated in Table 2.

Below, we provide a detailed time complexity analysis. For simplicity, consider a basic propagator $\boldsymbol{AX}$, with time complexity $O(|E| \times d)$, where $d$ is the feature dimension and $|E|$ is the edge number. The transformation $\boldsymbol{HW}$ has complexity $O(|V| \times d_{in} \times d_{out})$, where $d_{in} = d_f$ is the input feature dimension, $d_{out}$ is hidden dimension and $|V|$ is node number. PROPGCL utilizes pure propagation as $O(|E| \times d_f)$, while the backbone combines both, *i.e.*, $O(|V| \times d_f \times d_{out} + |E| \times d_{out})$. The time improvement is $\Delta = O(|V|(d_f(d_{out} - s) + s \times d_{out}))$, where $s = |E|/|V|$ is the sparsity factor. The key insights are (1) for *typical graphs* (low $s$, moderate $d_f$), PROPGCL's gains grow with $d_f$, as $d_{out} > s$ often holds for real-world sparse graphs, validated in Table 28. and (2) for *dense and high-dimensional cases*, while gains may narrow, we can lightweightly fix it by prepending a random projection layer before propagation. Table 2 verifies random projections' efficacy, and their no-training nature preserves efficiency. Therefore, PROPGCL's speedup holds across most practical settings.

To verify the feasibility of the random projector, we construct synthetic graphs using the Erdős–Rényi model, consisting of 1000 nodes with a feature dimension of 10,000 and an edge probability of 0.5, resulting in a dense graph with extremely high-dimensional features. To generate meaningful yet non-trivial node features, we combine topological properties (degree, clustering coefficient) and community structure (from spectral clustering). To prevent overly discriminative features, we further

corrupt them with Gaussian noise (std=1.0). Node labels are assigned based on communities, and the data is split into train/validation/test sets following the paper's settings. We evaluate three variants: (1) DGI: vanilla GCL with spectral GNNs as the backbone, (2) PROP-DGI: the method proposed in the paper, removing the transformation entirely, (3) PROP-DGI-RAND: extends PROP-DGI by adding a frozen random projection layer before propagation. The results are shown in Table 29. Although sacrificing a modest performance compared with PROP-DGI, PROP-DGI-RAND still significantly improves over DGI on test accuracy (90.90% vs. 58.05%). Moreover, the random projection further decreases the training time for PROP-DGI from 0.1918s to 0.0227s, demonstrating its efficiency on high-dimensional dense graphs.

Table 25: Training time per epoch in seconds between PROP-GRACE and GRACE. Experiments are all conducted on a single 24GB NVIDIA GeForce RTX 3090, except those denoted with $*$ on 48GB Nvidia A40 for out-of-memory. *Improvement* refers to the percentage increase in speed of the -PROP version compared to the baseline, *i.e.*, $(t_{\text{GRACE}} - t_{\text{PROP-GRACE}})/t_{\text{GRACE}}$.

| Basis | Method | Cora | CiteSeer | PubMed | Photo | Computers | CS | Squirrel | Chameleon | Actor |
|---|---|---|---|---|---|---|---|---|---|---|
| Chebyshev | GRACE | 0.1611 | 0.1939 | 0.2795 | 0.2872 | 0.4639 | 1.5111* | 0.7004 | 0.2295 | 0.2872 |
| | PROP-GRACE | 0.1409 | 0.1478 | 0.2650 | 0.2400 | 0.3626 | 0.2374* | 0.2581 | 0.1450 | 0.2073 |
| | *Improvement* | 12.54% | 23.79% | 5.18% | 16.44% | 21.84% | 84.29% | 63.15% | 36.82% | 27.83% |
| Bernstein | GRACE | 0.1515 | 0.2215 | 0.2513 | 0.4878 | 0.9293 | 6.7666* | 1.8997 | 0.4079 | 0.2619 |
| | PROP-GRACE | 0.1226 | 0.1178 | 0.2334 | 0.3832 | 0.6968 | 0.6038* | 0.5175 | 0.1653 | 0.1789 |
| | *Improvement* | 19.03% | 46.79% | 7.10% | 21.45% | 25.02% | 91.08% | 72.76% | 59.47% | 31.69% |
| Monomial | GRACE | 0.1114 | 0.1023 | 0.1217 | 0.1606 | 0.2340 | 1.2487* | 0.3714 | 0.1524 | 0.1202 |
| | PROP-GRACE | 0.1024 | 0.1224 | 0.1221 | 0.1428 | 0.1928 | 0.1927* | 0.1650 | 0.1151 | 0.1109 |
| | *Improvement* | 8.06% | 16.42% | 0.31% | 11.12% | 17.61% | 84.57% | 55.56% | 24.46% | 7.74% |

Table 26: Memory consumption of encoder in KBs between PROP-GRACE and GRACE. *Improvement* refers to the percentage decrease in the memory consumption of the -PROP version compared to the baseline. *i.e.*, $(m_{\text{GRACE}} - m_{\text{PROP-GRACE}})/m_{\text{GRACE}}$.

| Method | Cora | CiteSeer | PubMed | Photo | Computers | CS | Squirrel | Chameleon | Actor |
|---|---|---|---|---|---|---|---|---|---|
| GRACE | 3894.04 | 8434.04 | 2028.04 | 2518.04 | 2562.04 | 2562.04 | 5206.04 | 5678.04 | 2892.04 |
| PROP-GRACE | 11.24 | 28.97 | 3.95 | 5.86 | 6.04 | 6.04 | 16.36 | 18.21 | 7.32 |
| *Improvement* | 99.71% | 99.66% | 99.81% | 99.77% | 99.76% | 99.76% | 99.69% | 99.68% | 99.75% |

Table 27: Training time per epoch in seconds and memory consumption of encoder in KBs between GGD and PROP-GGD on OGB benchmarks. Experiments are conducted on a single 80GB NVIDIA A100. *Improvement* refers to the percentage increase in speed or decrease in memory consumption.

| Metric | Method | ogbn-arxiv | ogbn-products |
|---|---|---|---|
| | GGD | 1.0270 (2324.00) | 284.3968 (12740.00) |
| Time (Memory) | PROP-GGD | 0.7892 (3.5) | 212.0509 (3.52) |
| | *Improvement* | 23.15% (99.85%) | 25.44% (99.97%) |

We also include direct comparisons with an efficient GCL method SimGCL (Yu et al., 2022), which is explicitly designed to reduce augmentation overhead. SimGCL reduces the cost of heavy graph augmentations by replacing them with uniform embedding noise. In contrast, PROPGCL removes transformation weights entirely, thus eliminating both forward and backward propagation associated with parameterized transformations, which is the dominant computation in most GCL architectures. Table 30 summarizes the time savings in epoch-level training: SimGCL reduces training time by 17.51% on average, while PROP-GRACE achieves a 61.05% reduction, largely due to the elimination of transformation modules. This demonstrates that the core source of efficiency is different: SimGCL optimizes augmentation, while PROPGCL fundamentally simplifies the representation operator itself, leading to a deeper reduction in computation.

Table 28: The relationships of sparse factor $s$ and hidden dimension $d_{out}$ in popular benchmarks

| Dataset | $s$ | $d_{out}$ in best practice | Relationships |
|---------|-----|---------------------------|---------------|
| Cora | 1.95 | 64-512 | $d_{out} > s$ |
| CiteSeer | 1.36 | 64-512 | $d_{out} > s$ |
| PubMed | 2.25 | 64-512 | $d_{out} > s$ |
| Photo | 15.57 | 64-512 | $d_{out} > s$ |
| Computers | 17.88 | 64-512 | $d_{out} > s$ |
| Chameleon | 15.85 | 64-512 | $d_{out} > s$ |
| Squirrel | 41.74 | 64-512 | $d_{out} > s$ |

Table 29: Test accuracy (%) and training time (seconds) on the high-dimensional dense graph.

| Method | Accuracy | Training Time |
|--------|----------|---------------|
| DGI | $58.05 \pm 1.40$ | 0.6293 |
| PROP-DGI | $100.00 \pm 0.00$ | 0.1918 |
| PROP-DGI-RAND | $90.90 \pm 1.30$ | 0.0227 |

## R  COMPARISON WITH SUPERVISED CONTRASTIVE LEARNING

We hypothesize that the failure *partly* of learning effective transformation weights stems from the unsupervised nature of the contrastive task, which leads to inefficient optimization without sufficient guidance. As an initial exploration, we devise a supervised contrastive loss by selecting positive and negative pairs according to ground-truth labels, following the principles of *supervised contrastive learning* (Khosla et al., 2020; Graf et al., 2021). We apply the modified loss to the GCA framework (termed SUP-GCL) and compare the learned transformation weights with those of GCL and SL. As shown in Figure **??**, incorporating supervised signals slightly mitigates the smooth characteristic of GCL weights, but can't fully solve the limitations. We believe the intrinsic reasons behind the ineffective learning of transformation weights remain to be further explored. Fortunately, we find that GCL promisingly captures propagation coefficients and, building on this insight, we propose removing the transformation while retaining only propagation.

## S  TRIALS ON LEARNING EFFECTIVE TRANSFORMATION WEIGHTS IN GCL

According to the analysis in Section 5.1, GCL learns uninformative weights that are excessively smooth. Here we try three ways to solve this problem: 1) enforcing the sparsity of weights with $l_1$ normalization; 2) using whitening methods (Bell & Sejnowski, 1997; Kessy et al., 2018); 3) using normalization methods (Huang et al., 2018; Hua et al., 2021; Guo et al., 2023a).

$l_1$ **regularization.** As a typical technique, the $l_1$ regularization encourages sparsity by driving some weights to zero and retaining the most relevant features. In practice, we add a penalty proportional to the sum of the absolute values of the encoder parameters to the contrastive loss, *i.e.*, $\mathcal{L}_{\text{total}} = \mathcal{L}_{\text{CL}} + \lambda \sum_i |w_i|$, where $\mathcal{L}_{\text{CL}}$ is the contrastive loss, $\lambda$ is the regularization strength, and the $w_i$ is the parameters of the encoder. We conduct experiments on ChebNetII with the $l_1$ regularized GRACE training objective, varying the regularization strength $\lambda$ in $[1 \times 10^{-4}, 1 \times 10^{-5}, 1 \times 10^{-6}]$. As shown in Table 31, the $l_1$ regularization improves performance over the original GRACE on the Squirrel, Chameleon, Texas, Wisconsin, and Cornell datasets, though it still lags behind PROP, except on Wisconsin. However, for Cora, Citeseer, and PubMed, $l_1$ regularization negatively impacts performance.

**Whitening methods.** Whitening methods are used to decorrelate and normalize data. By making dimensions mutually independent, whitening methods implicitly solve the representation collapse problem. Here we consider the typical Zero-phase Component Analysis (ZCA) whitening (Kessy et al., 2018), which transforms the input data such that it has zero mean and identity covariance matrix, while also preserving data structure as much as possible. It is computed by multiplying the data by the inverse square root of its covariance matrix, *i.e.*, $\hat{x} = V\Lambda^{-\frac{1}{2}}V^\top x$, where $V$ is the

Table 30: Time efficiency comparison, with the percentage denoting the decrease of the epoch time consumption (seconds).

| Method | Photo | Computers | CS | Squirrel | Chameleon | *Average* |
|---|---|---|---|---|---|---|
| GRACE | 0.2872 | 0.4639 | 1.5111 | 0.7004 | 0.2295 | 0.6384 |
| SimGCL | 0.2637 (↓8.18%) | 0.3947 (↓14.91%) | 1.0329 (↓31.64%) | 0.6374 (↓8.99%) | 0.1893 (↓17.51%) | 0.5036 (↓21.11%) |
| PROP-GRECE | 0.2400 (↓16.44%) | 0.3626 (↓21.84%) | 0.2374 (↓84.29%) | 0.2581 (↓63.15%) | 0.1450 (↓36.82%) | 0.2486 (↓61.05%) |

Table 31: Test accuracy (%) of node classification benchmarks. We train ChebNetII using the $l_1$ regularized GRACE objective. $\lambda$ denotes the regularization strength. **Bold** indicates the best, while underlined represents the second-best choice.

| | Cora | CiteSeer | PubMed | Squirrel | Chameleon | Texas | Wisconsin | Cornell |
|---|---|---|---|---|---|---|---|---|
| PROP | **85.48 ± 0.76** | **78.87 ± 0.63** | 82.89 ± 0.48 | **58.48 ± 1.03** | **68.82 ± 1.42** | **86.23 ± 3.11** | 89.00 ± 3.25 | **86.23 ± 3.11** |
| $\lambda$=0 (GRACE) | 83.42 ± 0.92 | 74.79 ± 0.57 | **84.92 ± 0.26** | 37.90 ± 0.79 | 55.67 ± 0.96 | 77.87 ± 2.79 | 86.38 ± 3.63 | 75.74 ± 3.61 |
| $\lambda$=1e-4 | 53.71 ± 1.10 | 26.97 ± 0.50 | 81.20 ± 0.21 | 33.07 ± 0.89 | 48.60 ± 1.42 | 80.98 ± 2.30 | 70.00 ± 1.88 | 82.79 ± 2.46 |
| $\lambda$=1e-5 | 78.87 ± 1.17 | 73.29 ± 0.63 | 84.17 ± 0.23 | 37.46 ± 0.89 | 56.37 ± 1.01 | 56.56 ± 1.97 | **91.88 ± 2.25** | 81.80 ± 2.30 |
| $\lambda$=1e-6 | 77.75 ± 0.80 | 73.90 ± 0.74 | 84.16 ± 0.21 | 38.27 ± 1.02 | 56.91 ± 1.09 | 52.79 ± 4.76 | 86.88 ± 2.88 | 74.26 ± 7.38 |

matrix of eigenvectors and $\Lambda$ is the diagonal matrix of eigenvalues of the covariance matrix of $x$. We conduct experiments under the GRACE framework with a ZCA whitening layer added to the encoder ChebNetII. As shown in Table 32, the whitening improves performance over the original GRACE on the PubMed and Chameleon datasets but drastically deteriorates most of the other datasets.

Table 32: Test accuracy (%) of node classification benchmarks. We train ChebNetII using GRACE with the ZCA whitening. **Bold** indicates the best, while underlined represents the second-best choice.

| | Cora | CiteSeer | PubMed | Squirrel | Chameleon | Texas | Wisconsin | Cornell |
|---|---|---|---|---|---|---|---|---|
| PROP | **85.48 ± 0.76** | **78.87 ± 0.63** | 82.89 ± 0.48 | **58.48 ± 1.03** | **68.82 ± 1.42** | **86.23 ± 3.11** | **89.00 ± 3.25** | **86.23 ± 3.11** |
| GRACE | 83.42 ± 0.92 | 74.79 ± 0.57 | 84.92 ± 0.26 | 37.90 ± 0.79 | 55.67 ± 0.96 | 77.87 ± 2.79 | 86.38 ± 3.63 | 75.74 ± 3.61 |
| GRACE+ZCA | 79.29 ± 1.71 | 47.29 ± 0.70 | **85.76 ± 0.29** | 36.72 ± 0.91 | 58.60 ± 1.07 | 43.77 ± 8.36 | 27.38 ± 3.63 | 38.52 ± 6.23 |

**Normalization methods.** For normalization methods, we consider the widely used Batch Normalization (BN) (Ioffe, 2015), and the recently proposed Decorrelate ContraNorm (DCN) (Guo et al., 2023a). Batch normalization scales and shifts the mini-batch of data to have a mean of zero and a standard deviation of one, *i.e.*, $\hat{x} = (x - \mu_B)/\sqrt{\sigma_B^2 + \epsilon}$, where $\mu_B$ and $\sigma_B^2$ are the mean and variance of the mini-batch $B$, and $\epsilon$ is a small constant for numerical stability. DCN scatters representations in the embedding space and leads to a more uniform distribution. The formulation of DCN is $\hat{x} = x - s \times x \times \text{softmax}(x^\top x)$, where $s$ is the scale factor. We conduct experiments under the GRACE framework with a BN or DCN layer added to the encoder ChebNetII. As shown in Table 33, BN and DCN both fail to bring substantial improvement over the original GRACE.

In summary, these techniques offer limited effectiveness for GCL with polynomial GNNs. We think the possible reason is that the learning of transformation weights needs a high-quality supervision signal. Although these methods help prevent representation collapse, they do not carry extra information. Therefore, GCL still fails to learn good transformation weights.

Table 33: Test accuracy (%) of node classification benchmarks. We train ChebNetII using GRACE with BN or DCN normalization. $s$ denotes the scale factor in DCN. **Bold** indicates the best, while underlined represents the second-best choice.

| | Cora | CiteSeer | PubMed | Squirrel | Chameleon | Texas | Wisconsin | Cornell |
|---|---|---|---|---|---|---|---|---|
| PROP | **85.48 ± 0.76** | **78.87 ± 0.63** | 82.89 ± 0.48 | **58.48 ± 1.03** | **68.82 ± 1.42** | **86.23 ± 3.11** | 89.00 ± 3.25 | **86.23 ± 3.11** |
| GRACE | 83.42 ± 0.92 | 74.79 ± 0.57 | 84.92 ± 0.26 | 37.90 ± 0.79 | 55.67 ± 0.96 | 77.87 ± 2.79 | 86.38 ± 3.63 | 75.74 ± 3.61 |
| GRACE + BN | 82.25 ± 1.00 | 72.78 ± 1.00 | **85.10 ± 0.24** | 39.56 ± 0.47 | 54.77 ± 0.74 | 76.07 ± 2.95 | 72.63 ± 4.75 | 75.90 ± 2.79 |
| GRACE + DCN ($s$=0.5) | 79.79 ± 0.99 | 73.86 ± 0.86 | 84.00 ± 0.37 | 38.17 ± 0.95 | 56.19 ± 1.03 | 71.15 ± 2.13 | 83.25 ± 2.50 | 71.64 ± 4.59 |
| GRACE + DCN ($s$=1.0) | 75.19 ± 1.08 | 74.91 ± 0.63 | 83.06 ± 0.22 | 38.28 ± 1.12 | 57.35 ± 0.98 | 74.26 ± 1.64 | **90.50 ± 1.50** | 76.72 ± 3.11 |
| GRACE + DCN ($s$=5.0) | 74.40 ± 1.15 | 74.46 ± 0.63 | 79.41 ± 0.35 | 38.01 ± 0.79 | 58.97 ± 1.33 | 72.95 ± 3.44 | 83.25 ± 2.75 | 73.44 ± 3.44 |

# T  CHARACTERIZATION OF LEARNED TRANSFORMATION WEIGHTS

In Section 5.1, we demonstrated the transformation weights learned by DGI and SL on the Cora dataset. Here, we extend these findings by presenting comprehensive results across various benchmarks and GCL methods including GRACE, GCA, BGRL. As depicted from Figure 10 to Figure 14, the weights learned by SL display diverse, data-dependent distributions, while those learned by CL consistently follow a Gaussian-like distribution that centers at zero. Although we can't exhaust all GCL methods, these representative methods provide further evidence that GCL often struggles to learn effective transformation weights. In Figure 15, we provide results of SUP-CL on more benchmarks, verifying that the participation of supervision signals slightly mitigates the ineffective transformation learning problem.

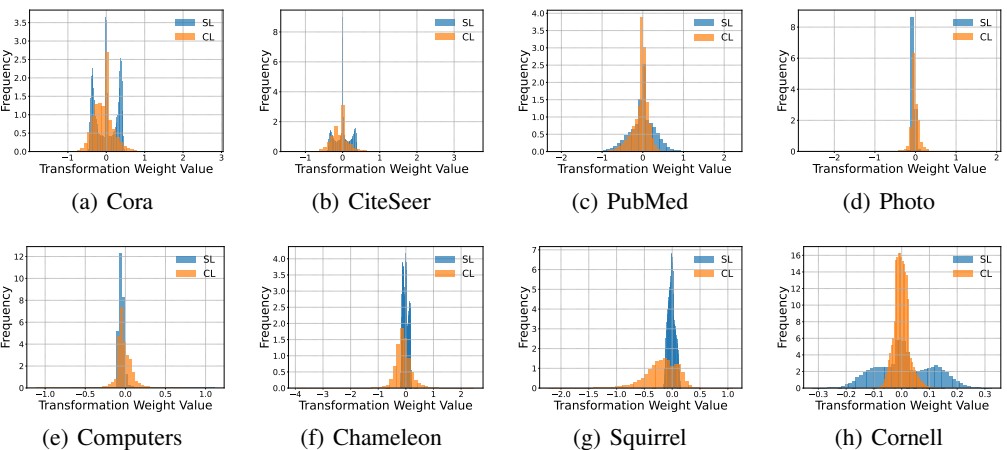

Figure 10: Distribution of the transformation weights learned by GRACE and SL.

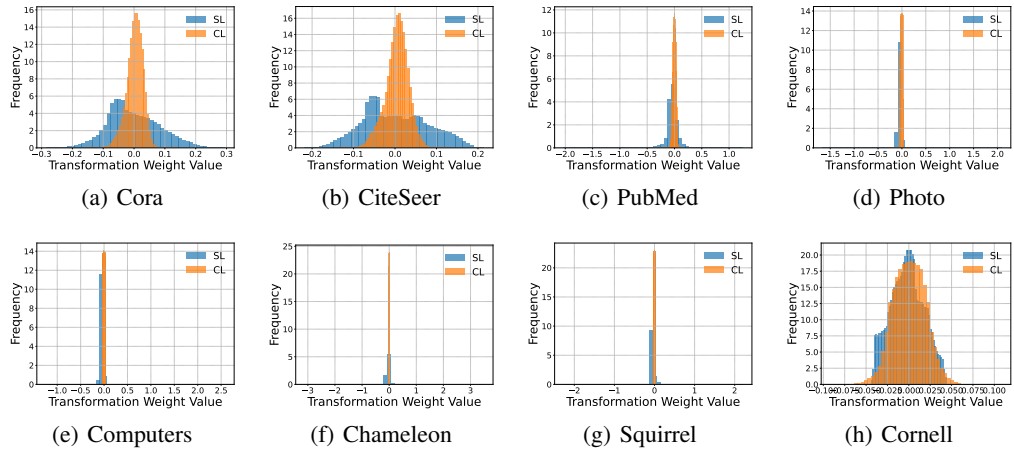

Figure 11: Distribution of the transformation weights learned by DGI and SL.

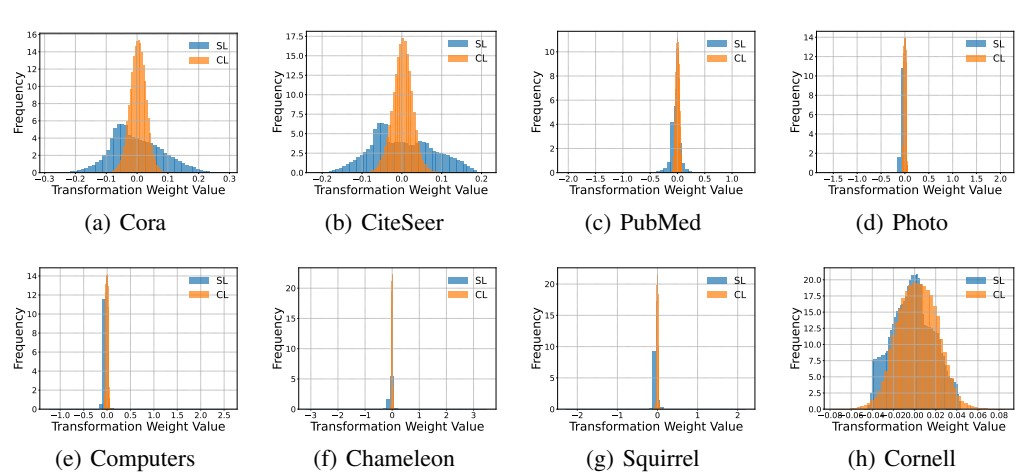

Figure 12: Distribution of the transformation weights learned by GCA and SL.

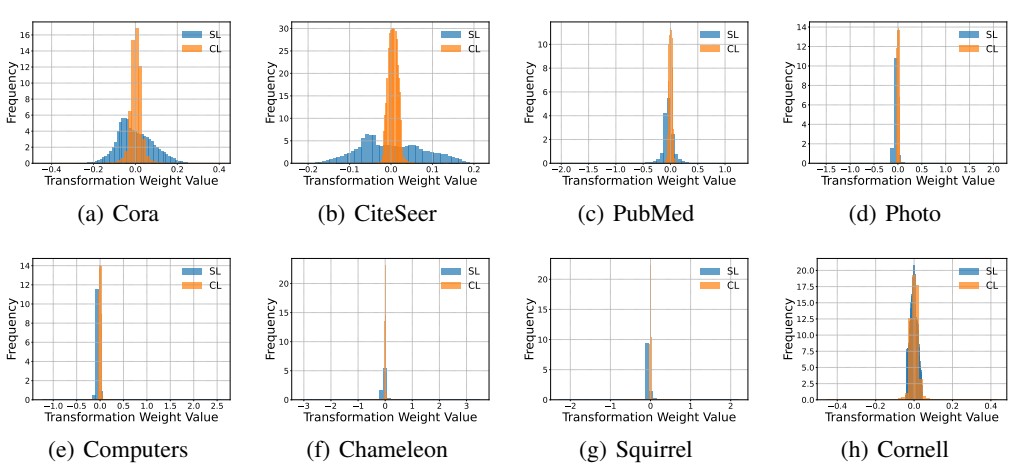

Figure 13: Distribution of the transformation weights learned by BGRL and SL.

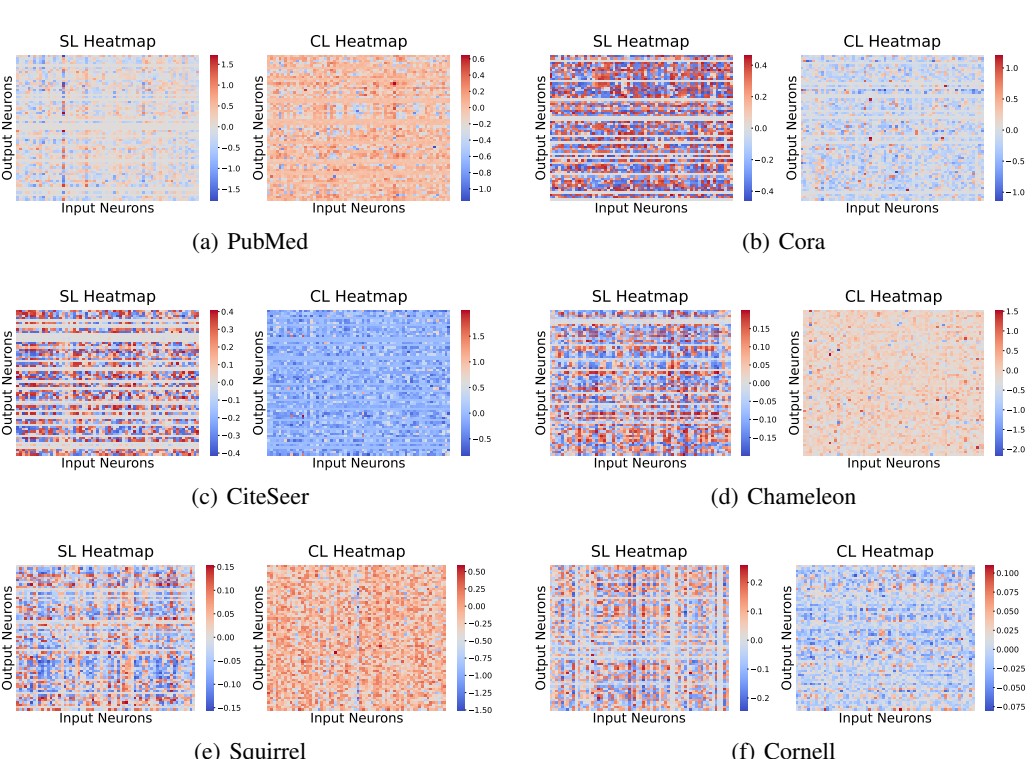

Figure 14: Heatmap of the transformation weights learned by GRACE and SL.

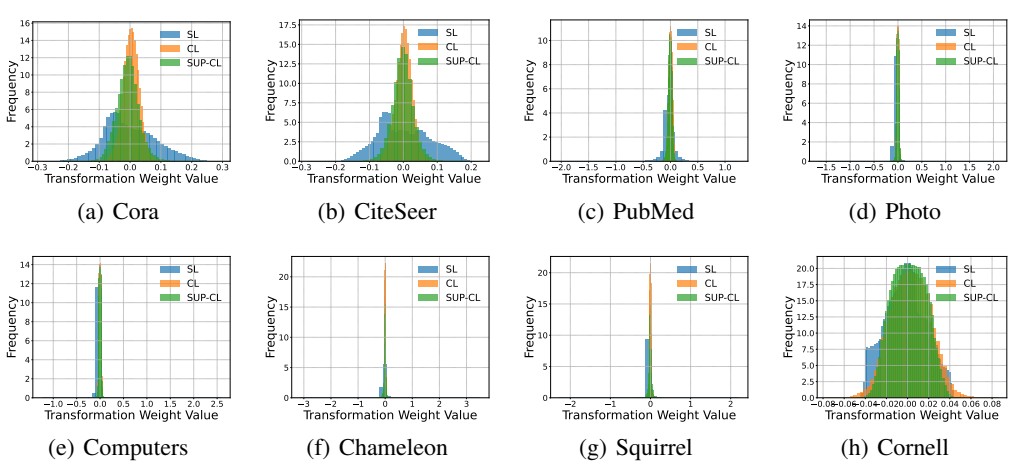

Figure 15: Distribution of the transformation weights learned by GCA, SUP-GCA, and SL.

## U CHARACTERIZATION OF LEARNED PROPAGATION COEFFICIENTS

In section 5.2, we find after fixing the transformation weights with supervised ones, the model trained via GCL performs as well as in a supervised manner. To verify that given well-trained transformation weights, GCL can learn effective propagation coefficients. We compare the propagation coefficients learned by SL, GCL, and the fix-transformation GCL. As shown in Figure 16, compared with CL, the propagation coefficients learned by fix-transformation GCL are closer to those in SL, demonstrating that GCL can learn effective propagation coefficients fitting the given transformation weights.

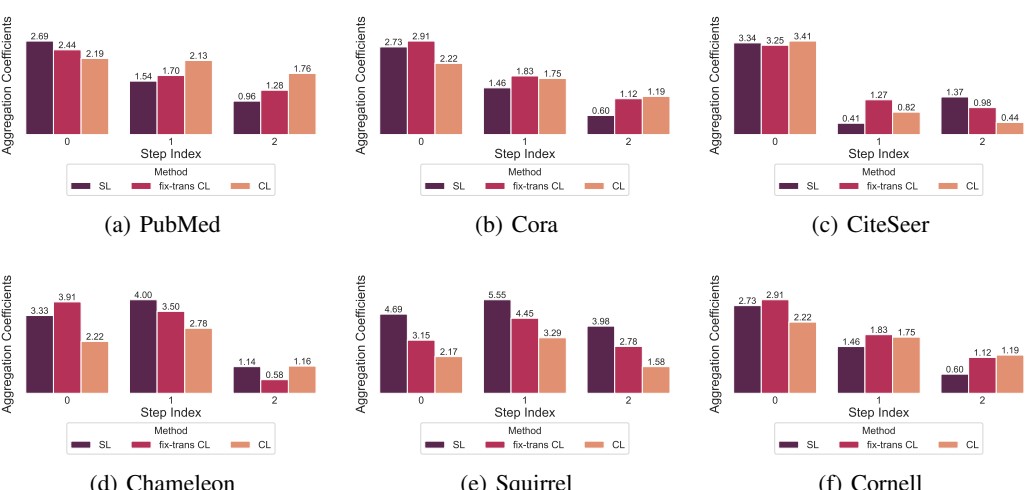

(a) PubMed       (b) Cora       (c) CiteSeer

(d) Chameleon       (e) Squirrel       (f) Cornell

Figure 16: Propagation coefficients of supervised learning (SL), contrastive learning (CL), and fix-transformation contrastive learning (fix-trans CL) introduced in Section 5.2. We show the first three propagation coefficients for the space limit.

## V EXPERIMENTAL DETAILS OF PROP AND PROPGCL

### V.1 BENCHMARKS

**Node classification benchmarks**. 1) *Citation Networks* (Sen et al., 2008; Namata et al., 2012). Cora, CiteSeer, and PubMed are three popular citation graph datasets. In these graphs, nodes represent papers and edges correspond to the citation relationship between two papers. Nodes are classified according to academic topics. 2) *Amazon Co-purchase Networks* (Shchur et al., 2018). Photo and Computers are collected by crawling Amazon websites. Goods are represented as nodes and the co-purchase relationships are denoted as edges. Node features are the bag-of-words representation of product reviews. Each node is labeled with the category of goods. 3) *Wikipedia Networks* (Rozemberczki et al., 2021). Squirrel and Chameleon are collected from the English Wikipedia, representing page-page networks on specific topics. Nodes represent articles and edges are mutual links between them. 4) *WebKB Networks* (Pei et al., 2020). In Texas, Wisconsin, and Cornell datasets, nodes represent web pages and edges represent hyperlinks between them. Node features are the bag-of-words representation of web pages. 5) *Actor Networks* Pei et al. (2020). Each node corresponds to an actor, and the edge between two nodes denotes co-occurrence on the same Wikipedia page. Node features correspond to some keywords on the Wikipedia pages. Statistics of datasets are shown in Table 34.

**Graph Classification benchmarks**. 1) *Molecules*. MUTAG (Debnath et al., 1991) is a dataset of nitroaromatic compounds and the goal is to predict their mutagenicity on Salmonella Typhimurium. NCI1 (Wale et al., 2008) is a dataset of chemical molecules that are annotated based on their activity against non-small cell lung cancer and ovarian cancer cell lines. 2) *Bioinformatics*. PROTEINS (Borgwardt et al., 2005) is a dataset of proteins that are classified as enzymes or non-enzymes. Nodes represent the amino acids and two nodes are connected by an edge if they are less than 6 Angstroms apart. DD (Dobson & Doig, 2003) consists of protein structures with nodes corresponding

to amino acids and edges indicating that two amino acids are within a certain number of angstroms. 3) *Social Networks*. IMDB-BINARY and IMDB-MULTI (Yanardag & Vishwanathan, 2015) are movie collaboration datasets consisting of a network of 1,000 actors/actresses who played roles in movies in IMDB. In each graph, nodes represent actors/actresses; corresponding nodes are connected if they appear in the same movie. COLLAB (Yanardag & Vishwanathan, 2015) is derived from three public collaboration datasets representing scientific collaborations between authors. For all benchmarks, we use collections from TUDataset (Morris et al., 2020). Statistics of datasets are shown in Table 35.

Table 34: Statistics of node classification benchmarks. $\mathcal{H}(G)$ denotes the edge homophily ratio introduced in Zhu et al. (2020a).

| Homo / Hetero | Category | Dataset | # Nodes | # Edges | # Features | # Classes | $\mathcal{H}(G)$ |
|---|---|---|---|---|---|---|---|
| Homophily | Citation | Cora | 2,708 | 5,278 | 1,433 | 7 | 0.81 |
| | | CiteSeer | 3,327 | 4,552 | 3,703 | 6 | 0.74 |
| | | PubMed | 19,717 | 44,338 | 500 | 3 | 0.80 |
| | Co-purchase | Photo | 7,650 | 119,081 | 745 | 8 | 0.83 |
| | | Computers | 13,752 | 245,861 | 767 | 10 | 0.78 |
| Heterophily | Wikipedia | Chameleon | 2,277 | 36,101 | 2,325 | 6 | 0.23 |
| | | Squirrel | 5,201 | 217,073 | 2,089 | 4 | 0.22 |
| | WebKB | Texas | 183 | 279 | 1703 | 5 | 0.11 |
| | | Wisconsin | 251 | 466 | 1703 | 5 | 0.21 |
| | | Cornell | 183 | 277 | 1703 | 5 | 0.30 |
| | Film-actor | Actor | 7,600 | 30,019 | 932 | 5 | 0.22 |

Table 35: Statistics of graph classification benchmarks. We report average numbers of nodes, edges, and features across graphs in graph classification datasets.

| Category | Dataset | #Graphs | # Nodes | # Edges | # Features | # Classes |
|---|---|---|---|---|---|---|
| Moleculars | MUTAG | 188 | 17.9 | 39.6 | 7 | 2 |
| | NCI1 | 4110 | 29.87 | 32.30 | 37 | 2 |
| Proteins | PROTEINS | 1113 | 39.1 | 145.6 | 0 | 2 |
| | DD | 1178 | 284.32 | 715.66 | 89 | 2 |
| Social Networks | IMDB-BINARY | 1000 | 19.8 | 193.1 | 0 | 2 |
| | IMDB-MULTI | 1500 | 13.0 | 131.9 | 0 | 3 |
| | COLLAB | 5000 | 74.49 | 2457.78 | 0 | 3 |

## V.2 BASELINES

We categorize baselines for the **node classification task** into 1) traditional graph embedding algorithms DeepWalk (Perozzi et al., 2014) and Node2Vec (Grover & Leskovec, 2016); 2) graph autoencoders GAE (Kipf & Welling, 2016), VGAE (Kipf & Welling, 2016); 3) graph contrastive methods GRACE (Zhu et al., 2020b), DGI (Velickovic et al., 2019), GCA (Zhu et al., 2021c), MV-GRL (Hassani & Khasahmadi, 2020), ProGCL (Xia et al., 2022); 4) graph non-contrastive methods CCA-SSG (Zhang et al., 2021) and BGRL (Thakoor et al., 2022), 5) heterophily baselines compared in Section 6.3, PolyGCL (Chen et al., 2024), HGRL (Chen et al., 2022), GraphACL (Xiao et al., 2024), SP-GCL (Wang et al., 2023), DSSL (Xiao et al., 2022).The design details are as follows.

1) *Traditional graph embeddings*.

- **DeepWalk** (Perozzi et al., 2014). DeepWalk leverages truncated random walks to capture local network structures. The algorithm treats the random walks as sequences of nodes, akin to sentences in language models. It learns latent representations by applying skip-gram to maximize the co-occurrence probabilities of nodes appearing in these random walks.

- **Node2Vec** (Grover & Leskovec, 2016). Node2Vec is built on DeepWalk by introducing a flexible biased random walk strategy to explore network neighborhoods. The key innovation is balancing breadth-first sampling (BFS) and depth-first sampling (DFS). This allows

Node2Vec to capture both homophily and structural equivalence, making the learned node embeddings more expressive.

2) *Graph autoencoders.*

- **GAE** (Kipf & Welling, 2016). GAE involves an encoder-decoder architecture, where the encoder is a GCN that transforms node features into latent embeddings by aggregating information from neighboring nodes. The embeddings are then used by the decoder, which typically applies a simple inner product operation to reconstruct the graph structure, such as predicting edges between nodes.

- **VGAE** (Kipf & Welling, 2016). VGAE extends GAE by introducing a probabilistic framework using a variational autoencoder (VAE) setup. It models latent variables with Gaussian distributions, enabling the generation of node embeddings that capture uncertainty. This design improves the model's ability to capture complex structures in graphs, especially in tasks like link prediction.

3) *Graph contrastive methods.*

The mode of GCL has three mainstreams: local-to-local, global-to-global, and global-to-local (Zhu et al., 2021b). A classic example of local-to-local is GRACE (Zhu et al., 2020b), which generates two graph views by augmentations and the same nodes in augmented views are positive while all the other node pairs are negative. Global-to-global mode is often used with multiple graphs in the graph classification task, with GraphCL (You et al., 2020) as an early but influential trial. For the global-to-local perspective, positive pairs are taken as the global representation and nodes of augmented views, and negative pairs are the global representation and nodes of corrupted views. DGI (Velickovic et al., 2019) is a typical example.

- **GRACE** (Zhu et al., 2020b). GRACE generates two graph views by corruption and learns node representations by maximizing the agreement of node representations in these two views. To provide diverse node contexts for the contrastive objective, GRACE proposes a hybrid scheme for generating graph views on both structure and attribute levels.

- **GCA** (Zhu et al., 2021c). GCA proposes adaptive augmentation that incorporates various priors for topological and semantic aspects of the graph. On the topology level, GCA designs augmentation schemes based on node centrality measures, while on the node attribute level, GCA corrupts node features by adding more noise to unimportant node features.

- **DGI** (Velickovic et al., 2019). DGI relies on maximizing mutual information between patch representations and corresponding high-level summaries of graphs—both derived using established graph convolutional network architectures. The learned patch representations summarize subgraphs centered around nodes of interest, and can thus be reused for downstream node-wise learning tasks.

- **MVGRL** (Hassani & Khasahmadi, 2020). MVGRL introduces a self-supervised approach for learning node and graph-level representations by contrasting structural views of graphs. MVGRL shows that contrasting multi-scale encodings does not improve performance, and the best performance is achieved by contrasting encodings from first-order neighbors and graph diffusion.

- **ProGCL** (Xia et al., 2022). ProGCL observes limited benefits when adopting existing hard negative mining techniques of other domains in graph contrastive learning. ProGCL proposes an effective method to estimate the probability of a negative being true and devises two schemes to boost the performance of GCL.

4) *Non-contrastive methods.*

- **CCA-SSG** (Zhang et al., 2021). CCA-SSG optimizes a novel feature-level objective that aligns features across different graph augmentations. It uses decorrelation to prevent degenerate solutions, allowing the model to learn invariant node representations. The model avoids a mutual information estimator or negative samples, which simplifies training and reduces computational complexity.

- **BGRL** (Thakoor et al., 2022). BGRL avoids the use of negative samples by predicting different augmentations of the input graph. BGRL relies on a bootstrapping mechanism,

where one branch predicts the output of another branch that is not updated by gradient descent. This method eliminates the complexity of contrastive learning and negative sampling, making it more scalable.

5) *Heterophily baselines*.

- **PolyGCL** (Chen et al., 2024). PolyGCL integrates spectral polynomial filters into graph contrastive learning, enabling it to handle both homophilic and heterophilic graphs. The method generates different spectral views using polynomials and incorporates high-pass information into the contrastive objective.

- **HGRL** (Chen et al., 2022). HGRL introduces self-supervised learning for heterophilic graphs by capturing distant neighbors and preserving original node features. It achieves this through carefully designed pretext tasks optimized via high-order mutual information, avoiding reliance on labels.

- **GraphACL** (Xiao et al., 2024). GraphACL focuses on an asymmetric view of neighboring nodes. The algorithm captures both one-hop local neighborhood information and two-hop monophily similarity, crucial for modeling heterophilic structures.

- **SP-GCL** (Wang et al., 2023). SP-GCL introduces a single-pass graph contrastive learning method without augmentations. It theoretically guarantees performance across both homophilic and heterophilic graphs by studying the concentration property of features obtained through neighborhood propagation.

- **DSSL** (Xiao et al., 2022). DSSL decouples neighborhood semantics in self-supervised learning for node representation. It introduces a latent variable model that decouples node and link generation, making it flexible to different graph structures. The method utilizes variational inference for scalable optimization, improving downstream performance without relying on homophily assumptions.

We categorize the baselines in the **graph classification task** into 1) graph kernel methods including GL (Shervashidze et al., 2009), WL (Shervashidze et al., 2011), and DGK (Yanardag & Vishwanathan, 2015), 2) traditional graph embedding methods including node2vec (Grover & Leskovec, 2016), sub2vec (Adhikari et al., 2018), and graph2vec (Narayanan et al., 2017), 3) contrastive learning methods including InfoGraph (Sun et al., 2020), GraphCL (You et al., 2020), MVGRL (Hassani & Khasahmadi, 2020), JOAOv2 (You et al., 2021), ADGCL (Suresh et al., 2021) as introduced in recent works. The design details are as follows.

1) *Graph kernel methods*.

- **Graphlet Kernel** (GL) (Shervashidze et al., 2009). GL works by counting the number of small subgraphs (known as graphlets) of a fixed size that appear in each graph. The comparison of these counts across graphs allows the kernel to capture the local topological structures of the graphs, making it useful for tasks such as graph classification.

- **Weisfeiler-Lehman Sub-tree Kernel** (WL) (Shervashidze et al., 2011). WL extends the concept of graph kernels by applying the Weisfeiler-Lehman test of isomorphism on graphs. It involves iteratively relabeling the nodes of the graphs based on the labels of their neighbors and then using these relabelings to define a kernel, typically counting matching sub-trees.

- **Deep Graph Kernel** (DGK) (Yanardag & Vishwanathan, 2015). DGK combines deep learning techniques with graph kernels. It first learns a low-dimensional representation of the graphs through unsupervised learning (often using a form of graph embedding or autoencoders), then applies traditional kernel methods to these representations.

2) *Traditional graph embeddings*.

- **Node2Vec** (Grover & Leskovec, 2016). Node2Vec is built on DeepWalk by introducing a flexible biased random walk strategy to explore network neighborhoods. The key innovation is balancing BFS and DFS. This allows Node2Vec to capture both homophily and structural equivalence, making the learned node embeddings more expressive.

- **Sub2Vec** (Adhikari et al., 2018). Inspired by the word2vec model, sub2vec learns vector representations for subgraphs in a graph. It treats each subgraph as a "word" and the

entire graph as a "document" to learn embeddings that capture the structural and contextual properties of subgraphs.

- **Graph2Vec** (Narayanan et al., 2017). Similar to sub2vec, graph2vec is designed to learn embeddings for entire graphs. By treating each graph as a "document" and graph sub-structures as "words," graph2vec employs a document embedding approach to learn a fixed-size vector representation for each graph.

3) *Graph contrastive methods.*

- **GraphCL** (You et al., 2020). GraphCL designs four types of graph augmentations to incorporate various priors and learns graph-level representations by maximizing the global representations of two views for a graph.

- **InfoGraph** (Sun et al., 2020). InfoGraph maximizes the mutual information between the graph-level representation and the representations of substructures of different scales (*e.g.*, nodes, edges, triangles). By doing so, the graph-level representations encode aspects of the data that are shared across different scales of substructures.

- **ADGCL** (Suresh et al., 2021). ADGCL proposes a novel principle, adversarial GCL, which enables GNNs to avoid capturing redundant information during training by optimizing adversarial graph augmentation strategies used in GCL.

- **JOAO** (You et al., 2021). JOAO proposes a unified bi-level optimization framework to automatically, adaptively, and dynamically select data augmentations when performing GraphCL on specific graph data.

### V.3 SETTINGS

For the node classification task, following Zhu et al. (2020b); Velickovic et al. (2019); Hassani & Khasahmadi (2020), we use linear evaluation protocol, where the model is trained in an unsupervised manner and feeds the learned representation into a linear logistic regression classifier. In the evaluation procedure, we randomly split each dataset with a training ratio of 0.8 and a test ratio of 0.1, and hyperparameters are fixed the same way for all the experiments. Each experiment is repeated ten times with mean and standard derivation of accuracy score.

For the graph classification task, we use Adam SGD optimizer with the learning rate selected in $\{10^{-3}, 10^{-4}, 10^{-5}\}$ and the number of epochs in $\{20, 100\}$. For PROP, we only search the propagation step $K$ in the range of $[0, 1, 2, 3, 5, 10]$. Following Sun et al. (2020); You et al. (2020), we feed the generated graph embeddings into a linear Support Vector Machine (SVM) classifier, and the parameters of the downstream classifier are independently tuned by cross-validation. The C parameter is tuned in $\{10^{-3}, 10^{-2}, \cdots, 10^2, 10^3\}$. We report the mean 10-fold cross-validation accuracy with standard deviation. All experiments are conducted on a single 24GB NVIDIA GeForce RTX 3090.

### V.4 HYPERPARAMETER

For all methods, we train the linear classifier for 2000 epochs with a learning rate of 0.01 and no weight decay. For hyperparameters of the model architecture and the unsupervised training procedure, we maintain consistency in the hyperparameter search space across methods as much as possible. Specifically, for GRACE, we search the temperature $\tau$ in [0.1, 0.5, 1.0], the projector hidden dimension in [128, 256, 512], the learning rate in [0.01, 0.001], fix the patience as 50, and all augmentation rates as 0.2. For DGI, we search the learning rate in [0.01, 0.001], the early-stopping patience in [50, 100], and the hidden dimension in [128, 256, 512]. For CCA-SSG, we search the training epochs in [20, 50, 100], $\lambda$ in [1e-3, 5e-4], the hidden dimension in [128, 256, 512], and fix all augmentation ratios as 0.2. For GCA, we search the temperature $\tau$ in [0.1, 0.5, 1.0], the projector hidden dimension in [128, 256, 512], the drop scheme in [pr, degree, evc], and fix the early-stopping patience as 50, the learning rate as 0.01, and all augmentation ratios as 0.2. For BGRL, we search the predictor hidden dimension in [128, 256, 512], the learning rate in [1e-4, 1e-5], the weight decay in [0, 1e-5], fix the learning rate warmup epochs as 1000, the momentum moving as 0.99. For DeepWalk, we search the vector dimension in [128, 256, 512], the context window size in [5, 10], the walk number in [10, 20], and the walk length in [40, 80]. For Node2Vec, we search the vector dimension in [128, 256, 512], the walk number in [10, 20], the probability $p$ in [0.5, 1.0], $q$ in [0.5, 1.0], and

fix the context window size as 10, and the walk length as 80. For MVGRL, we search the learning rate in [0.01, 0.001], the early stopping patience in [50, 100], and the hidden dimension in [128, 256, 512]. For GAE and VGAE, we search the learning rate in [0.01, 0.001], the early stopping patience in [50, 100], and the hidden dimension in [128, 256, 512]. For the heterophily baselines in 6.3, we use the optimal hyperparameter combinations provided in the original papers.

