# OpenReview forum: "PROPGCL: Unleashing the Power of Propagation in Graph Contrastive Learning"
_ICLR.cc/2026/Conference — Submitted to ICLR 2026_

### Official Review · Reviewer_8KxH · 2025-10-28

**Soundness:** 3
**Presentation:** 3
**Contribution:** 2
**Rating:** 6
**Confidence:** 3

**Summary:**

This paper investigates simple propagation operators in graph contrastive learning. The authors introduce PROP, a training-free propagation method that achieves competitive performance across node classification benchmarks. Through decoupling analysis, they reveal that existing GCL methods struggle to learn meaningful transformation weights while showing potential in learning propagation coefficients. Based on these insights, they propose PROPGCL, which eliminates transformation layers and only learns graph-adaptive propagation coefficients. The method achieves strong results on heterophilic datasets with significant computational advantages.

**Strengths:**

1.The core finding that simple propagation matches complex GCL methods is surprising and well-demonstrated. PROP consistently performs well across benchmarks, particularly on heterophilic graphs where many GCL methods struggle.

2. Thm. 4.1 now formally connects propagation to Dirichlet energy minimization with proper mathematical rigor. Thm. 6.1 provides theoretical guarantees for PROPGCL's advantage when contrastive and downstream objectives are misaligned. These additions address major theoretical gaps from prior versions.

3. The ablation studies in sec. 5.2 are particularly convincing. Tab. 3 shows that (1) GCL struggles to learn effective transformation weights even with optimal propagation coefficients, and (2) GCL can learn informative propagation coefficients when paired with well-trained transformation weights. This provides clear evidence for the main claims.

3. Experiments compare against diverse architectures including spectral-based models etc. The addition of large-scale experiments on ogbn-products addresses scalability. The efficiency analysis in sec. 6.4 shows 99% memory reduction and substantial training time improvements.

**Weaknesses:**

1. The title and narrative discuss "graph contrastive learning" broadly, but the method focuses on node classification. App. B shows a 2.82% performance gap on graph classification tasks. This suggests fundamental limitations when global graph representations are needed. The paper should either narrow the title and claims to node-level tasks, or provide deeper analysis of why propagation works for node classification but not graph classification.

2. Thm. 6.1 assumes the downstream-relevant component corresponds to low-frequency signals with smoothness. However, heterophilic graphs violate this assumption since connected nodes have different labels. Yet PROPGCL shows its strongest improvements on exactly these datasets. The paper should discuss when these assumptions hold and provide empirical validation of the decomposition in def. 6.1.

3. The paper frames PROP as performing implicit alignment in contrastive learning (thm. 4.2). Yet PROP requires no negative samples and no learned parameters. This blurs what constitutes contrastive learning versus effective preprocessing. Is the finding that simple propagation works well a contribution to GCL methodology, or evidence that the GCL paradigm is unnecessary? The paper should address this more explicitly.

**Questions:**

1.Fig. 2 shows training loss rapidly approaching zero for GCL with transformation. Could you also show validation or downstream performance curves? This would directly visualize the negative transfer effect. Does early stopping based on downstream validation help?


2. Tab. 2 shows random weights achieve 73.4% vs 72.8% for GCL-learned weights. Could you provide statistical significance tests? Have you tried other random initialization schemes beyond Gaussian?

3. The paper uses Chebyshev basis functions. How sensitive is PROPGCL to this choice? Do the conclusions hold across different filter families?

---

> ### Author Response · Authors · 2025-11-27
>
> We thank Reviewer 8KxH for the constructive and insightful comments. We found the feedback extremely helpful in improving the clarity and technical depth of our work. We summarize and address your concerns below.
>
> ---
>
> **Q1.** The title and narrative discuss "graph contrastive learning" broadly, but the method focuses on node classification. App. B shows a 2.82% performance gap on graph classification tasks. This suggests fundamental limitations when global graph representations are needed. The paper should either narrow the title and claims to node-level tasks, or provide deeper analysis of why propagation works for node classification but not graph classification.
>
> **A1.** Thanks for the constructive suggestion. **We have narrowed our title and claims to node-level tasks in the revised paper.**
>
> ---
>
> **Q2.** Thm. 6.1 assumes the downstream-relevant component corresponds to low-frequency signals with smoothness. However, heterophilic graphs violate this assumption since connected nodes have different labels. Yet PROPGCL shows its strongest improvements on exactly these datasets. The paper should discuss when these assumptions hold and provide empirical validation of the decomposition in def. 6.1.
>
> **A2.** We appreciate the opportunity to clarify this. **Our original phrasing conflated spectral high frequency with high frequency noise in Chebyshev polynomial theory.** To avoid this confusion, we revisited the assumption:
> - **Downstream-relevant component $f$**: corresponds to **informative continuous signals**. These may be low-frequency (homophily) or high-frequency (heterophily), as long as they are continuous over the graph domain.
> - **Downstream-irrelevant component $g$**: represents **discontinuous noise (random patterns, sampling artifacts)**.
>
> According to the Chebyshev approximation theory, continuous functions admit exponentially fast polynomial approximation, while discontinuous mappings incur large approximation error. This explains why PROPGCL learns useful signals (continuous structure), even on heterophilic graphs.
>
> **We have updated the definitions accordingly in the manuscript:**
>
> ***Definition 6.1 (Optimal Propagation Decomposition).*** Let $T^* = \arg\min_{T} \mathcal{L}_{\text{CL}}(T \cdot X)$ be the optimal  operator for the contrastive learning loss. We assume $T^{\*}$ is a function of bounded variation, i.e., $T^* \in BV( \Omega; \mathbb{R}^d)$. By the Jordan decomposition theorem for bounded variation mappings, $T^* = f + g$,  where $f$ is the continuous component and $g$ is the discontinuous component.
>
>
> ***Assumption 6.2 (Approximation Properties).*** We assume the continuous $f$ corresponds to the downstream-relevant component, e.g, informative signals on the graph, while the discontinuous $g$ represents nuisance or disconnected signals. Based on Chebyshev polynomial theory, continuous functions can be well approximated by polynomials: $\inf_{\theta} \vert f - \sum_k \theta_k A^k\vert_F \leq \epsilon_f = C_f K^{-s}$, where $s > 0$ and $C_f$ is a constant. Discontinuous noises yield large polynomial approximation errors: $\inf_{\theta} \vert g - \sum_k \theta_k A^k\vert_F \geq \epsilon_g > 0$, where $\epsilon_g \gg \epsilon_f > 0$.
>
> ---
>
> **Q3.** The paper frames PROP as performing implicit alignment in contrastive learning (thm. 4.2). Yet PROP requires no negative samples and no learned parameters. This blurs what constitutes contrastive learning versus effective preprocessing. Is the finding that simple propagation works well a contribution to GCL methodology, or evidence that the GCL paradigm is unnecessary? The paper should address this more explicitly.
>
> **A3.** Our main contribution is methodological: **establishing propagation as a strong baseline within the GCL pipeline.** We show that contrastive objectives are powerful even without parametric transformations. Therefore, **PROP is not evidence that the GCL paradigm is unnecessary, but rather evidence that many current parametric designs may be overcomplicated.** We hope future work will consider PROP as a basic baseline to test whether the transformation components truly add value.
>
> Thank you for this insightful point. We have explicitly discussed this in the revised paper.
>
> ---

---

> ### Author Response · Authors · 2025-11-27
>
> **Q4.** Fig. 2 shows training loss rapidly approaching zero for GCL with transformation. Could you also show validation or downstream performance curves? This would directly visualize the negative transfer effect. Does early stopping based on downstream validation help?
>
> **A4.** **We have revised Figure 2 to include downstream test accuracy curves** (also see https://ibb.co/1J2ww9Qq). With the learnable transformation, the loss steadily decreases to ≈0.05. However, the test accuracy peaks at ~200 epochs (54.70%) but **drops to 41.79% after 500 epochs (−12.91%)**. Without transformation, the test accuracy peaks at 72.42% at ~280 epochs and **remains stable (71.33% at 500 epochs, −1.09%)**. This directly displays the negative effect of the overfitting brought by including transformation weights.
>
> For the early stopping technique, all GCL baselines in Table 1 applied early stopping; however, it brought no substantial help for surpassing PROP.
>
> ---
>
> **Q5.** Tab. 2 shows random weights achieve 73.4% vs 72.8% for GCL-learned weights. Could you provide statistical significance tests? Have you tried other random initialization schemes beyond Gaussian?
>
> **A5.** To address this concern, we conduct **a paired t-test** for the overall comparison between GCL-learned weights and random weights, where **the overall means across all datasets show no statistically significant difference (t = -0.613, p = 0.5594)**. It confirms that the overall performance between GCL-learned weights and random weights is statistically indistinguishable.
>
> **For additional random initialization schemes, we explored it in Appendix I.** We compared GCL weights with four random initializations: Gaussian, Uniform, Kaiming, and Xavier. Appendix Table 15 shows that all randomized weights perform comparably to (even slightly better than) GCL-trained weights, confirming the GCL weights deficiency.
>
> ---
>
> **Q6.** The paper uses Chebyshev basis functions. How sensitive is PROPGCL to this choice? Do the conclusions hold across different filter families?
>
> **A6.** **We discussed the ablation of basis functions in Appendix L.** We additionally considered the Bernstein basis and monomial basis. As shown in ***Appendix Table 18*** and ***Appendix Table 19***, **the performance of PROPGCL is relatively robust in the choice of basis functions**. For homophily benchmarks, PROP-GRACE with Chebyshev basis and the PROP-DGI with monomial basis achieve the best, surpassing the second slightly by 0.05% on average. For heterophily benchmarks, the best PROP-DGI with the Chebyshev basis achieves 73.71% on average, and the Bernstein basis ranks second.
>
> ---
>
> Thanks for your comments and hope our answers could address your concerns. Please let us know if there is more to clarify. We are happy to address them during the discussion stage.

---

### Official Review · Reviewer_rEYN · 2025-10-30

**Soundness:** 2
**Presentation:** 3
**Contribution:** 1
**Rating:** 2
**Confidence:** 4

**Summary:**

This paper claimed that matrix transform in traditional GCL degrades model performance. So, the authors removed it, purely relied on raw feature propagation. Moreover, they learned the coefficients to combine different order of graph Laplacian, enhancing propagation.

**Strengths:**

1. This paper gave a comprehensive analysis of each component in GCL.
2. The authors gave enough experiments and theorems to support their claims.

**Weaknesses:**

1. The experimental results on PROP is very strange. In table 1, PROP, a pure $A^kX$, can perfrom extremely well on heterophilic graphs. The authors explained this that $A^kX$ can smooth all k-hop neighbors. That's ture, but we know Dirichlet energy $H^TLH=\sum A_{ij}||h_i-h_j||^2$. For $A^k$, it becomes to $\sum A^k_{ij}||h_i-h_j||^2$. Therefore, minimizing Dirichelet norm just equals to make all nodes within k-hops have similar embeddings. This is against the fact that we have to differentiate neighbor nodes embeddings to work on heterophilic graphs. From graph spectrum, $A^kX$ is a definitely a low-pass filter, while only high-pass filter can perform well on heterophilic graphs.

2. In table 1, GSSL methods performed severely worse than supervised methods. This is also against consensus that GSSL methods are better than vanilla supervised GNN, like GCN.

**Questions:**

In figure 2 GCLw/o transformation, the loss nearly did not decrease, or the the model was not trained. So, if GCL is without transformation, which paramters are learnt in it? Since the propation part is just H_PROP, also without paramters.

---

> ### Author Response · Authors · 2025-11-27
>
> We sincerely appreciate Reviewer rEYN for the thoughtful suggestions. We summarize and respond to your concerns as follows.
>
> ---
>
> **Q1.** The experimental results on PROP are very strange. In Table 1, PROP, a pure $A^kX$, can perform extremely well on heterophilic graphs. The authors explained that $A^kX$ can smooth all k-hop neighbors. That's true, but we know Dirichlet energy $H^\top LH = \sum A_{ij} \Vert h_i - h_j \Vert^2$. For $A^k$, it becomes to $\sum A^k_{ij} \Vert h_i-h_j \Vert^2.$ Therefore, minimizing the Dirichlet norm just equals to make all nodes within k-hops have similar embeddings. This is against the fact that we have to differentiate neighbor node embeddings to work on heterophilic graphs. From the graph spectrum, $A^k$ is definitely a low-pass filter, while only a high-pass filter can perform well on heterophilic graphs.
>
> **A2.** We appreciate the reviewer’s insightful point and agree that **infinite-order** smoothing promotes global homogeneity. The key nuance is that, in our setting, **propagation depth $k$ is finite**, and **finite powers of the normalized adjacency matrix do not behave as *pure* low-pass filters**.
>
> Let $\hat{A} = D^{-1/2}AD^{-1/2}$. The propagation operator $\hat{A}^k$ raises each eigenvalue to the $k$-th power, changing both magnitude and sign. For moderate $k$, this produces mixed filtering behavior:
> - Low-frequency eigenvalues ($\lambda \approx 1$) remain close to 1 → smoothing behavior.
> - High-frequency eigenvalues ( $\lambda < 0$) flip signs and increase in magnitude → amplification of oscillatory / heterophily signals.
>
> **This is fundamentally different from the asymptotic case $k \rightarrow \infty$, where negative eigenvalues vanish, and only the leading eigenvector survives.**
>
> Empirically, prior work **UGCN** [1] confirms this: 1-hop propagation is optimal for strongly homophilic graphs, and 2-hop propagation performs best under strong heterophily. **GPRGNN** [2] further shows that mixtures of $A, A^2, \dots, A^k$ capture diverse frequency components—a phenomenon impossible if all $A^k$ were purely low-pass.
>
> In our design, **PROP uses small $k$ (1–5), operating in a finite regime where**:
> - oversmoothing does not arise;
> - high-frequency components are preserved or amplified;
> - multiple-hop structures are encoded.
>
> Thus, **PROP’s strong performance on heterophilic graphs is consistent with spectral theory under finite powers of $\hat{A}$, and is not contradictory to classical low-pass intuition that applies in the infinite limit.**
>
> [1]. Jin, Di, et al. "Universal graph convolutional networks." NeurIPS, 2021.
>
> [2]. Chien, Eli, et al. "Adaptive universal generalized pagerank graph neural network." ICLR, 2021.
>
> ----

---

> ### Author Response · Authors · 2025-11-27
>
> **Q2.** In Table 1, GSSL methods performed severely worse than supervised methods. This is also against the consensus that GSSL methods are better than vanilla supervised GNN, like GCN.
>
> **A2.** Thank you for raising this concern. **The key reason is the difference in training split protocols.** Many contrastive-learning papers—e.g., DGI [3] and GRACE [4]—report supervised performance using very scarce labels (often 20 nodes per class). In such settings, GSSL offers strong gains because learned representations generalize better than direct supervised training. In our experiments, we adopt a 60/20/20 train–validation–test split, providing substantially more supervision. **This explains why the supervised baselines appear stronger than in prior reports, even though our GSSL results are competitive or above the original papers.** For example, GRACE on Cora: 83.3 ± 0.4 in [4] vs. 86.9 ± 1.0 in our experiments.
>
> **We emphasize that the supervised results are not the focus of the paper, but serve only as a reference baseline.** The central conclusion—that PROP performs competitively with existing GCL methods despite being training-free—remains valid regardless of supervised performance.
>
> [3]. Veličković, Petar, et al. "Deep graph infomax." arXiv:1809.10341 (2018).
>
> [4]. Zhu, Yanqiao, et al. "Deep graph contrastive representation learning." arXiv:2006.04131 (2020).
>
> ---
> **Q3.** In Figure 2 GCL w/o transformation, the loss nearly did not decrease, or the model was not trained. So, if GCL is without transformation, which parameters are learnt in it? Since the propagation part is just H_PROP, also without parameters.
>
> **A3.** Thank you for pointing out this ambiguity. In Figure 2, **we use the ChebNet polynomial expansion as the propagation operator**: $\sum_{k=0}^{K-1} \theta_k g_k(\mathbf{L})\mathbf{X}$, where $\mathbf{\theta} \in\mathbb{R}^K$ are the learnable propagation coefficients. Thus, even in the no-transformation setting, the propagation kernel contains trainable parameters, and the network is indeed being optimized. **We have updated the paper to clearly reflect this design to avoid confusion**.
>
> ---
>
> Thanks for your comments and hope our answers could address your concerns. Please let us know if there is more to clarify. We are happy to address them during the discussion stage.

---

### Official Review · Reviewer_3DvS · 2025-10-31

**Soundness:** 3
**Presentation:** 3
**Contribution:** 3
**Rating:** 4
**Confidence:** 4

**Summary:**

This paper introduces PROP, a training-free propagation operator that aggregates k-hop neighbor features, establishing it as a strong baseline for self-supervised node classification by showing its competitive performance across homophilic and heterophilic datasets, even outperforming many dedicated GCL methods on heterophilic benchmarks.

**Strengths:**

The paper challenges the prevailing paradigm in graph contrastive learning (GCL) that complex parameterized encoders are indispensable for high performance.
By decomposing contrastive loss into downstream-relevant and irrelevant components, it proves that PROPGCL outperforms both PROP and traditional GCL when objectives misalign.
The paper conducts multiple experimental evaluations to validate its claims.

**Weaknesses:**

PROP and PROPGCL depend on the propagation step K, but the paper lacks a systematic analysis of how K optimally adapts to diverse graph structures and provides no heuristic for K selection without brute-force tuning.
The paper omits comparisons with recent lightweight GCL methods (e.g., SimGCL, LiteGCL) that also prioritize efficiency.
PROPGCL does not integrate with graph attention mechanisms to prioritize important neighbors, limiting flexibility for graphs with unevenly relevant neighbors.
The paper compares PROPGCL with established GCL methods but omits newer lightweight GCL approaches that also prioritize efficiency.
The paper does not test PROPGCL on inductive benchmarks with heterogeneous node types or dynamic graph splits.

**Questions:**

For the training-free PROP operator, how does its performance scale with graph size and density?
The paper reveals that GCL-learned transformation weights perform no better than random weights, but only tests Gaussian, Uniform, Kaiming, and Xavier random initializations. Do other randomization strategies yield different results, and could they potentially outperform GCL-learned weights more significantly?
How does Fix-prop SL compare to dedicated few-shot GCL methods, and can further tuning of hyperparameters improve its performance?

---

> ### Author Response · Authors · 2025-11-27
>
> We thank Reviewer 3DvS for the valuable feedback. In response, we have conducted additional analyses and clarified several aspects of the paper. We summarize our responses and new results below.
>
> ----
>
> **Q1.** PROP and PROPGCL depend on the propagation step K, but the paper lacks a systematic analysis of how K optimally adapts to diverse graph structures and provides no heuristic for K selection without brute-force tuning.
>
> **A1.** **We provided the ablation study on the propagation step, covering multiple datasets (Appendix E in the original submission).** The results consistently show that shallow propagation (e.g., K=1 or K=2) achieves competitive performance, and further increasing K does not lead to noticeable gains. This aligns with widely observed patterns in graph learning that most graphs benefit from limited ranges of message passing before oversmoothing occurs. **This indicates that PROP can be used effectively without intensive hyperparameter tuning.**
>
> ----
>
> **Q2.** The paper omits comparisons with recent lightweight GCL methods (e.g., SimGCL, LiteGCL) that also prioritize efficiency.
>
> **A2.** We thank the reviewer for pointing this out. **We include direct comparisons with SimGCL [1], which is explicitly designed to reduce augmentation overhead.** We were unable to locate the manuscript or reference for LiteGCL, and would appreciate it if the reviewer could provide the citation so we may incorporate it.
>
> SimGCL reduces the cost of heavy graph augmentations by replacing them with uniform embedding noise. In contrast, PROPGCL removes transformation weights entirely, thus eliminating both forward and backward propagation associated with parameterized transformations, which is the dominant computation in most GCL architectures.
>
> ***Table R1*** summarizes the time savings in epoch-level training:
> - SimGCL reduces training time by 17.51% on average, while
> - PROP-GRACE achieves 61.05% reduction, largely due to the elimination of transformation modules.
>
> This demonstrates that the core source of efficiency is different: SimGCL optimizes augmentation, while **PROPGCL fundamentally simplifies the representation operator itself, leading to a deeper reduction in computation**. We have added this comparison in the revised paper.
>
> ***Table R1. Time efficiency comparison, with the percentage denoting the decrease of the epoch time consumption (seconds).***
> | Method | Photo | Computers | CS | Squirrel | Chameleon | *Average* |
> |-|-|-|-|-|-|-|
> | GRACE | 0.2872 | 0.4639 | 1.5111 | 0.7004 | 0.2295 | 0.6384 |
> | SimGCL | 0.2637  (↓8.18%) | 0.3947 (↓14.91%) | 1.0329 (↓31.64%) | 0.6374 (↓8.99%) | 0.1893 (↓17.51%) | 0.5036 (↓21.11%) |
> | PROP-GRACE | 0.2400 (↓16.44%) | 0.3626 (↓21.84%) | 0.2374 (↓84.29%) | 0.2581 (↓63.15%) | 0.1450 (↓36.82%) | 0.2486 (↓61.05%) |
>
> [1]. Yu, Junliang, et al. "Are graph augmentations necessary? simple graph contrastive learning for recommendation." SIGIR. 2022.
>
> ----
>
> **Q3.** PROPGCL does not integrate with graph attention mechanisms to prioritize important neighbors, limiting flexibility for graphs with unevenly relevant neighbors.
>
> **A3.** We appreciate this point. PROPGCL is intentionally designed around **polynomial spectral filters (e.g., Chebyshev)**, which perform **global structural aggregation rather than local, feature-conditioned attention**. This fixed nature is essential to achieving **low computational overhead and stable optimization**, which are central to our contribution.
>
> We acknowledge that attention-based GCLs may better accommodate local semantic relevance. However, our empirical results (Section 6.3) show that **even without attention, PROPGCL achieves competitive or superior performance on most benchmarks**, indicating that a strong structural prior can be sufficient for many graph representation tasks.
>
> We view this not as a limitation but as a design trade-off:
> - Attention-based methods prioritize expressivity at a higher computational and optimization cost;
> - PROPGCL prioritizes efficiency, simplicity, and robustness—while still delivering strong accuracy.
>
> We believe the community may explore hybrid models (attention + spectral propagation), and we view this as an exciting direction beyond this work.
>
> ----

---

> ### Author Response · Authors · 2025-11-27
>
> **Q4.** The paper does not test PROPGCL on inductive benchmarks with heterogeneous node types or dynamic graph splits.
>
> **A4.** We included inductive evaluations on both single-graph and multi-graph benchmarks in **Appendix C of the original submission**.
> - Reddit (single graph, inductive):
>  PROP (K=2) achieves 0.8452 F1, outperforming GRACE (0.8185).
> - PPI (multiple graphs, inductive):
>  PROP (K=2) achieves 0.7527, comparable to GRACE (0.7548) without additional parameters.
>
> **These results (Appendix C, Table 9) confirm that propagation-based representations remain effective under inductive settings.**
>
> **Appendix D** evaluated models under alternative public splits differing from Section 4.1. Across 10 benchmarks, PROP achieves the best performance on 6, and surpasses the runner-up ProGCL by an average of 4.23%. These results indicate that the method is robust not only to inductive settings, but also to data partitioning and distributional variations.
>
> ---
>
> **Q5.** For the training-free PROP operator, how does its performance scale with graph size and density?
>
> **A5.** We thank the reviewer for this question. To systematically examine scalability, we conduct controlled simulations using Erdős–Rényi graphs, which allow independent control of graph size and sparsity. To generate informative but non-trivial features, we construct node representations by combining topological statistics (degree and clustering coefficient) with community assignments (via spectral clustering), and we inject Gaussian noise (std = 1.0) to avoid feature leakage. Labels are assigned based on the community structure.
>
> As shown in ***Table R2***, **PROP scales robustly across graph sizes up to 20,000 nodes when sparsity remains within realistic ranges (0.0001–0.02)**. For example, at 20,000 nodes and sparsity 0.02, PROP achieves 97.30% accuracy. We observe performance degradation when sparsity becomes high (≥ 0.05), which we believe is natural: dense neighborhoods reduce the discriminative value of local propagation and lead to oversmoothing effects similar to those observed in GNNs with deep message passing.
>
> Importantly, **most real-world benchmarks fall well below this density range.** ***Table R3*** shows that typical graph sparsity is between 0.0002 and 0.016, a regime in which PROP consistently performs well. These findings suggest that PROP is well-suited to practical graph scales.
>
> ***Table R2. Performance (%) of PROP when varying graph size and graph sparsity.***
> | Size \ Sparsity | 0.0001 | 0.005 | 0.01 | 0.02 | 0.05 | 0.1 | 0.2 |
> | - | - | - | - | - | - | - | - |
> | 500 | 100.00 ± 0.00 | 100.00 ± 0.00 | 99.87 ± 0.09 | 95.54 ± 1.18 | 95.90 ± 1.00 | 96.70 ± 0.80 | 93.90 ± 1.90 |
> | 1000 | 100.00 ± 0.00 | 99.98 ± 0.03 | 94.40 ± 0.85 | 96.75 ± 0.90 | 96.00 ± 0.55 | 95.40 ± 0.80 | 93.55 ± 1.00 |
> | 5000 | 99.99 ± 0.01 | 96.50 ± 0.27 | 96.70 ± 0.33 | 96.47 ± 0.23 | 96.97 ± 0.45 | 70.36 ± 0.33 | 68.33 ± 0.83 |
> | 10000 | 100.00 ± 0.00 | 97.39 ± 0.29. | 97.18 ± 0.31 | 97.30 ± 0.24 | 69.97 ± 0.38 | 68.71 ± 0.45 | 67.76 ± 0.45 |
> | 20000 | 100.00 ± 0.00 | 97.79 ± 0.20 | 98.02 ± 0.17 | 96.75 ± 0.23 | 68.60 ± 0.34 | 68.66 ± 0.21 | 68.29 ± 0.39 |
>
> ***Table R3. Sparsity of common real-world graph benchmarks.***
> | Dataset | Sparsity |
> | - | - |
> | PubMed | 0.0002 |
> | CiteSeer | 0.0008 |
> | Cora | 0.0014 |
> | Computers | 0.0026 |
> | Photo | 0.0041 |
> | Chameleon | 0.0139 |
> | Squirrel | 0.0160 |
>
> ----

---

> ### Author Response · Authors · 2025-11-27
>
> **Q6.** The paper reveals that GCL-learned transformation weights perform no better than random weights, but only tests Gaussian, Uniform, Kaiming, and Xavier random initializations. Do other randomization strategies yield different results, and could they potentially outperform GCL-learned weights more significantly?
>
> **A6.**  We extend our evaluation to two additional initialization strategies: **orthogonal initialization** [2] and **LeCun initialization** [3]. As summarized in ***Table R4***, these schemes maintain the same overall trend that Orthogonal achieves 73.96% average accuracy and LeCun achieves 74.10% average accuracy. **Both are statistically indistinguishable from the GCL-learned transformation weights.** This observation further supports our central finding: the transformation module in GCL does not consistently learn meaningful structure beyond what random matrices already provide.
>
> We would like to emphasize that **our goal is not to argue that random initializations outperform GCL.** Instead, the comprehensive study across six widely used schemes (Gaussian, Uniform, Kaiming, Xavier, Orthogonal, and LeCun) demonstrates that **GCL-learned transformation weights fail to deliver consistent gains over random baselines.** This reinforces our argument that transformation components are often a bottleneck in GCL design.
>
> ***Table R4. Test accuracy (%) of node classification benchmarks comparing the GCL-learned transformation weights and different random weights.***
> |Training | Cora | CiteSeer | PubMed | Squirrel | Chameleon | Texas | Wisconsin | Cornell | Mean |
> | - | - | -| - | -| -| - | - | - |- |
> | GCL | 84.79 +- 0.80 | 75.47 +- 0.76 | 82.25 +- 0.24 | 40.74 +- 0.61 | 58.99+-1.40 | 80.33 +- 1.97 | 87.00 +- 2.50 | 80.33 +- 1.80 | 73.74 |
> | Orthogonal-random | 84.95 +- 1.12 | 76.10 +- 0.71 | 83.55 +- 0.18 | 42.11 +- 0.85 | 57.88 +- 1.15 | 79.50 +- 3.88 | 87.13 +- 2.75 | 79.80 +- 3.20 | 73.87 |
> | Lecun-random | 84.88 +- 0.95 | 75.92 +- 0.80 | 84.05 +- 0.20 | 42.55 +- 0.92 | 58.01 +- 1.35 | 80.50 +- 3.50 | 87.30 +- 2.90 | 80.11 +- 3.10 | 74.16 |
>
> [2]. Saxe, Andrew M., James L. McClelland, and Surya Ganguli. "Exact solutions to the nonlinear dynamics of learning in deep linear neural networks." arXiv preprint arXiv:1312.6120 (2013).
>
> [3] .LeCun, Yann, et. al,  "Efficient BackProp", Neural Networks: Tricks of the Trade (1998).
>
> ---
>
> **Q7.** How does Fix-prop SL compare to dedicated few-shot GCL methods, and can further tuning of hyperparameters improve its performance?
>
> **A7.** To provide a fair comparison, we further tune Fix-prop SL hyperparameters (propagation steps, learning rate, hidden dimension, dropout) and evaluate on the 3-shot setting. We compare against **Meta-GNN** [4], a method explicitly designed for meta-learning in few-shot node classification. As shown in ***Table R5***, our Fix-prop-GCN achieves 70.43% on Cora and 58.62% on CiteSeer, outperforming both the supervised GCN baseline (39.19% / 37.52%) and its GCL counterpart GRACE-GCN (64.46% / 55.85%). While Meta-GNN achieves higher accuracy (e.g., 77.19% on Cora), it employs specialized meta-learning strategies, whereas Fix-prop SL uses no meta-optimization.
>
> **We view our few-shot experiments as initial evidence that propagation-based representations transfer well even in extremely low-label scenarios, but not as a complete few-shot solution.** Developing a principled meta-learning adaptation for PROPGCL is a promising direction that we intentionally leave for future work to maintain focus on our core contributions.
>
> ***Table R5. Test accuracy (%) of node classification benchmarks in the 3-shot scenarios. Results of Meta-GNNs are copied from the original paper.***
> | Methods | Cora | CiteSeer |
> | - | - | - |
> | Supervised-GCN  | 39.19 +- 3.96 |37.52 +- 2.25 |
> | GRACE-GCN | 64.46 +- 4.34 | 55.85 +- 5.15 |
> | Meta-GCN | 77.19  | 68.65 |
> | Meta-SGC | 76.78 | 69.43 |
> | Fix-prop-GCN (ours) | 70.43 +- 2.86 | 58.62 +- 3.20 |
>
> [4]. Zhou, Fan, et al. "Meta-gnn: On few-shot node classification in graph meta-learning." Proceedings of the 28th ACM international conference on information and knowledge management. 2019.
>
> ---
>
> Thanks for your comments and hope our answers could address your concerns. Please let us know if there is more to clarify. We are happy to address them during the discussion stage.

---

### Official Review · Reviewer_os8L · 2025-11-01

**Soundness:** 3
**Presentation:** 3
**Contribution:** 2
**Rating:** 4
**Confidence:** 3

**Summary:**

This paper claims that the feature transformation component of GNNs is detrimental in Graph Contrastive Learning, performing no better than random weights. The authors propose PROPGCL, which removes the transformation layer entirely and only learns propagation coefficients, and can outperform complex GCL encoders. The method is shown to be both effective, particularly on heterophilic graphs, and computationally efficient.

**Strengths:**

1. The paper analyzes a problem in GCL where complex transformation layers are often poorly learned and perform no better than random weights.
2. The proposed method is evaluated on heterphily graph benchmarks.

**Weaknesses:**

1. The novelty is a bit limited - the propagation of graph node features serves as a good representation is well studied in the literature
2. The paper excels at showing that the transformation fails but provides little insight into why. Why is the GCL objective sufficient for learning propagation coefficients $\theta$ but not transformation weights $W$?

**Questions:**

1. Why is the GCL objective sufficient for learning propagation coefficients $\theta$ but not transformation weights $W$?

---

> ### Author Response · Authors · 2025-11-27
>
> We appreciate Reviewer os8L for the careful reading and the constructive comments. We summarize and address your comments as follows.
>
> ---
>
> **Q1.** The novelty is a bit limited - the propagation of graph node features serves as a good representation is well studied in literature.
>
> **A1.** We sincerely thank the reviewer for the comment. We agree that graph feature propagation as an operator is well established. Our contribution, however, is not simply introducing another propagation mechanism. Rather, **we offer new theoretical insights and a systematic re-evaluation of GCL design that have not been previously studied.** Our novelty lies in the following three aspects:
> 1. **Revisiting PROP as a strong, overlooked self-supervised baseline for GCL.**
>  While propagation has existed in earlier literature, prior work has not systematically established PROP as a competitive baseline in the GCL setting. Through extensive evaluation (Section 4), we show that PROP alone can outperform or match many GCL methods, and we provide a theoretical link (Theorem 4.2) showing that its effectiveness arises from implicit alignment.
> 2. **Identifying a failure mode of learned transformations in GCL.**
>  We are, to our knowledge, the first to show that GCL-learned transformation weights are often ineffective, performing no better than random initialization under a variety of training schemes (Table 2, Section 5). This directly challenges current assumptions that transformation modules are essential for representation quality in GCL.
> 3. **Introducing the PROPGCL framework.**
>  Built on these insights, PROPGCL excludes transformation and only uses a training-free propagation kernel. This leads to state-of-the-art or competitive performance while substantially reducing model complexity (Section 6). We further provide a theoretical justification via optimal CL decomposition, clarifying why this design is not only empirically strong but also principled.
>
> Overall, our contribution is not in reintroducing propagation, but in revealing an underexplored failure mode of GCL architectures, grounding it theoretically, and providing a practical, efficient alternative that performs reliably across node classification benchmarks. We believe this insight is both non-trivial and impactful for the community.
>
> ---

---

> ### Author Response · Authors · 2025-11-27
>
> **Q2.** The paper excels at showing that the transformation fails but provides little insight into why. Why is the GCL objective sufficient for learning propagation coefficients but not transformation weights?
>
> **A2.** Thank you for this insightful question. We agree that explaining why this failure occurs is crucial, and we address both the intuition and the theoretical mechanism.
>
> ---
>
> **Intuition.**
>
> In contrastive learning for images, augmentation priors are straightforward (e.g., color jitter, rotations), and transformations help capture invariances. In graphs, however, the dominant **structural prior** mainly comes from neighborhood relationships. **Propagation naturally exploits this structural prior by aggregating informative neighborhood signals. By contrast, the transformation module must learn these relational patterns from scratch, which often redundantly re-parametrizes what propagation has already captured.** This is reflected in our randomness experiments (Section 5.1, Table 2), where learned transformation weights behave nearly indistinguishably from random matrices.
>
> Moreover, when the contrastive objective is misaligned with downstream semantics, **the high-capacity transformation tends to overfit to contrastive artifacts** rather than meaningful graph signals. We empirically observe this degradation in Section 5.3 (Figure 2).
>
> ---
>
> **Theory.**
>
>  In **Section 6.2**, we further formalize this intuition. We decompose the optimal operator under the contrastive objective: $T^* = \arg\min_T \mathcal{L}_{CL}(T \cdot X) = f + g$, where $f$ represents continuous, downstream-relevant information, and $g$ corresponds to discontinuous or nuisance signals that the CL objective may inadvertently emphasize.
>
> Leveraging Chebyshev polynomial approximation, we show that:
>
> ***Theorem 6.1.*** When $\alpha > \frac{\epsilon_f}{\vert f \vert_F}$, we have
> $$\vert H_{\text{PROPGCL}} - f X \vert_F < \min ( \vert H_{\text{PROP}} - f X \vert_F,  \vert H_{\text{GCL}} - f X \vert_F ).$$
>
> This result indicates that **under contrastive–downstream misalignment (large $\alpha$), propagation filters preserve downstream-relevant components $f$, while transformation modules tend to fit discontinuous noise $g$**. This provides a principled explanation for the empirical behavior we observe: PROPGCL are inherently more robust, whereas transformation modules are vulnerable to overfitting.
>
> ----
>
> Thanks for your comments and hope our answers could address your concerns. Please let us know if there is more to clarify. We are happy to address them during the discussion stage.

---

### Author Response · Authors · 2025-12-01
**Rebuttal Summary**

Dear Program Chairs, Senior Area Chairs, Area Chairs, and Reviewers,

We sincerely thank the Area Chairs and Reviewers for their time, careful evaluation, and constructive comments. We provide below a concise summary of the discussion phase and our corresponding revisions, with the goal of assisting the Area Chair in tracking the progress of our responses.

---

### **Theoretical Improvements (Reviewer 8KxH, Q2)**

We revised ***Definition 6.1*** and ***Assumption 6.2*** to explicitly accommodate both heterophilic and homophilic regimes. The revisions clarify that PROPGCL focuses on continuous graph signals—regardless of whether they are low- or high-frequency. This addresses the reviewer’s concern that the original formulation did not fully explain performance in heterophilic settings.

---

### **Clarifications of Methodological Concerns**

**(a) Spectral interpretation on heterophilic performance (Reviewer rEYN, Q1).**

 We appreciate the reviewer’s observation regarding $A^kX$ and low-pass filtering. We clarify that:
- Classical “low-pass” intuition applies to the ***infinite*** smoothing limit,
- Whereas PROPGCL operates under ***finite*** propagation, where bounded powers of $\hat{A}$ remain consistent with spectral theory and can retain task-relevant high-frequency components.

**(b) Experimental settings (Reviewer rEYN, Q2–Q3).**

 We have clarified the settings of:
- ***Table 1***, to detail the splitting protocol and avoid ambiguity;
- ***Figure 2***, to clearly describe the propagation operator.

These revisions remove potential sources of misunderstanding and ensure full reproducibility.

**(c) Scope of the paper (Reviewer 8KxH, Q1).**

Following the reviewer’s guidance, we have narrowed the title and claims to ***node-level*** tasks and clarified throughout the paper that graph-level tasks are beyond our intended scope.

**(d) Contribution of PROP (Reviewer os8L, Q1; Reviewer 8KxH, Q3).**

- We revisit PROP as a strong, overlooked self-supervised baseline for GCL.
- PROP is not evidence that the GCL paradigm is unnecessary, but rather evidence that many current parametric designs may be overcomplicated.

We adjusted the framing in Sections 4.3 to better reflect this nuance.

**(e) Why does the GCL objective sufficiently learn propagation but not transformation (Reviewer os8L, Q2).**

- ***Section 5.1 and 5.3*** provide the intuition. Propagation inherently aligns with structural priors and preserves downstream-relevant components, whereas heavier transformation modules tend to rediscover already encoded structure and may be more susceptible to contrastive–downstream misalignment.
- The revised ***Theorem 6.1*** provides theoretical grounding for this behavior.

---

### **Additional Experimental Results**

**(a) Lightweight GCL comparison (Reviewer 3DvS, Q2).**

 We have added comparisons with ***SimGCL***, showing a deeper reduction in computation while maintaining competitive accuracy. This quantifies the practical efficiency gains of PROP.

**(b) Scaling with graph size and density (Reviewer 3DvS, Q5).**

We now include synthetic scaling experiments. PROP scales robustly across graph sizes up to 20,000 nodes when sparsity remains within realistic ranges.

**(c) Extended random initializations (Reviewer 3DvS, Q6).**

 Beyond the four initializations in the original submission, we evaluate ***orthogonal*** and ***LeCun initialization***.
 Both produce representation quality that is statistically indistinguishable from GCL-learned transformation weights.

**(d) Statistical significance tests (Reviewer 8KxH, Q5).**

Across all datasets, a ***paired t-test*** shows no statistically significant difference between GCL-learned weights and random weights (t = −0.613, p = 0.5594). This supports our claim that heavier transformation layers provide limited practical benefit.

---

### **Additional Clarifications**

For concerns regarding:
- Appropriate propagation depth (Reviewer 3DvS, Q1),
- Inductive benchmarks (Reviewer 3DvS, Q4),
- Ablation of basis functions (Reviewer 8KxH, Q6),

We point reviewers to ***Appendix E***, ***Appendix C***, and ***Appendix L***, respectively, where these issues were addressed in detail.

---

### Meta-Review · Area_Chair_HBQK · 2026-01-07

**Summary:**

While the rebuttal improves clarity and adds several missing experiments, the core contribution still reads as “a strong propagation baseline plus removing the transformation module” in a space where propagation-only and spectral/PR-style baselines are already well-studied. The paper’s strongest empirical message is that transformations can overfit the contrastive objective and harm downstream performance, but this is not yet established as a broadly new principle rather than a setting-dependent observation. Given the limited novelty and the narrowed scope (now explicitly node-level only), the work does not clear the bar for ICLR.

**Reviewer Concerns:**

Addressed concerned:

1. Scope narrowed to node-level tasks (8KxH).
2. Added heterophily explanation via finite propagation and revised definitions/assumptions (rEYN, 8KxH).
3. Added SimGCL comparison, scaling experiments, extra random inits, and a paired t-test (3DvS, 8KxH).
4. Added downstream accuracy curves showing overfitting with transformation (8KxH).

---

Remaining concerns:

1. Novelty remains limited: Propagation-as-representation and polynomial/spectral propagation are established; the paper’s “unleashing propagation” framing overstates what is new (os8L, rEYN).
2. Mechanistic claim not fully nailed down: The “transformation is inadequately learned / no better than random” story is supported on their 3. benchmarks, but it is not convincingly generalized into a robust principle across objectives, architectures, and training protocols.
4. Model selection issue remains: K (and related choices) is still handled by ablations and shallow heuristics, with no principled selection rule (3DvS).
5. Narrow impact: Once restricted to node classification, the remaining gain over existing simple baselines is not compelling enough for acceptance.

**Reviewer Scores:**

Given the lack of discussion, it is hard to predict the scores. However, I think that many of the gaps still remain, and thus my assessment of the predicted score is as follows:

8KxH: 6 ---> 6

3DvS: 4 ---> 4 (additional experiments help, but core limitations remain)

os8L: 4 ---> 4

rEYN: 2 ---> 2 or 3 (clarified confusion, but still rejects on perceived implausibility/novelty)

---

### Decision · Program_Chairs · 2026-01-26

Reject